# LLM-Driven Algorithm Design for Quantum Circuit Synthesis based on Binary Decision Diagrams

## Abstract

Quantum circuits are central to implementing quantum algorithms on quantum devices, where quantum gates must be reversible. Many quantum algorithms rely on Boolean functions, which must therefore be implemented reversibly within quantum circuits. Reversible circuit synthesis provides a way to translate such Boolean functions into reversible circuits. Binary decision diagrams (BDDs) offer a scalable approach to this task, but the resulting BDDs and circuits depend heavily on variable ordering. Existing ordering heuristics commonly minimize BDD size because it is closely tied to the circuit size. However, BDD size is an imperfect proxy for the quantum cost of the synthesized circuit (QCC). We propose `QuantumEvo`, an evolutionary framework that uses an LLM as a heuristic generator for QCC-aware BDD variable ordering. Instead of predicting orderings directly, `QuantumEvo` searches over ordering heuristics initialized from multiple heuristic families. Candidate heuristics directly manipulate variable orderings using standard BDD operations and are selected by downstream QCC. The discovered heuristic, HGA-QE, modifies the sifting step inside a genetic algorithm so that the procedure is better aligned with QCC. Across the benchmark set, HGA-QE achieves a 70.9% tie-or-win rate against the per-function best baseline and is strictly best on 13.5% of the functions. The results demonstrate broadly competitive QCC performance, with HGA-QE showing a clearer relative advantage in strict wins on the two benchmark suites drawn from sources different from the data used for heuristic discovery

## 1 Introduction

Quantum computing has attracted increasing attention with advances in both quantum algorithms and quantum hardware (Fedorov et al., 2022; Blekos et al., 2024; Bluvstein et al., 2024). To run a quantum algorithm on a quantum device, its operations must be expressed as a quantum circuit, i.e., a cascade of quantum gates. Since quantum operations are unitary, quantum circuits are *reversible* in nature. Many quantum algorithms contain Boolean components, such as oracle functions (Grover, 1996; Simon, 1997) and arithmetic subroutines (Vedral et al., 1996). To implement such components in a quantum circuit, a Boolean function must be embedded into a reversible transformation. Reversible circuit synthesis addresses this task by translating Boolean functions into reversible circuits that can be mapped to a target gate library (Shende et al., 2003; Wille et al., 2016). The quality of a synthesized circuit is often evaluated by its quantum cost, which reflects the cost of implementing the circuit under a quantum gate library. We refer to this metric as the quantum cost of the synthesized circuit (QCC).

Several approaches have been developed for reversible circuit synthesis, including transformation-based (Miller et al., 2003) and cycle-based (Saeedi et al., 2010). Such methods can produce high-quality

circuits for small functions, but their scalability is limited as the number of inputs grows. A BDD-based synthesis aims to improve scalability by representing a Boolean function as a binary decision diagram (BDD) and translating the resulting BDD structure into a reversible circuit (Wille et al., 2009). Since variable ordering strongly affects the resulting BDD, and the BDD size provides an upper bound on the circuit size, minimizing the BDD size is a common objective for ordering methods. Previous studies on variable ordering methods, including metaheuristics (Bollig et al., 1995; Awad et al., 2022) and learning-based methods (Miao et al., 2026), have used the BDD size as an objective. However, BDD size is an imperfect proxy for QCC. A smaller BDD does not necessarily imply a lower QCC. This motivates the search for BDD variable ordering heuristics that directly reduce downstream QCC.

We formulate this as a heuristic design problem. The goal is to discover a reusable BDD variable ordering heuristic that maps each Boolean function to a variable ordering with low QCC. Recent LLM-driven algorithm design methods use LLMs to explore a broader space of heuristic programs beyond a predefined set (Novikov et al., 2025; Ye et al., 2024; Liu et al., 2026; Huang et al., 2026). Building on this idea, we propose `QuantumEvo`[1], an evolutionary framework that uses an LLM as a heuristic generator. It initializes the search from multiple heuristic families and evaluates executable heuristics that manipulate variable ordering using standard BDD operations.

Our main contributions are as follows: (1) We formulate QCC-aware BDD variable ordering as a heuristic design problem and introduce `QuantumEvo`, an LLM-driven heuristic search framework that evaluates executable ordering heuristics using downstream QCC rather than BDD size; (2) `QuantumEvo` discovers HGA-QE, an effective hybrid genetic algorithm that replaces standard sifting with an LLM-discovered targeted sifting procedure better aligned with downstream QCC. Across the benchmark set, HGA-QE demonstrates broadly competitive QCC performance against both classical and learning-based baselines, with a clearer relative advantage in strict wins on the two benchmark suites drawn from sources different from the data used for heuristic discovery.

## 2 Background and Related Work

This section reviews the background and related work relevant to our approach.

### 2.1 Reversible Logic and Circuit Synthesis

A Boolean function $f : \mathbb{B}^n \to \mathbb{B}^m$ maps $n$ binary input bits to $m$ binary output bits, where $\mathbb{B} = \{0, 1\}$. A reversible function is a bijection with the same number of input and output bits. While general Boolean functions are not necessarily bijective, they can be embedded into reversible functions by introducing ancilla inputs and garbage outputs (Wille & Drechsler, 2010). Due to its information-preserving property, reversible logic is relevant to a range of applications, most notably quantum computing, low power design, optical computing, and nanotechnologies (Landauer, 1961; Saeedi & Markov, 2013; Cuykendall & Andersen, 1987; Merkle, 1993).

Reversible circuit synthesis realizes Boolean functions as circuits over a gate library. We focus on the widely used *NCT library*, which comprises NOT, CNOT, and Toffoli gates; see Appendix A.1. Each gate in the NCT library with $k \in \{0, 1, 2\}$ control bits is assigned a QCC of $\text{Cost}(k) = 2k^2 - 2k + 1$. based on its decomposition into elementary quantum gates, following Szyprowski & Kerntopf (2013). The total QCC of a circuit is the sum of its gate costs.

---

[1]An anonymized implementation is available at `https://anonymous.4open.science/r/QuantumEvo-Circuit`.

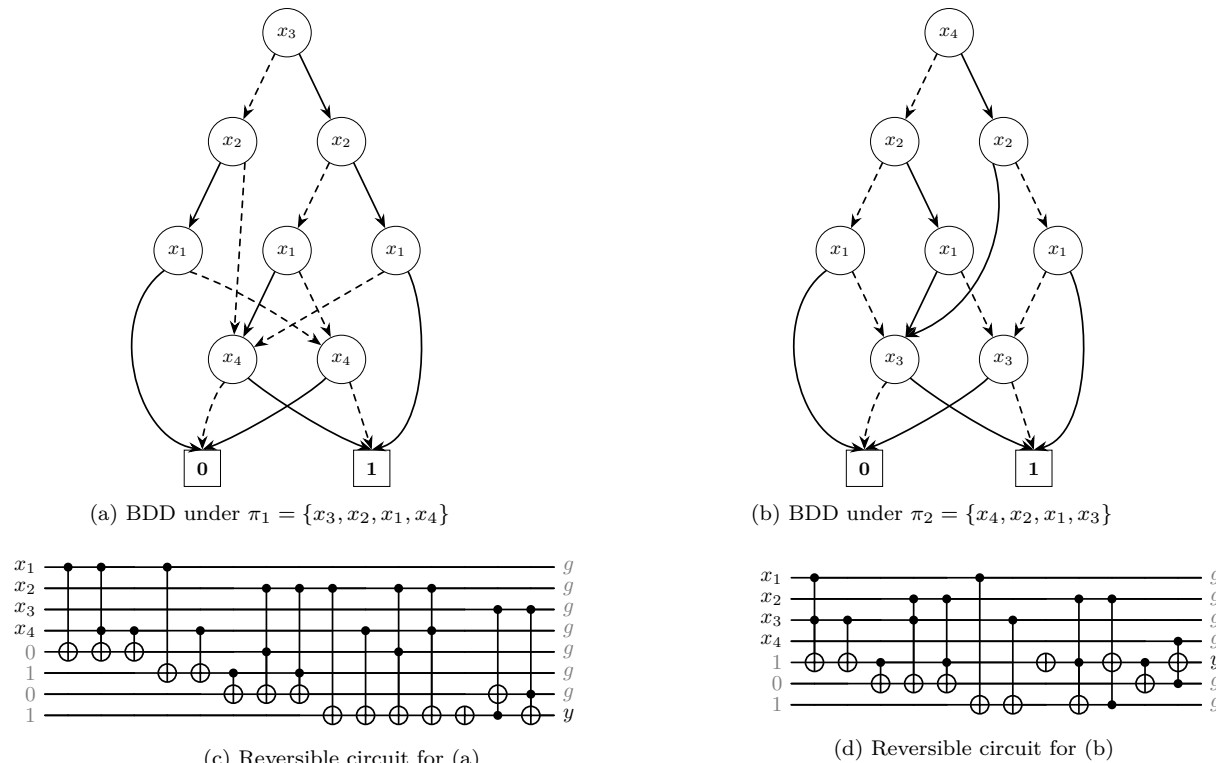

(a) BDD under $\pi_1 = \{x_3, x_2, x_1, x_4\}$

(b) BDD under $\pi_2 = \{x_4, x_2, x_1, x_3\}$

(c) Reversible circuit for (a)

(d) Reversible circuit for (b)

Figure 1: Impact of variable ordering on QCC for the function `sf_232`. Solid/dashed edges denote 1/0-branches. BDD size is 8 for both orderings, whereas QCC is 43 for $\pi_1$ and 36 for $\pi_2$. Here, $y = x_2 \oplus x_3 \oplus x_4 \oplus x_1 \cdot (x_2 \oplus x_3 \oplus x_2 x_3 \oplus x_2 x_4)$

.

Before synthesis, a Boolean function must be represented in a form suitable for manipulation. Although a truth table provides a canonical representation of Boolean functions by enumerating all $2^n$ rows, they become impractical at scale due to their exponential size. BDDs provide compact intermediate representations for scalable reversible circuit synthesis (Wille et al., 2009; Soeken et al., 2016). Following Wille et al. (2009), the synthesis procedure considered in this work constructs reversible circuits by replacing each non-terminal BDD node with a cascade of reversible gates.

## 2.2 Binary Decision Diagrams and Variable Ordering

A BDD is a directed acyclic graph representation of a Boolean function (Bryant, 1986). Each non-terminal node is labeled by one variable and has two branches pointing to terminal or non-terminal child nodes. When the same subfunction appears in multiple places, it is represented once and reused as a shared subgraph. The variable ordering is important when constructing a BDD, as it strongly affects the size of the resulting graph. The BDD size is typically measured by the number of non-terminal nodes, and finding an optimal ordering that minimizes this size is $\mathcal{NP}$-complete (Bollig & Wegener, 1996). Although the BDD size is a natural proxy for the QCC because the size of the resulting circuit is bounded by the BDD size (Wille et al., 2009), it still remains an indirect proxy. Figure 1 shows that two orderings of the same function can produce BDDs with the same size but different QCCs.

Prior work on BDD variable ordering includes exact algorithms (Friedman & Supowit, 1987; Jeong et al., 1993), heuristic algorithms such as sifting (Rudell, 1993; Panda et al., 1994; Panda & Somenzi, 1995; Meinel et al., 1997), and metaheuristics including genetic algorithms (GA), simulated annealing (SA), and swarm-based search (Drechsler et al., 1996; Bollig et al., 1995; Awad et al., 2022). More recently, BDD2Seq (Miao et al., 2026) introduced a learning-based approach that trains a graph-to-sequence model

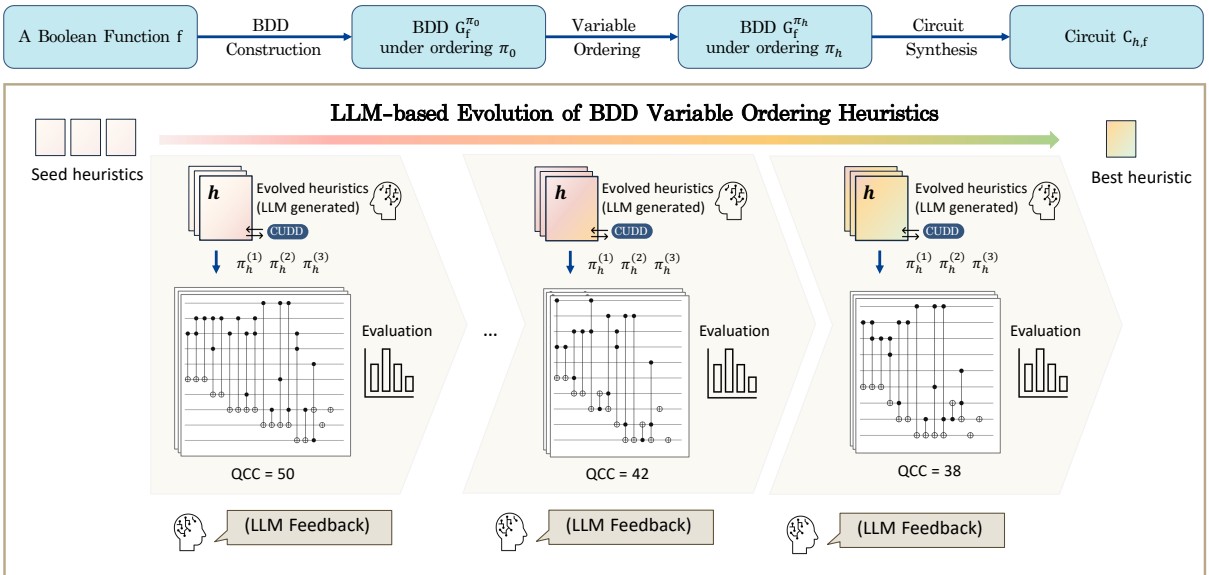

Figure 2: Overview of the `QuantumEvo` framework. Each individual heuristic reorders the initial variable order of BDD, and the resulting circuit is evaluated by QCC.

using supervised learning to predict the orderings. However, all of these methods primarily select the orderings based on BDD size, while downstream QCC is observed only after synthesis. This motivates a QCC-aware objective that optimizes the orderings by the QCC rather than by the BDD size.

## 2.3   Related Work

We review the LLM-based approaches related to our work. The first line examines how LLMs have been used in quantum-computing workflows. Recent Quantum–LLM studies address broad problem settings within the quantum-computing workflow. Some works evaluate LLMs on generating quantum code from textual task prompts (Vishwakarma et al., 2024; Guo et al., 2025). Others fine-tune LLMs to generate parameterized quantum circuits for quantum optimization tasks (Jern et al., 2025). QCircuitBench (Yang et al., 2026) benchmarks LLMs on quantum algorithm design tasks, including oracle construction and the implementation of standard quantum algorithms. Oracle construction is related to our setting because it often requires implementing Boolean functions as reversible circuits. However, QCircuitBench focuses mainly on textbook-level or algorithm-specific oracles, whereas `QuantumEvo` discovers BDD variable ordering heuristics for synthesizing large Boolean functions under downstream QCC as the objective.

The second covers LLM-driven algorithm design, which provides the methodological basis for our approach. A complementary line of work uses LLMs to generate heuristic programs rather than predict solutions directly. FunSearch (Romera-Paredes et al., 2024) introduced LLM-driven algorithm search for mathematical and heuristic design, and ReEvo (Ye et al., 2024) extended it with reflective evolution for combinatorial optimization. Recent frameworks have further advanced LLM-driven algorithm design (Zheng et al., 2025; Novikov et al., 2025; Ye et al., 2025; Yao et al., 2025; Dat et al., 2025; Liu et al., 2026; Huang et al., 2026). This paradigm has also been applied to domain-specific heuristic design (Zhao et al., 2026; Shi & Zhen, 2026; Wang et al., 2026). `QuantumEvo` extends this line of domain-specific work to BDD variable ordering for reversible circuit synthesis.

## 3 Methodology

This section describes our methodology. We first define the objective used to evaluate BDD variable orderings in terms of QCC. We then introduce `QuantumEvo`, an LLM-driven evolutionary framework for discovering BDD variable ordering heuristics. Figure 2 gives an overview of the framework.

### 3.1 Problem Formulation

Given a Boolean function $f : \mathbb{B}^n \to \mathbb{B}^m$, let $\mathcal{H}$ denote the search space of candidate heuristics for BDD variable ordering. An initial BDD $G_f^{\pi_0}$ is first constructed under an initial variable ordering $\pi_0$. A heuristic $h \in \mathcal{H}$ then reorders the BDD and produces a variable ordering $\pi_h$ for $f$. The reordered BDD $G_f^{\pi_h}$ is synthesized into a reversible circuit by a fixed BDD-based synthesis procedure $\mathcal{S}$ following Wille et al. (2009), as implemented in RevKit (Soeken et al., 2012). The resulting circuit is denoted by $C_{h,f} = \mathcal{S}\left(G_f^{\pi_h}\right)$. Let $q(C)$ denote the QCC of a circuit $C$. For the given Boolean function $f$, the goal is to find a heuristic that minimizes the QCC of the synthesized circuit within a time budget:

$$
\begin{aligned}
h^* = \arg\min_{h \in \mathcal{H}} \quad & q(C_{h,f}) \\
\text{s.t.} \quad & t_{\text{syn}}(h, f) \leq \tau_U,
\end{aligned}
\tag{1}
$$

where $t_{\text{syn}}(h, f)$ denotes the total time required to construct the BDD for $f$ using the ordering produced by heuristic $h$ and synthesize the resulting circuit, and $\tau_U$ is the time budget.

### 3.2 QuantumEvo: LLM-driven Evolutionary Heuristic Design

`QuantumEvo` primarily builds on the reflective evolutionary search framework of ReEvo (Ye et al., 2024), in which LLMs generate executable heuristic programs that are iteratively improved through evaluation and feedback. Starting from a set of seed heuristics, the LLM proposes new heuristics by revising existing ones, guided by natural-language feedback on previous fitness results. We instantiate this paradigm for BDD variable ordering through three domain-specific design choices that shape the search process: how individual heuristics are represented, how the initial heuristics are constructed, and how the fitness function is defined.

**Individual representation**   Each individual is represented as a `C` language function linked with CUDD (Colorado University Decision Diagram), a widely used `C` library for constructing and manipulating decision diagrams. This allows generated heuristics to operate directly on the BDD manager, avoiding the need to implement low level BDD operations from scratch. The relevant CUDD API calls are listed in Appendix D.1.

**Initial heuristics construction**   The initial population is seeded using three established BDD variable ordering methods: sifting (Rudell, 1993), GA (Drechsler et al., 1996), and SA (Bollig et al., 1995). Whereas ReEvo initializes its search from a single seed heuristic, `QuantumEvo` uses seed heuristics from these three distinct algorithmic families. Each method represents a different search strategy, allowing the LLM-driven evolutionary process to recombine and adapt their algorithmic components when generating new heuristics.

**Fitness Function**   `QuantumEvo` defines the fitness function using the downstream QCC. Each valid individual is compiled, executed on the search set, and evaluated after full reversible circuit synthesis. Let $\mathcal{T}$ denote the search set used during evolution. For each instance $f \in \mathcal{T}$, let $q_f^{\text{base}}$ denote the lowest QCC

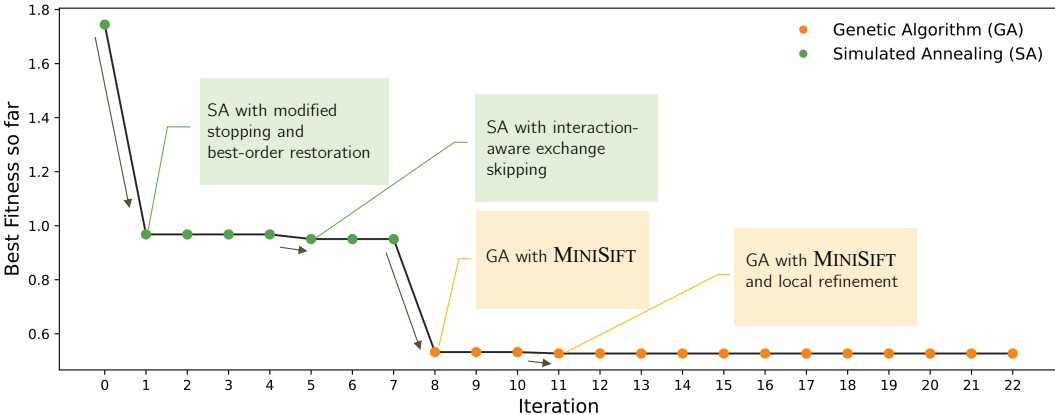

Figure 3: Fitness of the best individual over iterations, showing the family transition at iteration 8. Lower values indicate better fitness.

achieved on $f$ among the classical baseline methods considered in this work. The fitness of a generated heuristic $h$ is defined as the average relative QCC gap to the instance-wise best baseline:

$$\text{fitness}(h) = \frac{1}{|\mathcal{T}|} \sum_{f \in \mathcal{T}} \frac{q(C_{h,f}) - q_f^{\text{base}}}{q_f^{\text{base}}}. \tag{2}$$

Lower values are better, and negative values indicate improvement over the best baseline for that instance. Because generated heuristics may contain stochastic components, each valid individual is evaluated multiple times per instance.

### 3.3 Analysis of the Discovered Algorithm

We analyze the best heuristic selected by `QuantumEvo`, tracing its evolution across iterations. Figure 3 shows the fitness trajectory of the overall best individual. In the best run, the best-so-far individual starts from an SA-family heuristic, transitions to a GA-family heuristic at iteration 8, and later incorporates a local refinement step. We refer to the best heuristic as HGA-QE, a hybrid GA discovered by `QuantumEvo` using a targeted sifting procedure. HGA-QE uses two generated subroutines, PARTIALSIFT and MINISIFT. The algorithm is summarized in Algorithm 1, where the highlighted phase indicates the modification from the original GA. The subroutines are detailed in Appendix D.1. PARTIALSIFT reorders the current BDD according to a candidate variable order, identifies the affected levels, and calls MINISIFT to locally sift only those levels. This replaces the standard sifting step used in the original GA, while keeping the same fitness function based on inverse BDD size.

Table 1 shows that this replacement improves QCC while reducing runtime. The main gain comes from MINISIFT, while the additional local refinement step gives only marginal improvement. We infer that the improvement is related to greater diversity among candidate orderings and the resulting BDD structures. The standard sifting tends to pull different candidates toward similar BDD structures optimized for BDD size, whereas MINISIFT produces more diverse orderings with comparable BDD size. The diagnostic analysis in Appendix D.2 supports this interpretation and further examines cases where the BDD size is unchanged but MINISIFT lowers QCC. In these cases, the resulting BDDs have synthesis-relevant structural differences, accompanied by fewer Toffoli gates and total controls. This suggests that MINISIFT may expose a more diverse set of variable orderings, some of which are more favorable for synthesis and are associated with lower QCC, even though the GA itself still selects candidates by the BDD size.

---

**Algorithm 1 HGA-QE**: Hybrid GA discovered by `QuantumEvo`. Subroutines PARTIALSIFT and MINISIFT are defined in Appendix D.1.

---

**Require:** BDD manager $T$, variable range $[\ell, u]$
**Ensure:** Improved variable ordering in $T$

    **Phase 1 — Baseline sifting**
1: SIFT$(T, \ell, u)$                 ▷ existing CUDD API; sift variables to reduce BDD size

    **Phase 2 — Population initialization**
2: $n \leftarrow u - \ell + 1$;    popsize $\leftarrow \max\{4, \min(2n, 40)\}$;    nCross $\leftarrow \min(3n,$ popsize, $60)$
3: $P[0] \leftarrow$ current ordering of $T$ after sift;    $P[0].\text{cost} \leftarrow |T.\text{nodes}|$
4: $P[1] \leftarrow$ reverse of $P[0]$
5: **for** $i = 2$ **to** popsize $- 1$ **do**
6:      $P[i] \leftarrow$ uniformly random permutation of variables in $[\ell, u]$
7: **end for**
8: **for** $i = 1$ **to** popsize $- 1$ **do**
9:      $(P[i], P[i].\text{cost}) \leftarrow$ PARTIALSIFT$(T, P[i], \ell, u)$                    ▷ Algorithm 2
10: **end for**

    **Phase 3 — Evolutionary loop**
11: **for** $m = 1$ **to** nCross **do**
12:      $(p_1, p_2) \leftarrow$ ROULETTESELECT$(P)$ such that $p_1 \neq p_2$      ▷ roulette selection weighted by $1/\text{cost}$
13:      $(c_1, c_2) \leftarrow$ PMX$(P[p_1], P[p_2])$           ▷ PMX crossover for valid permutations
14:      **for** each offspring $c \in \{c_1, c_2\}$ **do**
15:          $(c, c.\text{cost}) \leftarrow$ PARTIALSIFT$(T, c, \ell, u)$
16:          $w \leftarrow$ worst replaceable individual in $P$    ▷ do not discard the only tracked copy of a repeated order
17:          **if** $c.\text{cost} < P[w].\text{cost}$ **then**
18:              $P[w] \leftarrow c$                     ▷ replace worst with offspring
19:          **end if**
20:      **end for**
21: **end for**
22: $b \leftarrow \arg\min_i P[i].\text{cost}$
23: $(P[b], P[b].\text{cost}) \leftarrow$ PARTIALSIFT$(T, P[b], \ell, u)$           ▷ restore $T$ to the best ordering
24: $O^\star \leftarrow$ current ordering of $T$;    $C^\star \leftarrow P[b].\text{cost}$

    **Phase 4 — Local refinement**
25: **if** $n \geq 2$ **then**
26:      Choose $x \sim \text{Unif}\{\ell, \ldots, u - 1\}$
27:      SWAPADJACENT$(T, x, x + 1)$       ▷ existing CUDD API; exchange two adjacent BDD levels
28:      MINISIFT$(T, \ell, u, \{0, \ldots, n - 1\})$
29:      newCost $\leftarrow |T.\text{nodes}|$
30:      **if** newCost $< C^\star$ **then**
31:          $O^\star \leftarrow$ current ordering of $T$
32:          $C^\star \leftarrow$ newCost
33:      **else**
34:          RESTOREORDER$(T, O^\star, \ell, u)$               ▷ revert to the saved best order
35:      **end if**
36: **end if**

---

# 4 Experiment

We compare the best heuristic discovered by `QuantumEvo` against baselines, and conduct additional studies on the design of framework. All experiments are conducted using a single thread on an AMD Ryzen 9 5900X machine with 64 GB of RAM.

Table 1: Comparison of GA variants on the search set using best-of-3 seeds. The mean gaps are signed percentage differences relative to the best known value per instance. The win/tie/loss counts are relative to GA with sifting.

| | QCC | | BDD | | |
|---|---|---|---|---|---|
| Method | Mean Gap ($\downarrow$) | W/T/L | Mean Gap ($\downarrow$) | W/T/L | Total Time (s) |
| GA with sifting | 2.096% | - | $-0.211\%$ | - | 23.24 |
| GA with MINISIFT | 0.532% | 34/53/13 | $-0.350\%$ | 4/94/2 | 21.95 |
| GA with MINISIFT + local refinement | 0.527% | 34/53/13 | $-0.354\%$ | 4/94/2 | 22.01 |

*Note.* The initial GA heuristic, corresponding to GA with sifting, uses a reduced population for runtime efficiency. The negative mean BDD gap indicates smaller BDDs than the best baseline value.

## 4.1 Experimental Setup

This subsection describes the datasets, baselines, and LLM configuration used in our experiments. The hyperparameters are summarized in Table 5.

**Dataset** We use benchmark functions from RevLib, LGSynth91, and ISCAS85/89. RevLib (Wille et al., 2008) provides benchmarks for reversible and quantum circuit design,[2] while LGSynth91 and ISCAS85/89 are taken from the `hdl-benchmarks` repository.[3] We augment RevLib functions using input negation, output complementation, and don't-care expansion. The resulting instances are split into a search set for evaluating generated heuristics during evolution and a validation set for selecting the best heuristic across independent runs. Final evaluation uses a benchmark set of 148 reversible functions not used during search or validation, consisting of 96 RevLib, 48 LGSynth91, and 4 ISCAS85/89 circuits. A detailed comparison between the search/validation set and the benchmark set is provided in Appendix B.3.

**Baseline Algorithms** We compare against five classical BDD variable ordering methods implemented in CUDD: sifting, symmetric sifting, group sifting, GA and SA (details in Appendix B.1). We also compare against a learning-based method, BDD2Seq (Miao et al., 2026), using the QCC reported in the original paper; our benchmark set is constructed to match the functions evaluated therein.

**LLM configuration** The best selected heuristic is obtained from an evolutionary run using the local LLM `gpt-oss-120b`,[4] an open-source 120B-parameter mixture-of-experts model released by OpenAI under the Apache 2.0 license. Appendix C.2 compares independent runs with two model settings: the local model `gpt-oss-120b` and the commercial model `gpt-5.4-mini`. [5]

## 4.2 Experimental Results

We organize the experimental results around two main findings, followed by an ablation study of `QuantumEvo`. Finding 1 examines whether BDD size is a reliable proxy for downstream QCC, and Finding 2 evaluates the discovered heuristic on the benchmark. We then present an ablation study of `QuantumEvo` to analyze the impact of key search design choices.

### 4.2.1 Finding 1: BDD size is an imperfect proxy for downstream QCC.

The BDD variable-ordering algorithms commonly aim to minimize BDD size. Since our objective is to minimize downstream QCC, we first examine whether BDD size serves as a reliable proxy for QCC.

---

[2]https://www.revlib.org/index.php
[3]https://github.com/ispras/hdl-benchmarks
[4]https://huggingface.co/openai/gpt-oss-120b
[5]https://platform.openai.com/docs/models/gpt-5.4-mini

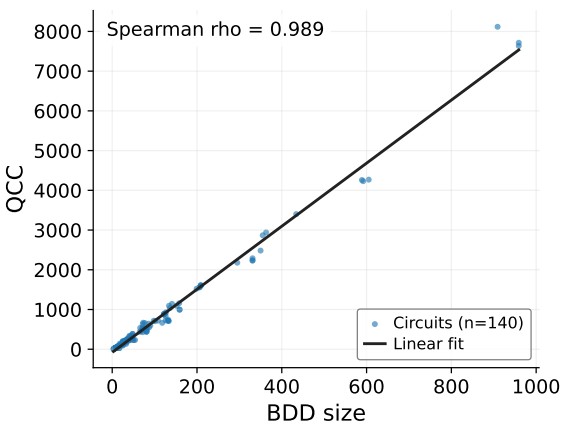 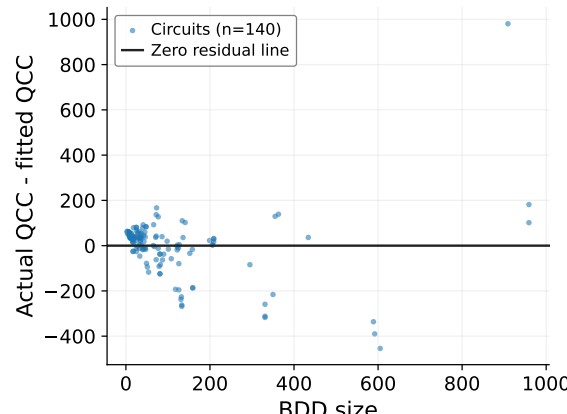

(a) Overall trend with Spearman $\rho = 0.989$ and linear fit

(b) Residuals from the linear fit, showing function-level deviations

Figure 4: BDD size vs. QCC for the 140 search and validation functions with BDD size at most 1000.

Figure 4 examines the relationship between BDD size and QCC on the 140 search/validation functions with BDD size at most 1000. The residual plot shows the difference between the actual and fitted QCC, with the horizontal line indicating zero residual. QCC is strongly correlated with BDD size, as indicated by Spearman $\rho = 0.989$, but residuals around the fitted trend are substantial. The mean absolute deviation is 79.3 up to 980. Moreover, 35% of these functions exhibit at least one ordering pair in which a smaller BDD yields a higher QCC, suggesting that BDD size is not a reliable proxy. These observations motivate the QCC-aware heuristic design used in `QuantumEvo`.

### 4.2.2 Finding 2: HGA-QE is competitive with existing variable ordering baselines.

We select the heuristic with the best validation fitness across independent runs, corresponding to the one analyzed in Section 3.3. Table 2 reports results by suite, where Best counts wins or ties and Strict counts unique wins. For stochastic methods, QCC statistics are computed over five seeds, and for BDD2Seq we use the better of BDD2Seq(B*) and BDD2Seq(E*) reported in the original paper. Overall, HGA-QE matches or improves upon the best baseline on 105 of 148 functions and is uniquely best on 20 functions. On RevLib, it achieves the highest Best count, although its mean QCC is comparable to SA and BDD2Seq has more strict wins. Its gains are clearer on LGSynth91 and ISCAS85/89, suggesting transfer beyond the augmented RevLib functions used for search and validation. The performance profile in Figure 5a shows that HGA-QE remains competitive across the benchmark set with respect to QCC. The quality–runtime Pareto curve in Figure 5b, aggregated using the geometric mean, summarizes the overall trade-off without being dominated by a few expensive instances. After two runs, HGA-QE achieves a trade-off comparable to BDD2Seq, while additional runs further improve QCC at the cost of increased cumulative runtime. While the overall trade-off is favorable, reducing runtime on some expensive instances remains an important direction for further optimization. Table 3 details the 20 functions on which our method achieves a lower QCC than all the baselines. The average time denotes the mean runtime per seed, whereas the total time denotes the cumulative runtime across all seeds. Both measurements include BDD construction, variable ordering, and reversible circuit synthesis. Because these steps are performed once for a given Boolean function and the resulting circuit can be reused for subsequent executions, we prioritize circuit quality as long as the synthesis runtime remains practically manageable.

Table 2: Per-suite benchmark performance. Best includes ties for the lowest Best QCC; Strict is counted only when the method is the unique best (5 random seeds for stochastic methods).

| Suite | Method | Best | Strict | Mean Best QCC | Mean Avg QCC | Mean Worst QCC |
|---|---|---|---|---|---|---|
| RevLib (96) | HGA-QE | **70** | 6 | **1416.04** | **1434.31** | **1450.76** |
| | SA | 68 | 5 | 1416.25 | 1437.27 | 1463.44 |
| | BDD2Seq | 65 | **13** | 1456.03 | – | – |
| | GA | 58 | 1 | 1445.28 | 1457.55 | 1468.35 |
| | SIFT | 42 | 0 | 1767.24 | – | – |
| | SYMM SIFT | 41 | 0 | 1757.90 | – | – |
| | GROUP SIFT | 41 | 1 | 1688.56 | – | – |
| LGSynth91 (48) | HGA-QE | **31** | **13** | **2088.54** | **2218.05** | **2344.25** |
| | SA | 22 | 3 | 2153.58 | 2267.06 | 2447.79 |
| | BDD2Seq | 22 | 5 | 2466.00 | – | – |
| | GA | 23 | 6 | 125739.94 | 202778.95 | 293303.17 |
| | SIFT | 16 | 1 | 4687.90 | – | – |
| | SYMM SIFT | 16 | 0 | 4682.71 | – | – |
| | GROUP SIFT | 15 | 0 | 5762.81 | – | – |
| ISCAS85/89 (4) | HGA-QE | **4** | **1** | **12134.25** | **12211.23** | **12288.75** |
| | SA | 3 | 0 | 12286.00 | 12403.25 | 12589.00 |
| | BDD2Seq | 1 | 0 | 13549.50 | – | – |
| | GA | 3 | 0 | 12296.00 | 12518.60 | 12681.25 |
| | SIFT | 0 | 0 | 19230.00 | – | – |
| | SYMM SIFT | 0 | 0 | 19230.00 | – | – |
| | GROUP SIFT | 1 | 0 | 23486.00 | – | – |

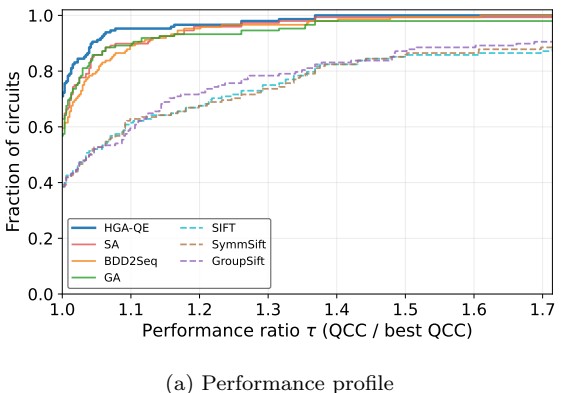

(a) Performance profile

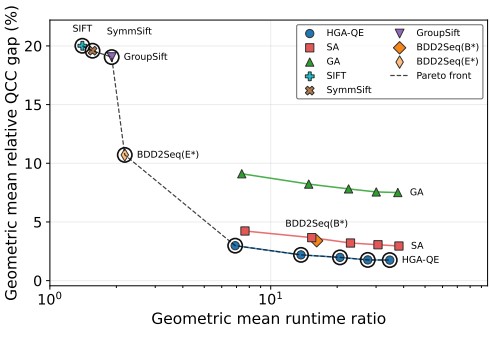

(b) Quality–runtime Pareto curve

Figure 5: (a) Performance profile over the whole benchmark set, using the ratio between each method's QCC and the best QCC achieved on the same function. (b) Quality–runtime Pareto curve over all 148 benchmark circuits. The $x$-axis reports the geometric mean runtime ratio relative to the per-circuit fastest result, and the $y$-axis reports the geometric mean relative QCC gap. Connected points correspond to cumulative run budgets for stochastic methods, whereas deterministic methods and BDD2Seq variants are shown as single points. Lower-left is better.

### 4.2.3 Ablation Study of `QuantumEvo`

We study three components of `QuantumEvo`: LLM guidance, multi-family initialization, and QCC-based fitness. For each setting, we conduct ten independent searches and compare their validation performance in Figure 6. We then select the individual with the best validation fitness from each setting and evaluate it on the full 148-function benchmark set. The corresponding benchmark results are summarized in Table 4.

Table 3: Compact comparison for functions where `QuantumEvo` is strictly best.

| Circuit | I | O | QuantumEvo Best | Avg | Avg Time | GA Best | Avg | Avg Time | SA Best | Avg | Avg Time |
|---|---|---|---|---|---|---|---|---|---|---|---|
| apex5_104[R] | 117 | 88 | **9184** | 9812.40 | 0.97 | 9551 | 10041.40 | 1.24 | 9397 | **9809.00** | 1.97 |
| apex6_orig[L] | 135 | 99 | **3736** | **3787.20** | 0.65 | 4159 | 4368.40 | 0.64 | 4392 | 4551.40 | 1.08 |
| apex7_orig[L] | 49 | 37 | **1431** | **1440.60** | 0.28 | 1441 | 1443.40 | 0.58 | 1703 | 1759.20 | 0.44 |
| b9_orig[L] | 41 | 21 | **713** | 719.40 | 0.17 | 715 | **719.00** | 0.27 | 804 | 804.00 | 0.19 |
| c432[I] | 36 | 7 | **10494** | **11053.20** | 245.19 | 11141 | 11170.60 | 6.20 | 11101 | 11148.60 | 1.46 |
| comp_orig[L] | 32 | 3 | **796** | **796.00** | 2.41 | 824 | 824.00 | 2.28 | 897 | 936.00 | 2.01 |
| cordic_138[R] | 23 | 2 | **304** | **308.00** | 0.16 | 306 | 312.00 | 0.23 | 316 | 323.20 | 0.17 |
| cordic_orig[L] | 23 | 2 | **304** | **308.00** | 0.12 | 306 | 312.00 | 0.20 | 316 | 323.20 | 0.14 |
| cu_orig[L] | 14 | 11 | **219** | **219.80** | 0.08 | 224 | 224.00 | 0.09 | 220 | 220.00 | 0.09 |
| dalu_orig[L] | 75 | 16 | **5230** | 5301.60 | 71.73 | 5240 | **5291.20** | 71.72 | 5316 | 5663.20 | 69.72 |
| frg1_160[R] | 28 | 3 | **559** | **597.40** | 0.17 | 560 | 610.20 | 0.24 | 567 | 637.40 | 0.17 |
| frg1_orig[L] | 28 | 3 | **559** | **597.40** | 0.15 | 560 | 610.20 | 0.27 | 567 | 637.40 | 0.17 |
| k2_orig[L] | 45 | 45 | **10156** | **10212.60** | 0.97 | 10200 | 10265.40 | 1.78 | 10276 | 10410.00 | 1.49 |
| misex1_178[R] | 8 | 7 | **279** | 290.40 | 0.06 | 287 | **287.00** | 0.06 | 288 | 288.00 | 0.07 |
| pair_orig[L] | 173 | 137 | **20236** | **22344.20** | 6.24 | 47110 | 602976.40 | 8.28 | 20729 | 22438.00 | 20.16 |
| rot_orig[L] | 135 | 107 | **23926** | 26838.00 | 845.19 | 5928978 | 9069721.00 | 62.87 | 24124 | **25464.60** | 25.99 |
| seq_201[R] | 41 | 35 | **9080** | 9151.00 | 1.17 | 9113 | **9142.00** | 3.03 | 9165 | 9237.40 | 1.99 |
| urf4_89[R] | 11 | 11 | **28406** | **28537.40** | 0.41 | 28538 | 28591.80 | 0.54 | 28489 | 28590.80 | 0.97 |
| x3_orig[L] | 135 | 99 | **3736** | **3787.20** | 0.62 | 4159 | 4368.40 | 0.66 | 4392 | 4551.40 | 1.15 |
| x4_orig[L] | 94 | 71 | **2437** | **2513.60** | 0.37 | 2505 | 2809.40 | 0.65 | 2738 | 2863.20 | 0.69 |

| Circuit | QuantumEvo Best | Total Time | BDD2Seq(B*) Best | Time | BDD2Seq(E*) Best | Time | SIFT Best | Time | SYMM_SIFT Best | Time | GROUP_SIFT Best | Time |
|---|---|---|---|---|---|---|---|---|---|---|---|---|
| apex5_104[R] | **9184** | 4.87 | 9852 | 12.83 | 10697 | 0.20 | 10358 | 0.06 | 10228 | 0.03 | 10227 | 0.06 |
| apex6_orig[L] | **3736** | 3.24 | 3895 | 16.72 | 4021 | 0.16 | 4822 | 0.05 | 4810 | 0.03 | 4269 | 0.03 |
| apex7_orig[L] | **1431** | 1.40 | 1455 | 2.40 | 1456 | 0.07 | 2653 | 0.04 | 2653 | 0.04 | 2377 | 0.04 |
| b9_orig[L] | **713** | 0.83 | 737 | 1.71 | 737 | 0.05 | 804 | 0.03 | 810 | 0.03 | 836 | 0.03 |
| c432[I] | **10494** | 1225.95 | 12255 | 10.01 | 20391 | 6.79 | 11101 | 0.04 | 11101 | 0.02 | 11141 | 0.04 |
| comp_orig[L] | **796** | 12.04 | 824 | 40.21 | 824 | 28.75 | 961 | 1.77 | 1071 | 1.96 | 884 | 1.79 |
| cordic_138[R] | **304** | 0.82 | 344 | 0.62 | 446 | 0.05 | 325 | 0.02 | 323 | 0.02 | 318 | 0.04 |
| cordic_orig[L] | **304** | 0.58 | 320 | 0.62 | 320 | 0.06 | 325 | 0.01 | 323 | 0.01 | 318 | 0.03 |
| cu_orig[L] | **219** | 0.39 | 224 | 0.26 | 231 | 0.02 | 220 | 0.01 | 220 | 0.01 | 224 | 0.03 |
| dalu_orig[L] | **5230** | 358.67 | 5297 | 55.53 | 5652 | 28.17 | 31145 | 66.13 | 31145 | 71.95 | 47170 | 75.07 |
| frg1_160[R] | **559** | 0.85 | 598 | 0.95 | 629 | 0.03 | 747 | 0.03 | 747 | 0.03 | 827 | 0.03 |
| frg1_orig[L] | **559** | 0.77 | 757 | 0.95 | 762 | 0.03 | 747 | 0.03 | 747 | 0.02 | 827 | 0.02 |
| k2_orig[L] | **10156** | 4.85 | 10432 | 2.38 | 10686 | 0.19 | 11058 | 0.07 | 11058 | 0.13 | 11275 | 0.09 |
| misex1_178[R] | **279** | 0.30 | 281 | 0.12 | 289 | 0.02 | 288 | 0.01 | 288 | 0.03 | 289 | 0.03 |
| pair_orig[L] | **20236** | 31.21 | 20754 | 28.19 | 20799 | 1.06 | 46491 | 0.11 | 46391 | 0.17 | 49917 | 0.15 |
| rot_orig[L] | **23926** | 4225.96 | 38453 | 37.98 | 48344 | 15.69 | 78639 | 0.56 | 78374 | 0.62 | 110936 | 0.74 |
| seq_201[R] | **9080** | 5.85 | 9349 | 2.12 | 9908 | 0.33 | 19362 | 0.24 | 19350 | 0.28 | 15309 | 0.27 |
| urf4_89[R] | **28406** | 2.06 | 28488 | 0.68 | 28833 | 0.47 | 28523 | 0.06 | 28523 | 0.10 | 28523 | 0.11 |
| x3_orig[L] | **3736** | 3.09 | 3941 | 16.78 | 4645 | 0.19 | 4822 | 0.03 | 4810 | 0.05 | 4269 | 0.03 |
| x4_orig[L] | **2437** | 1.85 | 2662 | 8.31 | 2782 | 0.11 | 4470 | 0.02 | 4470 | 0.04 | 4334 | 0.04 |

*Superscripts denote source dataset.* [I] ISCAS85/89; [L] LGSynth91; [R] RevLib.
*Hardware normalization.* BDD2Seq(B*) runtime is normalized from the original dual Xeon 8375C platform to our single-thread Ryzen 9 5900X platform. Because BDD2Seq(B*) uses beam search with a beam width of 20 and synthesizes all 20 candidate orderings, we normalize its additional CPU-based synthesis cost using the PassMark single-thread ratio between the two platforms. BDD2Seq(E*) is left unchanged because its single synthesis call constitutes only a small fraction of its total runtime. The normalization assumptions and derivation are provided in Appendix G.

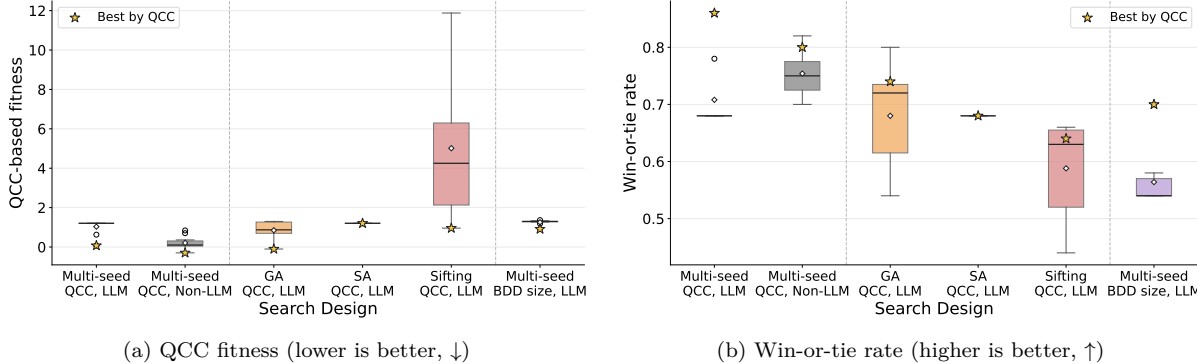

(a) QCC fitness (lower is better, ↓)   (b) Win-or-tie rate (higher is better, ↑)

Figure 6: Validation-set distributions over ten independent runs per setting. The star marks the run selected by the lowest validation QCC fitness; the corresponding win-or-tie rate is reported for the same run.

Table 4: Benchmark evaluation of the best-validation individual from each setting on the full benchmark set with five random seeds. Starting from the full `QuantumEvo` setting (bold), each block varies one factor: Guidance, Initialization, or Fitness.

| Initialization | Fitness | Guidance | Best/Total (%) | Strictly Best/Total (%) | Mean Best QCC | Mean Avg QCC | Mean Worst QCC |
|---|---|---|---|---|---|---|---|
| Sifting, GA, SA | QCC | LLM | **70.9** | **13.5** | **1923.83** | **1979.76** | **2033.46** |
|  |  | Non-LLM | 66.9 | 5.4 | 2075.15 | 2220.27 | 2432.07 |
| GA | QCC | LLM | 63.5 | 3.4 | 2061.84 | 2172.50 | 2264.26 |
| SA | QCC | LLM | 66.2 | 0.7 | 1949.11 | 2004.30 | 2084.68 |
| Sifting | QCC | LLM | 55.4 | 4.1 | 2203.34 | 2435.04 | 2670.19 |
| Sifting, GA, SA | BDD size | LLM | 61.5 | 5.4 | 1944.65 | 2003.03 | 2099.68 |

**LLM guidance.** To isolate the contribution of the LLM, we construct a non-LLM heuristic generation that retains the same evolutionary procedure, fitness function, initialization, and search and validation sets as `QuantumEvo`. Instead of generating and revising code with an LLM, it assembles candidates from predefined components of sifting, GA, and SA, as detailed in Appendix C.1. As shown in Figure 6, the non-LLM search achieves stronger validation fitness, and all ten runs converge to GA-family individuals. These results do not establish that LLM guidance is uniformly superior, but suggest that LLM-driven algorithm design beyond a predefined component space can yield heuristics that transfer more effectively to unseen functions.

**Multi-family initialization.** To assess seed diversity, we compare the multi-family initialization of sifting, GA, and SA against initialization from each family alone. SA-only initialization has zero variance, with every run converging to the same individual despite LLM stochasticity, indicating that a single seed family can constrain the explored region. GA-only initialization achieves the better validation QCC fitness than multi-family initialization, whereas multi-family initialization attains a higher validation win-or-tie rate. On the full benchmark, however, the heuristic selected under GA-only initialization no longer leads, and the heuristics selected under SA-only and sifting-only initialization also underperform the HGA-QE.

**QCC-based fitness.** Finally, we replace the downstream QCC objective with BDD size while retaining the remaining components of `QuantumEvo`. The BDD-size setting shows a less favorable median validation QCC fitness in Figure 6. Its selected heuristic also underperforms the corresponding QCC-based setting on the benchmark, with lower Best and Strictly Best rates and higher mean QCC.

# 5 Discussion

While the results demonstrate that LLM-driven evolutionary search can discover competitive BDD variable ordering heuristics, several limitations remain. First, the discovered heuristic is still influenced by the initial heuristic families used to seed the search. Although the best lineage transitions from SA-family to GA-family individuals, the best heuristic combines and refines ideas related to sifting, SA, and GA rather than introducing an entirely new class of ordering strategy. Whether alternative framework designs can discover substantially different and more effective ordering strategies remains an open question. Second, `QuantumEvo` optimizes an important component of BDD-based reversible synthesis, namely the variable ordering. While this choice affects the BDD structure and downstream QCC, other resource measures, such as ancilla count, also depend on the synthesis procedure itself. Future work could combine `QuantumEvo` with post-synthesis ancilla reduction techniques or alternative synthesis methods that produce lower-cost circuits. Third, the non-LLM search remains competitive while avoiding the token and inference costs of LLM guidance. This suggests that hybrid strategies could reduce offline discovery cost by relying primarily on predefined evolutionary operators and invoking the LLM selectively when additional algorithmic variation is needed.

# 6 Conclusion

We presented `QuantumEvo`, an LLM-driven evolutionary framework that discovers interpretable and reusable BDD variable ordering heuristics for reversible circuit synthesis. The best discovered heuristic, HGA-QE, achieves competitive QCC across diverse benchmark circuits and improves over classical CUDD ordering methods and learning-based baselines on a substantial subset of functions. Its implementation as a general CUDD variable-ordering heuristic also enables its reuse in other BDD-based workflows. More broadly, `QuantumEvo` illustrates how LLM-based heuristic design can be adapted to domain-specific settings by evaluating candidates directly on the relevant downstream objective. This approach may extend to other optimization decisions embedded within specialized computational workflows.

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

# A    Example of Reversible Circuit Synthesis

## A.1    NCT library and Circuit

The NCT library consists of three gate types: NOT, CNOT, and Toffoli. The NOT gate flips a single bit $(x \mapsto \bar{x})$. The CNOT gate flips the target bit when the control bit is 1, $(x, t) \mapsto (x, t \oplus x)$, and the Toffoli gate flips the target bit when both control bits are 1, $(x_1, x_2, t) \mapsto (x_1, x_2, t \oplus (x_1 \wedge x_2))$. Figure 7 illustrates this with a 3-bit reversible function. The truth table lists all $2^3 = 8$ input combinations $(x_1, x_2, x_3)$ and their outputs $(y_1, y_2, y_3)$; each output row appears exactly once, confirming bijectivity. The right panel shows the NCT circuit synthesized from this function, with a total QCC of 8 under the cost model of Szyprowski & Kerntopf (2013).

| $x_1$ | $x_2$ | $x_3$ | $y_1$ | $y_2$ | $y_3$ |
|-------|-------|-------|-------|-------|-------|
| 0 | 0 | 0 | 0 | 0 | 1 |
| 0 | 0 | 1 | 0 | 0 | 0 |
| 0 | 1 | 0 | 0 | 1 | 1 |
| 0 | 1 | 1 | 0 | 1 | 0 |
| 1 | 0 | 0 | 1 | 0 | 1 |
| 1 | 0 | 1 | 1 | 1 | 1 |
| 1 | 1 | 0 | 1 | 0 | 0 |
| 1 | 1 | 1 | 1 | 1 | 0 |

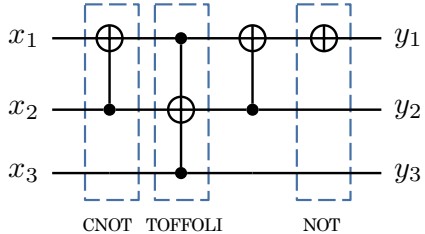

(a) Truth table of a 3-bit reversible function    (b) Synthesized reversible circuit (QCC = 8)

Figure 7: A 3-bit reversible function and its synthesized NCT circuit.

## A.2    BDD Variable Ordering and Circuit

Figure 8 provides another example illustrating the impact of variable ordering on BDD size and QCC. The same function under two orderings yields BDD sizes of 9 and 6, and QCC of 52 and 24, respectively. This shows that variable ordering can affect both BDD size and QCC.

# B    Detailed Experimental Setup

This section provides supplementary details needed to reproduce and interpret the experiments.

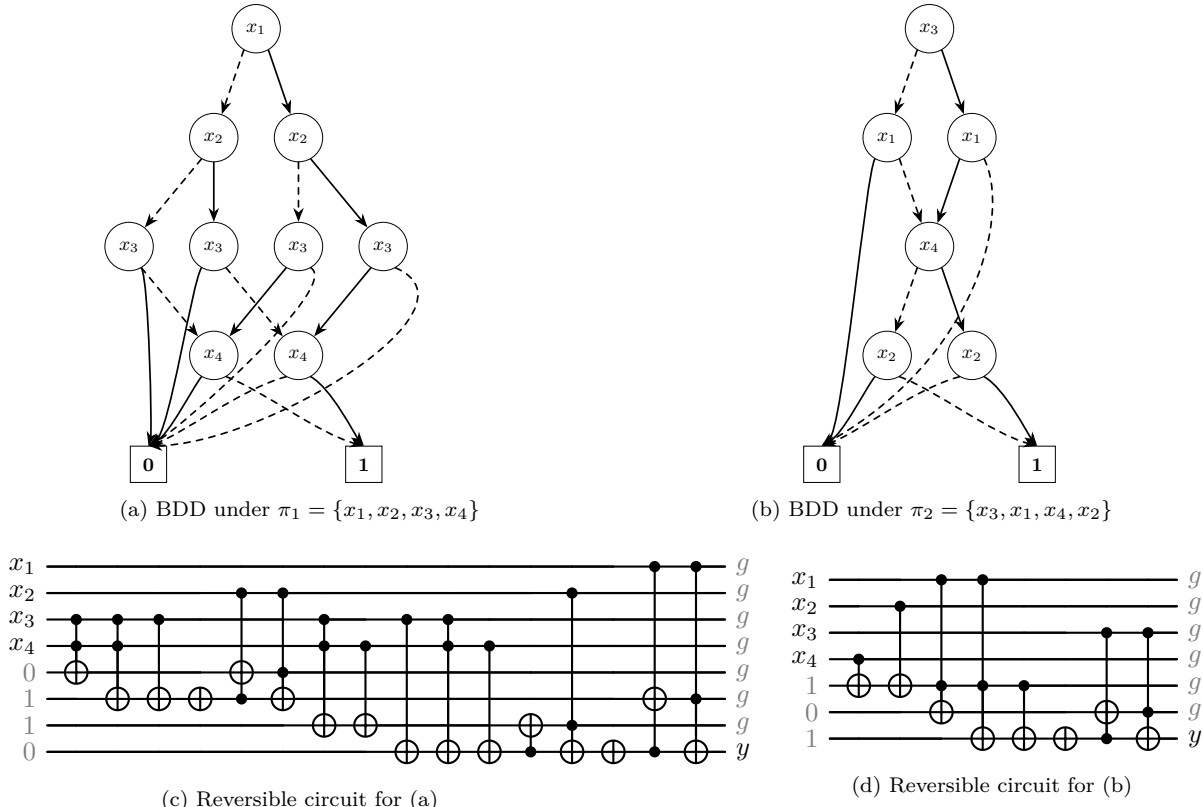

(a) BDD under $\pi_1 = \{x_1, x_2, x_3, x_4\}$

(b) BDD under $\pi_2 = \{x_3, x_1, x_4, x_2\}$

(c) Reversible circuit for (a)

(d) Reversible circuit for (b)

Figure 8: Impact of variable ordering on QCC for the function `4mod5_8`. Solid/dashed edges denote 1/0-branches. The BDD sizes are 9 under $\pi_1$ and 6 under $\pi_2$, whereas the corresponding QCC are 52 and 24. Here, $y = \overline{(x_1 \oplus x_3)} \cdot \overline{(x_2 \oplus x_4)}$

.

## B.1 Baseline Ordering Methods in CUDD

CUDD (Somenzi, 2023) is a `C` library for constructing and manipulating BDDs. The classical baselines implemented in CUDD are:

- **Sifting** (Rudell, 1993): Moves each variable through all possible levels and retains the position minimizing node count.
- **Symmetric Sifting** (Panda et al., 1994): Extends sifting by detecting symmetric variable pairs during traversal and grouping them for joint sifting.
- **Group Sifting** (Panda & Somenzi, 1995): Generalizes symmetric sifting to arbitrary dependent variable groups, sifting each group as a whole.
- **Genetic Algorithm** (Drechsler et al., 1996): Maintains a population of variable orderings; generates new individuals via crossover and sifts each before fitness evaluation, replacing the worst individual if the offspring improves.
- **Simulated Annealing** (Bollig et al., 1995): Randomly perturbs the ordering and accepts or rejects moves according to an annealing schedule.

## B.2 Experimental Setting

**Hyperparameters** Table 5 summarizes the hyperparameters used in all runs.

**Individual Representation** Each individual is a program linked directly to CUDD. Rather than implementing individuals at the Python level, we use functions linked into the BDD process, which avoids

Table 5: Hyperparameters used for all runs.

| Hyperparameter | Value | Hyperparameter | Value |
|---|---|---|---|
| Max. function evaluations | 200 | Per-iteration timeout | 40 s |
| Population size | 10 | Evaluation seeds per instance | 3 |
| Initial population size | 30 | Search instances | 100 |
| Mutation rate | 0.5 | Validation instances | 50 |

per-step interface overhead and provides more direct access to CUDD ordering routines and relevant internal data structures.

**Initial Heuristics**  The initial heuristics are generated from three functions corresponding to sifting, GA, and SA. The sifting and simulated annealing seeds follow the corresponding CUDD implementations, while the GA seed is a lightweight variant of GA in CUDD that only reduces the population size to keep evaluation cost manageable.

### B.3   Analysis of Dataset and Benchmarks

RevLib is aligned with reversible circuit synthesis and provides functions that can be repeatedly evaluated within the limited search budget. To increase the diversity of the search pool, we augment RevLib. Figure 9 compares the 100 search functions with the 148 benchmark functions by input count, output count, and gate count. The search set is drawn from augmented RevLib functions, whereas the benchmark set consists of RevLib, LGSynth91, and ISCAS85/89 functions with broader structural variety. The benchmark points that coincide with the search-set points correspond to RevLib benchmark functions. This overlap is only at the metadata level, and the search and benchmark sets share no Boolean functions. By contrast, the benchmark points that do not coincide with any search-set point correspond to the non-RevLib suites, LGSynth91 and ISCAS85/89. Thus, the non-RevLib suites introduce benchmark functions with input, output, and gate-count configurations not observed in the RevLib-based search set.

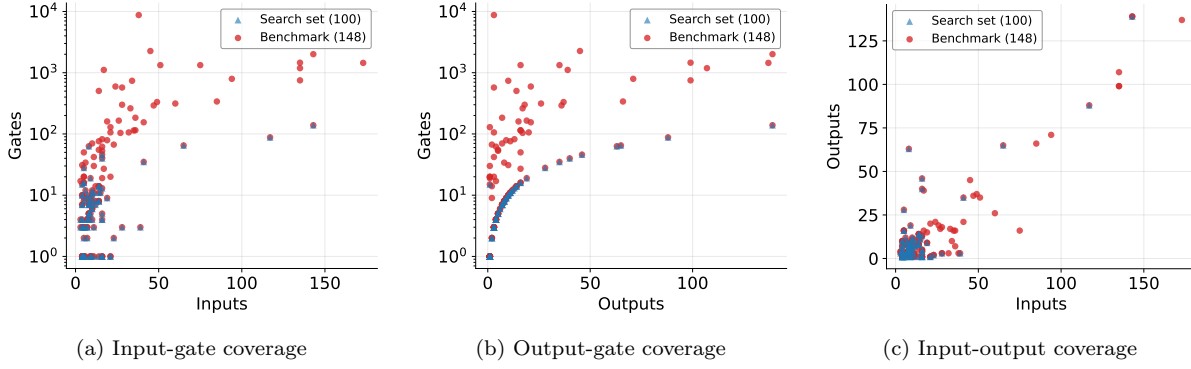

(a) Input-gate coverage          (b) Output-gate coverage          (c) Input-output coverage

Figure 9: Structural comparison of the search set (100 functions) and the benchmark set (148 functions)

## C   Additional Analyses of `QuantumEvo`

### C.1   Details of the Non-LLM Evolutionary Search

We implement the non-LLM control as an evolutionary search that constructs candidates from predefined algorithmic components, following the generation hyper-heuristic paradigm (Burke et al., 2009; 2013; Lin et al., 2020). It uses the same evolutionary procedure, fitness function, initialization scheme, and search and validation sets as `QuantumEvo`, differing only in heuristic generation. The candidates are assembled

by selecting and combining predefined algorithmic components from the search space summarized in Table 6, rather than being generated and iteratively revised by an LLM. Each individual encodes all dimensions, but only those associated with the selected family affect the resulting implementation. The search space consists of three heuristic families, sifting, GA, and SA. These correspond to the three seed heuristics used to initialize `QuantumEvo`. The initial population contains ten individuals from each family, matching the round-robin initialization used by `QuantumEvo`. The non-LLM search can modify and combine a broad range of predefined components, including the sifting procedure embedded within the GA. However, it cannot introduce a new algorithmic idea outside this predefined space, such as MINISIFT's change-aware refinement of only the levels affected by the preceding variation operator.

Table 6: Search dimensions available to each heuristic family in the non-LLM evolutionary search.

| Family | Search dimension | Available choices or range |
|---|---|---|
| Top level | Heuristic family | Sifting / SA / GA. The initial population is divided evenly across the three families. |
| Sifting | Search direction | Bidirectional / top-down only / bottom-up only. |
| | Search range | Full variable range / sliding window with width $w \in \{2, \ldots, 10\}$ / random subset containing $p \in [0.3, 0.9]$ of the variables in each pass. |
| Simulated annealing | Neighborhood operators | Seed exchange–jump mixture / exchange only / jump only / adjacent swap only / all four move types. |
| | Move probabilities | Exchange probability $p_{exc} \in [0.2, 0.6]$; upward-jump share $p_{up} \in [0.2, 0.8]$. |
| | Cooling schedule | Multiplicative cooling factor $\alpha \in [0.80, 0.95]$; secondary temperature-scaling parameter $\beta \in [0.3, 0.8]$. |
| | Restarts | Number of additional re-annealing runs $r \in \{1, 2, 3\}$, each initialized from the best solution found so far. |
| Genetic algorithm | Population initialization | Sifting-based / simulated-annealing-based / random-GA / uniformly random / multi-family initialization. |
| | Population parameters | Population size $n_{pop} \in \{8, \ldots, 40\}$; crossover offspring ratio $\rho_{cross} \in [0.5, 3.0]$; mutation rate $p_{mut} \in [0, 1]$; elitism ratio $\rho_{elite} \in [0, 0.5]$. |
| | Parent selection | Roulette-wheel selection / tournament selection with $k \in \{2, \ldots, 6\}$. |
| | Crossover operator | Partially matched crossover / order crossover. |
| | Perturbation operator | Adjacent swap / random-pair swap / insertion / block reversal. |
| | Offspring improvement | Sifting-based refinement using the dimensions above / adjacent-swap hill climbing. |

## C.2 Study on the Choice of LLM

We compare two LLM choices under the same pipeline, prompt structure, and seed initialization: `gpt-oss-120b` (120B-parameter open-source MoE, Apache 2.0) and `gpt-5.4-mini` (OpenAI API). For each model, we run ten independent searches, select the best individual on the validation set, and evaluate it on the 148 benchmark functions.

Figure 10 shows the distribution of validation results across ten independent runs. Runs using `gpt-5.4-mini` show wider variance and a lower median validation score, and even the best run does not reach the best result obtained with `gpt-oss-120b`. Figure 11 traces the search trajectory of the best `gpt-5.4-mini` run, showing how its best heuristic evolved during the search.

The search runs using `gpt-oss-120b` produce stronger validation results than those using `gpt-5.4-mini`, with a higher win-or-tie rate and lower mean QCC. The two settings also tend to converge to different

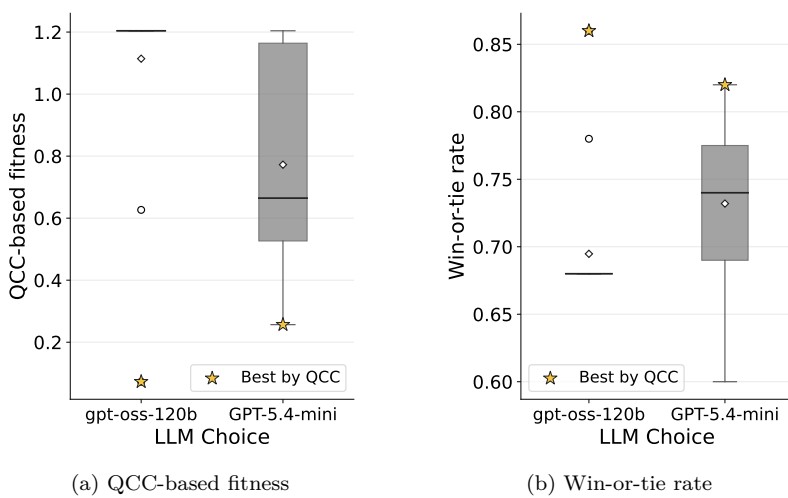

(a) QCC-based fitness       (b) Win-or-tie rate

Figure 10: Validation comparison between the two LLM choices over ten independent runs. The star marks the run selected by the lowest validation QCC fitness.

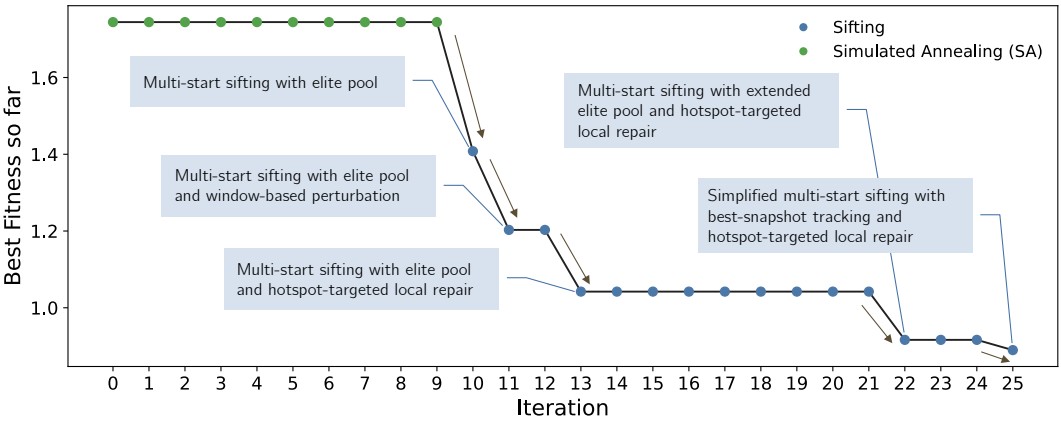

Figure 11: Evolution of the Best Discovered Ordering Heuristic for the choice of `gpt-5.4-mini`

heuristic families. Runs with `gpt-oss-120b` select SA-family individuals in 9 of 10 runs, whereas runs with `gpt-5.4-mini` more often converge to sifting-family individuals. Based on these validation results, we use the best individual generated with `gpt-oss-120b` for the main evaluation.

Table 7: Benchmark comparison of LLM choices. Family counts summarize the best individual selected in each of ten independent runs per model.

| LLM choice | Benchmark | | | | | Family Type | | |
|---|---|---|---|---|---|---|---|---|
| | Best/Total (%) | Strictly Best/Total (%) | Mean Best QCC | Mean Avg QCC | Mean Worst QCC | GA | SA | Sifting |
| `gpt-oss-120b` | **70.9** | **13.5** | **1923.83** | **1979.76** | **2033.46** | 1 | 9 | 0 |
| `gpt-5.4-mini` | 66.9 | 6.1 | 2080.71 | 2147.33 | 2211.11 | 2 | 3 | 5 |

**Family Composition across Iterations**  We classify each valid individual by its code-level family. Figure 12 traces the family composition over iterations for the best run from each LLM. In the `gpt-oss-120b` run, the population shifts from diverse seed families toward SA-family and later GA-family individuals. In contrast, `gpt-5.4-mini` tends to converge toward simpler sifting-style variants.

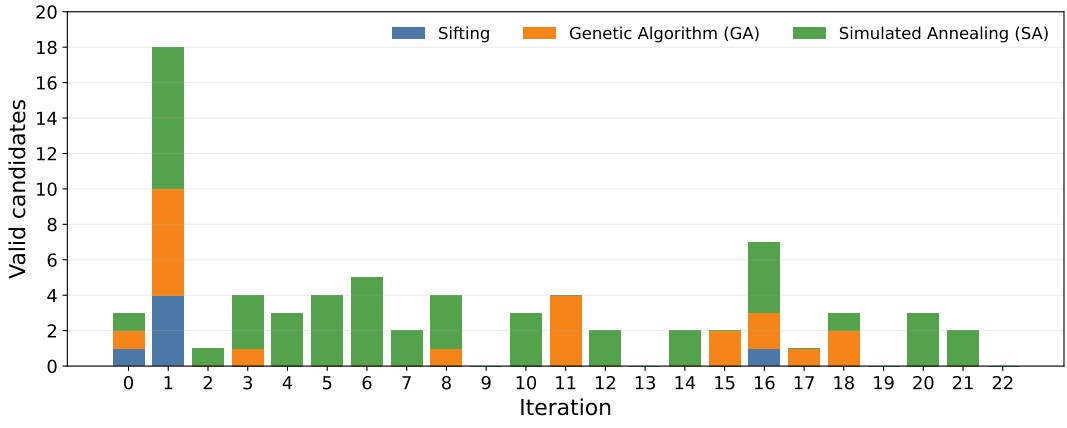

(a) Best run from `gpt-oss-120b` run

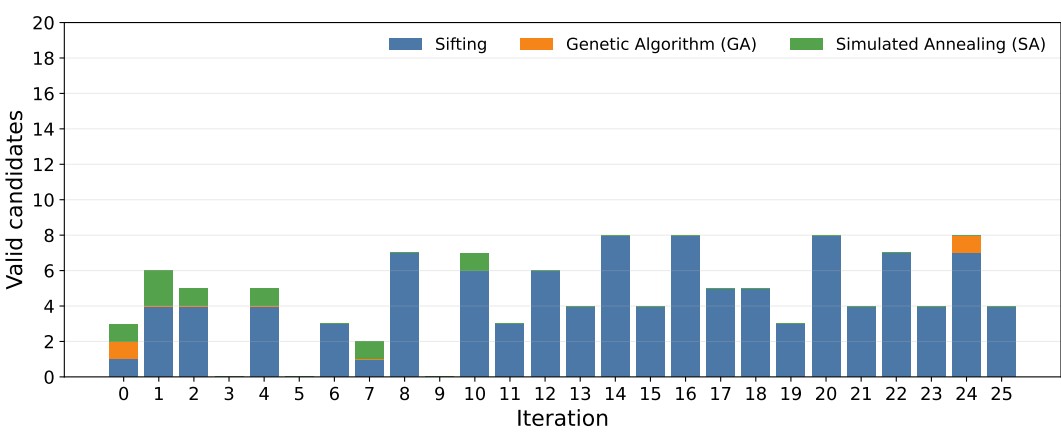

(b) Best run from `gpt-5.4-mini` run

Figure 12: Code-level family composition of valid individuals across iterations

## C.3 Elite-Candidate Comparison across All Settings

To provide a broader comparison across all seven settings considered in Section 4.2.3 and this section, we additionally examine their elite candidate populations rather than only the finalist selected from each run. For each setting, we pool all candidates evaluated across its ten runs, rank them by their logged search-set performance, and evaluate the top 5 and top 10 candidates on the validation set. The full `QuantumEvo` setting, which uses LLM guidance, multi-family initialization, and QCC-based fitness, achieves the best median validation fitness for both candidate pools. For the top 10 pool, however, its distribution widens substantially, whereas the non-LLM control remains tightly concentrated. Given the substantial overlap and the trade-off between median performance and variability, these results do not establish a clear overall advantage for either setting.

## C.4 Sensitivity Analysis

We provide a post-hoc analysis of the search set, validation set, and evaluation timeout. The analysis reuses candidates and runtime logs from the completed searches and characterizes sensitivity within the observed search outcomes.

**Search set.** We reevaluate the 30 leading candidates from the search run that produced the HGA-QE algorithm, across the original and nine alternate 100 function search sets, as reported in Table 8. HGA-QE

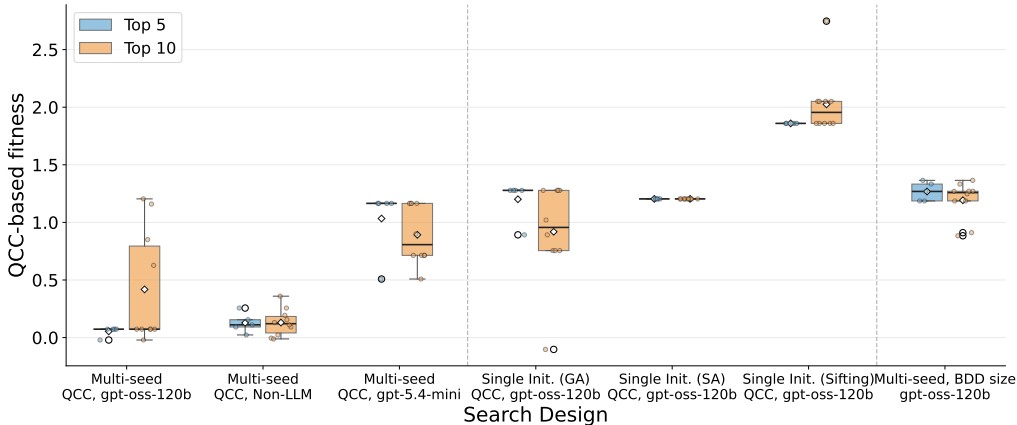

Figure 13: Validation-set fitness of the top 5 and top 10 candidates ranked by search set pooled across 10 runs for each setting. Dashed lines separate the guidance, initialization, and fitness comparisons.

ranks first in 5 of the 10 alternative search set, although the top-ranked candidate varies across sets. These results characterize sensitivity within the candidate pool observed in the completed run.

**Validation set.** For each setting, the original validation set selects one finalist from 10 independent runs. We repeat this selection using nine alternate 50 function sets and report the same-run rate in Table 8. The full `QuantumEvo` setting and the GA-only initialization are comparatively stable, whereas the other settings are more sensitive to the validation-set choice. Overall, the HGA-QE finalist selected under the full `QuantumEvo` setting is more robust to validation-set variation than those selected under the alternative settings.

Table 8: Sensitivity over the original and nine alternate sets. Rates denote top ranking of HGA-QE among 30 candidates or reselection of the original validation finalist.

| Analysis | Initialization | Fitness | Guidance | Rate |
|---|---|---|---|---|
| Search set | Multi-seed | QCC | `gpt-oss-120b` | 5/10 |
| | Multi-seed | QCC | `gpt-oss-120b`
`gpt-5.4-mini`
Non-LLM | 9/10
2/10
1/10 |
| Validation set | GA only
SA only
Sifting only | QCC
QCC
QCC | `gpt-oss-120b`
`gpt-oss-120b`
`gpt-oss-120b` | 8/10
Identical heuristic
4/10 |
| | Multi-seed | BDD size | `gpt-oss-120b` | 5/10 |

**Timeout.** We replay the recorded wall-clock times of candidate evaluations from the 10 production runs under shorter hypothetical timeout limits. An evaluation is counted as a failure when its recorded time exceeds the limit. The evaluations that already timed out at 40 seconds remain failures. The failure rate increases from 8.3% at 40 seconds to 9.8% at 32 seconds, then more rapidly to 17.7% at 20 seconds in Table 9.

Table 9: Candidate-evaluation failure rates under shorter timeout limits, based on the recorded wall times of evaluations from the 10 production runs.

| Budget fraction | Budget (s) | Failure rate |
|---|---|---|
| 1.0× | 40.0 | 8.3% |
| 0.9× | 36.0 | 9.0% |
| 0.8× | 32.0 | 9.8% |
| 0.7× | 28.0 | 11.6% |
| 0.6× | 24.0 | 14.1% |
| 0.5× | 20.0 | 17.7% |

# D    Additional Analysis of HGA-QE

## D.1    Mapping to CUDD Implementations

Algorithm 2 and Algorithm 3 define the subroutines called by Algorithm 1. Table 10 maps each pseudocode function to its corresponding CUDD function and source file.

---

**Algorithm 2** PARTIALSIFT: Individual-order evaluator with MINISIFT (subroutine of Algorithm 1)

---

**Require:** BDD manager $T$, individual permutation $\sigma$, range $[\ell, u]$
**Ensure:** $\sigma$ is replaced by the post-sift ordering; return the resulting BDD size
 1: $n \leftarrow u - \ell + 1$
 2: $\pi[i] \leftarrow$ variable currently at level $\ell + i$, for $i = 0, \ldots, n - 1$            ▷ snapshot before any move
 3: **for** $j = 0$ **to** $n - 1$ **do**
 4:      Move variable $\sigma[j]$ to target level $\ell + j$ via adjacent swaps
 5: **end for**
 6: $\Delta \leftarrow \{i \in [0, n) : \sigma[i] \neq \pi[i]\}$            ▷ levels changed by imposing the target order, before MINISIFT
 7: **if** $\Delta \neq \emptyset$ **then**
 8:      MINISIFT($T$, $\ell$, $u$, $\Delta$)
 9: **end if**
10: $\sigma[i] \leftarrow$ variable currently at level $\ell + i$, for $i = 0, \ldots, n - 1$
11: **return** $|T.\text{nodes}|$

---

**Algorithm 3** MINISIFT: targeted local sifting (subroutine of Algorithm 1 and Algorithm 2)

---

**Require:** BDD manager $T$, range $[\ell, u]$, changed-level set $\Delta \subseteq [0, n)$
**Ensure:** Marked levels are used as starting levels for local CUDD sifting within $[\ell, u]$
     **Upward pass: low levels first**
 1: **for** $i = \ell$ **to** $u$ **do**
 2:      **if** $i - \ell \notin \Delta$ **then skip**
 3:      **end if**
 4:      Sift variable at level $i$ upward within $[\ell, u]$; restore the best position found
 5: **end for**
     **Downward pass: high levels first**
 6: **for** $i = u$ **downto** $\ell$ **do**
 7:      **if** $i - \ell \notin \Delta$ **then skip**
 8:      **end if**
 9:      Sift variable at level $i$ downward within $[\ell, u]$; restore the best position found
10: **end for**

---

## D.2    Diagnostic Analysis of MINISIFT Algorithm

This section provides additional diagnostics for the comparison between the standard sifting and MINISIFT. The goal is to understand why MINISIFT can improve downstream QCC.

Table 10: Correspondence between pseudocode functions and CUDD source. $W$ = wrapped (unmodified API call), $A$ = adapted, $N$ = new.

| Pseudocode | CUDD function | Source file | Type | Notes |
|---|---|---|---|---|
| Sift | `cuddSifting` | `cuddReorder.c` | W | — |
| SwapAdjacent | `cuddSwapInPlace` | `cuddReorder.c` | W | — |
| PMX | `PMX` | `cuddGenetic.c` | W | — |
| RouletteSelect | `roulette` | `cuddGenetic.c` | A | Spin range unified to `popsize` for both parent selections |
| PartialSift | `build_dd` | `cuddGenetic.c` | A | Replaces full `cuddSifting` with MiniSift over marked starting levels |
| RestoreOrder | `restoreOrder` | `cuddAnneal.c` | A | Reimplemented locally; based on annealing restore logic (cf. `ddShuffle` in `cuddReorder.c`) |
| MiniSift | — | — | N | New function; internally calls `ddSiftingUp`, `ddSiftingDown`, `ddSiftingBackward` |

**Metrics** The diversity statistics are computed over all 100 functions in the search set and three independent seeds. The unique fraction is the fraction of distinct candidate orderings among the final GA population. The Hamming fraction measures the average normalized Hamming distance between pairs of candidate orderings, where a larger value indicates that the candidate orderings differ at more variable positions. The remaining metrics are computed on the 31 circuits where the GA with the standard sifting and MiniSift produce the same BDD size, but MiniSift achieves lower QCC. Child-node line preservation cases occur when a child value cannot be overwritten at a combining step because it remains required by another reference later in the synthesis. Fewer such cases indicate fewer restrictions on the use of circuit lines and are associated with fewer Toffoli gates and total controls. The toffoli gates, total controls, and QCC are computed from the synthesized reversible circuits.

**Statistical tests** All statistical tests are paired tests between GA with the standard sifting and MiniSift on the same set of circuits or circuit-seed pairs. The Wilcoxon signed-rank test additionally uses the ranks of the absolute paired differences, and therefore accounts for both the direction and relative magnitude of the changes. Smaller $p$-values indicate stronger evidence that the observed changes are systematic rather than random paired fluctuations.

Table 11: Diagnostic comparison between the standard sifting and MiniSift. Diversity rows use all search-set functions and three seeds; other rows use the case when both have the same BDD size but MiniSift lowers QCC. The change is relative to the standard sifting.

| Statistic | Sifting | MiniSift | Change (%) | Wilcoxon $p$ |
|---|---|---|---|---|
| Unique fraction | 0.553 | 0.615 | +11.2% | $4.35 \times 10^{-11}$ |
| Hamming fraction | 0.507 | 0.537 | +5.9% | $1.04 \times 10^{-4}$ |
| Child-node line preservation cases | 999 | 983 | −1.6% | 0.0100 |
| Toffoli gates | 3847 | 3797 | −1.3% | $4.70 \times 10^{-5}$ |
| Total controls | 10591 | 10479 | −1.1% | $4.98 \times 10^{-6}$ |
| Total QCC | 22550 | 22274 | −1.2% | $1.12 \times 10^{-6}$ |

Table 11 shows that MiniSift produces more diverse set of the candidate orderings than the standard sifting. On the same BDD size subset where MiniSift achieves lower QCC, it also reduces the child-node line preservation cases, Toffoli gates, total controls, and QCC. These results suggest that MiniSift can expose ordering candidates that have comparable BDD size but are more favorable for synthesis.

**Example.** Figure 14 shows a representative four-input, single-output augmented RevLib function from the search set, which is one of the 31 functions analyzed in Table 11. Standard sifting makes the order $(x_0, x_3, x_1, x_2)$, whereas MINISIFT makes $(x_3, x_0, x_1, x_2)$. Swapping only the top two variables preserves the BDD size while reducing QCC from 40 to 30. Solid and dashed edges denote then and else branches, respectively. An open circle marks a complemented edge, indicating that the complemented value of the referenced child function is used. Gray nodes are referenced by at least two parents. In the BDD produced by standard sifting, the upper-right $x_3$ node has two children, the gray $x_1$ and gray $x_2$ nodes. Because the gray $x_1$ node is also referenced elsewhere, its synthesized value must be preserved at this step, producing a child-noe line preservation case. In the BDD produced by MINISIFT, the corresponding upper-right $x_0$ node instead points to the same $x_1$ child on both branches, with one branch complemented. This eliminates that preservation case. Accordingly, the synthesis trace contains one fewer child-node line preservation case, together with two fewer Toffoli gates and five fewer controls. Across all 31 same-BDD-size functions, the results in Table 11 are consistent with this observation, showing fewer preservation cases together with statistically significant reductions in Toffoli gates, total controls, and QCC.

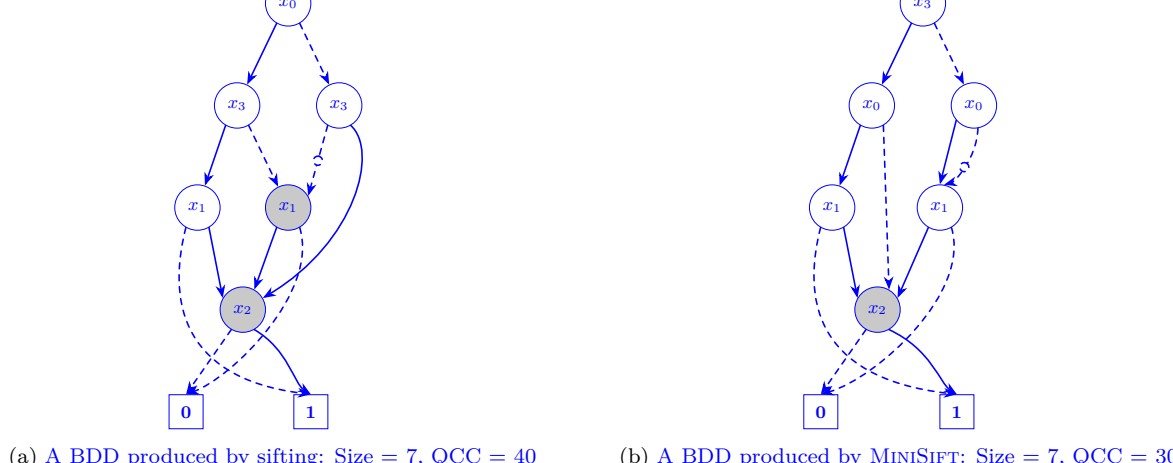

(a) A BDD produced by sifting: Size = 7, QCC = 40      (b) A BDD produced by MINISIFT: Size = 7, QCC = 30

Figure 14: A representative same-BDD-size example under sifting and MINISIFT. Gray nodes are referenced by at least two parents and indicate candidate sites for child-node line preservation. MINISIFT preserves the BDD size while eliminating one preservation case in the synthesis trace, reducing QCC.

### D.3 Per-instance performance comparison

The following presents statistical tests comparing HGA-QE against the baselines, followed by an analysis of the loss cases.

### D.3.1 Statistical test

Table 12 evaluates both the magnitude and the consistency of HGA-QE's per-circuit QCC differences against all six baselines. Mean $\Delta$QCC measures the average size and direction of the difference, where a negative value favors HGA-QE. For each baseline, we apply a paired Wilcoxon signed-rank test to the per-circuit QCC differences. The raw $p$-value reports the result of that comparison alone, and the Holm-adjusted $p$-value accounts for the six baseline comparisons conducted within the same suite. Statistical significance is determined using the adjusted values at $\alpha = 0.05$. Across all 148 functions, HGA-QE achieves significantly lower QCC than every baseline after Holm correction. The suite-level evidence is less uniform. On RevLib, HGA-QE significantly outperforms the three sifting-based methods

and BDD2Seq, while no significant difference is detected relative to GA or SA. On LGSynth91, HGA-QE significantly outperforms every baseline except GA. No comparison is significant on ISCAS85/89. This suite contains only four circuits and several tied results, so the tests have insufficient power and should not be interpreted as evidence that the methods perform equally.

Table 12: Paired Wilcoxon signed-rank tests comparing the per-circuit best QCC of HGA-QE with each baseline. The six comparisons within each suite are corrected using the Holm–Bonferroni procedure. Mean $\Delta$QCC is computed as HGA-QE minus the baseline, so negative values favor HGA-QE. Bold Holm-adjusted $p$-values indicate significance at $\alpha = 0.05$.

| Suite | Baseline | Mean $\Delta$QCC | $p$ (raw) | $p$ (Holm) |
|---|---|---:|---:|---:|
| RevLib (96) | SA | -0.21 | 0.571 | 0.571 |
| | BDD2Seq | -39.99 | 0.008 | **0.025** |
| | GA | -29.24 | 0.069 | 0.139 |
| | SIFT | -351.20 | $8.1 \times 10^{-8}$ | $\mathbf{3.2 \times 10^{-7}}$ |
| | SYMM_SIFT | -341.85 | $5.0 \times 10^{-8}$ | $\mathbf{2.5 \times 10^{-7}}$ |
| | GROUP_SIFT | -272.52 | $6.9 \times 10^{-9}$ | $\mathbf{4.1 \times 10^{-8}}$ |
| LGSynth91 (48) | SA | -65.04 | 0.003 | **0.006** |
| | BDD2Seq | -377.46 | $7.0 \times 10^{-4}$ | **0.002** |
| | GA | -123651.40 | 0.055 | 0.055 |
| | SIFT | -2599.35 | $4.5 \times 10^{-6}$ | $\mathbf{2.2 \times 10^{-5}}$ |
| | SYMM_SIFT | -2594.17 | $4.3 \times 10^{-6}$ | $\mathbf{2.2 \times 10^{-5}}$ |
| | GROUP_SIFT | -3674.27 | $2.6 \times 10^{-6}$ | $\mathbf{1.5 \times 10^{-5}}$ |
| ISCAS85/89 (4) | SA | -151.75 | 0.317 | 0.653 |
| | BDD2Seq | -1415.25 | 0.109 | 0.653 |
| | GA | -161.75 | 0.317 | 0.653 |
| | SIFT | -7095.75 | 0.125 | 0.653 |
| | SYMM_SIFT | -7095.75 | 0.125 | 0.653 |
| | GROUP_SIFT | -11351.75 | 0.109 | 0.653 |
| All (148) | SA | -25.33 | 0.004 | **0.008** |
| | BDD2Seq | -186.61 | $4.0 \times 10^{-6}$ | $\mathbf{1.2 \times 10^{-5}}$ |
| | GA | -40126.49 | 0.006 | **0.008** |
| | SIFT | -1262.61 | $3.2 \times 10^{-13}$ | $\mathbf{1.3 \times 10^{-12}}$ |
| | SYMM_SIFT | -1254.87 | $1.3 \times 10^{-13}$ | $\mathbf{6.6 \times 10^{-13}}$ |
| | GROUP_SIFT | -1675.23 | $1.9 \times 10^{-14}$ | $\mathbf{1.1 \times 10^{-13}}$ |

### D.3.2 Loss cases

Table 13 provides two complementary views of the loss cases. Table 13a applies the same best-or-not criterion, where ties count as best, to HGA-QE and the three strongest baselines. HGA-QE has the lowest loss rate overall and on every benchmark suite. Table 13b further shows that, among HGA-QE's 43 losses, BDD2Seq and SA account for 21 and 11 cases, respectively, confirming that they remain its strongest competitors.

### D.4 Application of HGA-QE to a Logic-Synthesis Flow

We further evaluate HGA-QE in an Field-Programmable Gate Array (FPGA) logic-synthesis flow. Prior work has examined how BDD variable ordering affects the quality of circuits synthesized from the resulting BDDs (Popel, 2002; Drechsler et al., 2004). We extend this evaluation to FPGA technology mapping, which transforms a Boolean network into a network of fixed-input lookup tables (LUT) supported by the target FPGA architecture (Cong & Ding, 1994). For this purpose, each ordered BDD is exported as a Boolean network in which each BDD node is represented as a multiplexer. The structural size of the resulting multiplexer network closely follows the BDD size. The network is then synthesized and mapped to four-input LUTs using Yosys (Wolf & Glaser, 2013).

Table 13: Comparison and breakdown of non-best cases.

(a) Loss rates by method.

| Method | RevLib | LGSynth91 | ISCAS85/89 | Overall (148) |
|--------|--------|-----------|------------|---------------|
| HGA-QE | 26/96 | 17/48 | 0/4 | **43/148 (29.1%)** |
| SA | 28/96 | 26/48 | 1/4 | 55/148 (37.2%) |
| BDD2Seq | 31/96 | 26/48 | 3/4 | 60/148 (40.5%) |
| GA | 38/96 | 25/48 | 1/4 | 64/148 (43.2%) |

(b) Breakdown of HGA-QE losses.

| Suite | HGA-QE losses | BDD2Seq | SA | GA | Others |
|-------|---------------|---------|----|----|--------|
| RevLib | 26/96 | 15 | 7 | 1 | 3 |
| LGSynth91 | 17/48 | 6 | 4 | 6 | 1 |
| ISCAS85/89 | 0/4 | 0 | 0 | 0 | 0 |
| Overall | 43/148 | 21 | 11 | 7 | 4 |

Table 14: Mux-network size and LUT-mapping results under a uniform 300s-per-seed budget for each stage. The Net. Size and # LUT rows use 49 and 48 complete-case circuits, respectively, for which all methods completed their required runs. Success reports each method's full completion count over all 52 circuits. Best includes ties, whereas Strictly Best denotes a unique best result. Mean quality and geometric mean runtime values are computed over the same complete-case set within each stage.

| Method | Stage | Success | Best | Strict | Mean Best | Mean Avg | Mean Worst | Geo-mean Runtime (s) |
|--------|-------|---------|------|--------|-----------|----------|------------|----------------------|
| HGA-QE | Net. Size | 49/52 | **47/49** | 5/49 | 217.39 | 222.12 | **227.86** | 0.066 |
| | # LUT | 49/52 | 30/48 | 3/48 | 83.46 | 87.45 | 91.73 | **3.036** |
| SA | Net. Size | **52/52** | 36/49 | 1/49 | **215.47** | **222.06** | 235.41 | 0.092 |
| | # LUT | **52/52** | **35/48** | 6/48 | 85.04 | 90.52 | 98.50 | 3.041 |
| GA | Net. Size | **52/52** | 42/49 | 1/49 | 274.24 | 1575.83 | 2564.90 | 0.080 |
| | # LUT | 50/52 | 30/48 | 4/48 | **83.38** | **87.24** | **90.35** | 3.060 |
| SIFT | Net. Size | **52/52** | 18/49 | 0/49 | 369.10 | 369.10 | 369.10 | **0.023** |
| | # LUT | **52/52** | 30/48 | 4/48 | 134.52 | 134.52 | 134.52 | 3.229 |

We conduct the experiment on 52 functions from LGSynth91 and ISCAS85/89, as these suites contain conventional Boolean logic benchmarks that are more directly aligned with FPGA synthesis and technology mapping than the reversible circuits in RevLib. The result is summarized in table 14. Sifting is deterministic and is run once, whereas GA, SA, and HGA-QE are each run with five seeds. For each benchmark and method, we report the best, average, and worst multiplexer-network sizes and LUT counts across the runs. Aggregate statistics include only benchmarks for which all required seeds completed within the 300s budget for each stage. HGA-QE completed only partially on `c432` and `c880`, while no seed completed on `rot_orig`. For GA, downstream mapping completed only partially on `pair_orig` and did not complete on `rot_orig`. These benchmarks were therefore excluded from the aggregate statistics. HGA-QE achieves the highest Best/Total count for the exported mux networks. But it does not outperform the strongest baselines, SA and GA, under the objective related to FPGA mapping.

# E    Offline and Online Cost

We distinguish between the one-time offline cost of discovering HGA-QE and the recurring online cost of applying the discovered heuristic to a new Boolean function. The offline phase includes LLM inference, candidate generation, compilation, evaluation on the search set, reversible synthesis, and QCC evaluation. The online phase begins after HGA-QE has been selected and compiled. It requires no LLM calls and consists only of applying the fixed `C` language heuristic, followed by synthesis and QCC evaluation.

**Offline Search Cost**   The `QuantumEvo` setting consists of 10 independent searches using `gpt-oss-120b`. Across these runs, each search used an average of 291.6 LLM calls, 3.11M input tokens, and 1.04M output tokens. The mean wall-clock time was 108.5 minutes. The model was self-hosted on a single NVIDIA RTX PRO 6000 GPU. We report only the inference resources used during the search and treat the cost of pretraining the underlying LLM as external. Classical CUDD heuristics require no analogous discovery phase. BDD2Seq reports its training hardware and hyperparameters, but not the wall-clock time or GPU-hours required for model training. Its original offline cost therefore cannot be directly compared with that of `QuantumEvo`.

Table 15: Average offline search requirements of one `QuantumEvo` production run.

| LLM Calls | Input Tokens | Output Tokens | Wall-clock | Hardware |
|-----------|--------------|---------------|------------|----------|
| 291.6 | 3.11M | 1.04M | 108.5 min | 1× NVIDIA RTX PRO 6000 |

**Offline Compilation and Evaluation Cost**   Each candidate reaching the evaluation stage is compiled and evaluated on the 100 function search set. For every function, the evaluation invokes the reversible-synthesis pipeline twice before computing QCC. With a maximum of 200 candidate evaluations per run, this corresponds to up to approximately 40,000 synthesis calls per run. In the `QuantumEvo` setting, 8.3% of candidate evaluations reached the 40 s timeout. The completed evaluations required a mean of 9.3 s, a median of 7.0 s, and a 95th percentile of 26.0 s.

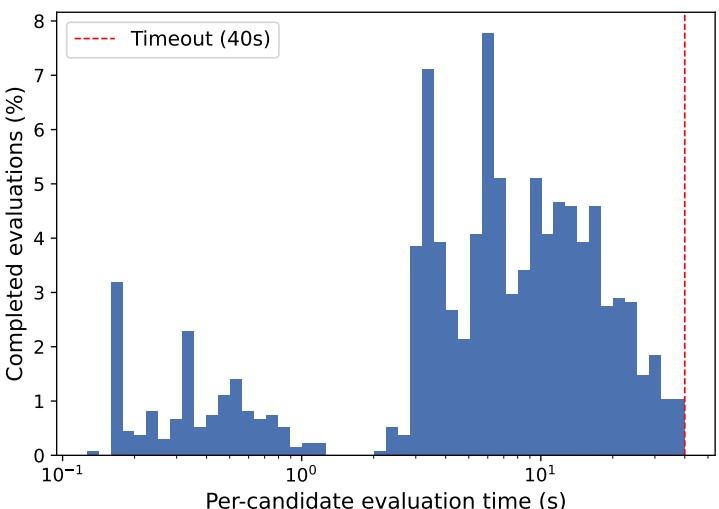

Figure 15: Distribution of per-candidate evaluation times for completed evaluations in the production setting. The dashed line indicates the 40 s timeout threshold.

**Online Cost**   Applying HGA-QE requires no LLM calls. After discovery, HGA-QE is deployed as a fixed compiled heuristic. Its online cost therefore consists only of BDD reordering, reversible synthesis, and QCC evaluation, all of which are included in the benchmark runtime measurements. Unlike HGA-QE, BDD2Seq requires per-instance neural inference.

## F   Prompts

We show the problem-specific prompts used in `QuantumEvo`, which are function signature, function description, and external knowledge. They are tailored to BDD variable ordering heuristic design and to

the `C` language individual representation. The remaining operator prompts, including mutation, crossover, and reflection, are available in the accompanying code.

```
int cuddFunc(DdManager *table, int lower, int upper)
```

Prompt 1: Function Signature

```
Implement 'cuddFunc()' for CUDD BDD reordering.
Primary goal:  reduce native BDD size and Quantum Cost.
If output hooks are present, you may also consult 'CURRENT_SIZE(table)', but native BDD
heap size is a valid primary objective.

Requirements:
- Return '1' on success and '0' on failure.
- On OOM, set 'table->errorCode = CUDD_MEMORY_OUT' and return '0'.
- On return, the live order must equal the best order found.
- Respect the active level range '[lower, upper]'.
- Free owned arrays with 'FREE' and owned move lists with 'cuddDeallocMove'.

Key CUDD contracts:
- 'cuddSwapInPlace(table, x, x + 1)' takes LEVELS, not variable indices.  Pass the lower
 adjacent level first; 'cuddSwapInPlace(table, curLevel, curLevel - 1)' is wrong.
- Keep 'Cudd_SetFuncOutputs(...)' and 'Cudd_ClearFuncOutputs(...)' exactly named.
- 'ddUndoMoves(moves)' returns an inverse history; free the original move list after calling
 it, and free the full list, not just the head node.
- For plain sifting, compose the rollback list as 'down + inv' before
'ddSiftingBackward(...)'.
- 'ddSiftingBackward(...)' does not free the move list for you.
- For annealing-style code, keep an explicit best-order snapshot and restore the best order
 before returning.
```

Prompt 2: Function Description

```
'cuddFunc()' should behave like a native CUDD reorderer.

Common mistakes:
- Passing a variable index where a level is required, especially in 'cuddSwapInPlace()'.
- Passing adjacent levels in reverse order.  Use 'cuddSwapInPlace(table, curLevel - 1,
 curLevel)', not 'cuddSwapInPlace(table, curLevel, curLevel - 1)'.
- Calling 'Cudd_ReduceHeap()' or another nested top-level reorderer inside 'cuddFunc()'.
- Returning without restoring the best order found.
- Mishandling move ownership:  freeing '(Move *)CUDD_OUT_OF_MEM', freeing only the head
 after 'ddUndoMoves(...)', or double-freeing move lists.
- Building the wrong rollback order for plain sifting; use 'down + inv', not 'inv + down'.

Do not:
- Swap outside '[lower, upper]'.
- Manually increment 'table->ddTotalNumberSwapping'.
- Redefine 'Move', 'IndexKey', 'ddMin', 'ddMax', 'cuddDynamicAllocNode', or
'cuddDeallocMove'.
- Use headers other than 'util.h', 'cuddInt.h', 'math.h', and 'string.h'.
```

Prompt 3: External Knowledge

## G    Full Per-Instance Results

Table 16–Table 21 report the full per-function results for all 148 benchmark functions, grouped by the suite. RevLib instances are listed in Table 16–Table 20. ISCAS85/89 and LGSynth91 instances are listed in Table 18 and Table 21 and denoted by superscripts [I] and [L], respectively. For ordering methods

with stochastic components, we report best QCC, average QCC with variance, and average runtime; for BDD2Seq and deterministic methods, we report best QCC and runtime.

**BDD2Seq runtime normalization.** The runtimes reported for BDD2Seq include both variable ordering and reversible synthesis. BDD2Seq(E*) synthesizes only its final ordering, whereas BDD2Seq(B*) uses beam search with a beam width of 20 and synthesizes all 20 candidate orderings. We leave the reported BDD2Seq(E*) runtime unchanged because its single synthesis call accounts for only a small fraction of its total runtime. For BDD2Seq(B*), we normalize only the estimated cost of the 19 additional CPU-based synthesis calls from the original dual Xeon 8375C platform to our single-thread Ryzen 9 5900X platform. Let $T_{B^*}$ and $T_{E^*}$ denote the reported per-circuit runtimes on the original dual Xeon 8375C platform. Assuming that $T_{B^*} - T_{E^*}$ corresponds to the 19 additional CPU-based synthesis calls, we rescale this component to our single-thread Ryzen 9 5900X platform using the PassMark single-thread ratio $\gamma = 2474/3465 \approx 0.714$:

$$T_{B^*}^{\mathrm{adj}} \approx T_{E^*} + \frac{20\gamma - 1}{19} \left( T_{B^*} - T_{E^*} \right) = T_{E^*} + 0.699 \left( T_{B^*} - T_{E^*} \right).$$

Table 16: Full benchmark results, RevLib functions, stochastic methods, part 1 of 2. Best and average QCC compared among QuantumEvo, GA, and SA; lowest values bold. Small values under Avg are $\pm$ one standard deviation of QCC across that function's 5 seeds.

| Function | I | O | QuantumEvo | | | GA | | | SA | | |
|---|---|---|---|---|---|---|---|---|---|---|---|
| | | | Best | Avg $\pm$Std | Time | Best | Avg $\pm$Std | Time | Best | Avg $\pm$Std | Time |
| 4gt10_22 | 4 | 1 | **17** | **17.00** $_{\pm 0}$ | 0.04 | **17** | **17.00** $_{\pm 0}$ | 0.06 | **17** | **17.00** $_{\pm 0}$ | 0.05 |
| 4gt11_23 | 4 | 1 | **5** | **5.00** $_{\pm 0}$ | 0.06 | **5** | **5.00** $_{\pm 0}$ | 0.06 | **5** | **5.00** $_{\pm 0}$ | 0.05 |
| 4gt12_24 | 4 | 1 | **17** | **17.00** $_{\pm 0}$ | 0.06 | **17** | **17.00** $_{\pm 0}$ | 0.06 | **17** | **17.00** $_{\pm 0}$ | 0.06 |
| 4gt13_25 | 4 | 1 | **10** | **10.00** $_{\pm 0}$ | 0.06 | **10** | **10.00** $_{\pm 0}$ | 0.06 | **10** | **10.00** $_{\pm 0}$ | 0.05 |
| 4gt4_20 | 4 | 1 | **19** | **19.00** $_{\pm 0}$ | 0.06 | **19** | **19.00** $_{\pm 0}$ | 0.06 | **19** | **19.00** $_{\pm 0}$ | 0.06 |
| 4gt5_21 | 4 | 1 | **12** | **12.00** $_{\pm 0}$ | 0.05 | **12** | **12.00** $_{\pm 0}$ | 0.06 | **12** | **12.00** $_{\pm 0}$ | 0.06 |
| 4mod5_8 | 4 | 1 | **24** | **24.00** $_{\pm 0}$ | 0.06 | **24** | **24.00** $_{\pm 0}$ | 0.06 | **24** | **24.00** $_{\pm 0}$ | 0.05 |
| 4mod7_26 | 4 | 3 | **86** | **86.00** $_{\pm 0}$ | 0.06 | **86** | **86.00** $_{\pm 0}$ | 0.05 | **86** | **86.00** $_{\pm 0}$ | 0.06 |
| 5xp1_90 | 7 | 10 | **254** | **254.00** $_{\pm 0}$ | 0.06 | 280 | 280.00 $_{\pm 0}$ | 0.06 | **254** | **254.00** $_{\pm 0}$ | 0.07 |
| 9symml_91 | 9 | 1 | **206** | **206.00** $_{\pm 0}$ | 0.06 | **206** | **206.00** $_{\pm 0}$ | 0.06 | **206** | **206.00** $_{\pm 0}$ | 0.07 |
| add6_92 | 12 | 7 | **118** | **118.00** $_{\pm 0}$ | 0.07 | **118** | **118.00** $_{\pm 0}$ | 0.08 | **118** | 122.60 $_{\pm 9.20}$ | 0.09 |
| adr4_93 | 8 | 5 | **74** | **74.00** $_{\pm 0}$ | 0.05 | **74** | **74.00** $_{\pm 0}$ | 0.06 | **74** | **74.00** $_{\pm 0}$ | 0.07 |
| alu1_94 | 12 | 8 | **139** | **139.00** $_{\pm 0}$ | 0.06 | **139** | **139.00** $_{\pm 0}$ | 0.07 | **139** | **139.00** $_{\pm 0}$ | 0.07 |
| alu2_96 | 10 | 6 | **1266** | **1276.80** $_{\pm 8.82}$ | 0.08 | 1269 | 1287.00 $_{\pm 11.22}$ | 0.09 | **1266** | 1278.80 $_{\pm 9.62}$ | 0.11 |
| alu3_97 | 10 | 8 | **430** | 434.80 $_{\pm 3.92}$ | 0.06 | **430** | **430.00** $_{\pm 0}$ | 0.07 | **430** | 431.60 $_{\pm 3.20}$ | 0.10 |
| alu4_98 | 14 | 8 | 4385 | 4422.00 $_{\pm 30.35}$ | 0.19 | **4255** | **4341.20** $_{\pm 47.48}$ | 0.20 | **4255** | 4361.40 $_{\pm 68.79}$ | 0.33 |
| alu_9 | 5 | 1 | **29** | **29.00** $_{\pm 0}$ | 0.05 | **29** | **29.00** $_{\pm 0}$ | 0.06 | **29** | **29.00** $_{\pm 0}$ | 0.06 |
| apex2_101 | 39 | 3 | 2663 | **2697.20** $_{\pm 25.46}$ | 0.62 | 2726 | 2929.00 $_{\pm 157.30}$ | 1.82 | **2660** | 2776.00 $_{\pm 101.95}$ | 0.89 |
| apex4_103 | 9 | 19 | 8236 | 8243.60 $_{\pm 3.83}$ | 0.13 | **8231** | **8235.80** $_{\pm 4.96}$ | 0.12 | **8231** | 8240.40 $_{\pm 5.85}$ | 0.26 |
| apex5_104 | 117 | 88 | 9184 | 9812.40 $_{\pm 489.75}$ | 0.97 | 9551 | 10041.40 $_{\pm 392.32}$ | 1.24 | **9397** | **9809.00** $_{\pm 448.99}$ | 1.97 |
| apla_107 | 10 | 12 | **731** | 734.60 $_{\pm 2.94}$ | 0.07 | **731** | **732.20** $_{\pm 2.40}$ | 0.08 | **731** | **732.20** $_{\pm 2.40}$ | 0.10 |
| bw_116 | 5 | 28 | **926** | **930.40** $_{\pm 5.39}$ | 0.06 | **926** | 934.80 $_{\pm 4.40}$ | 0.06 | **926** | **930.40** $_{\pm 5.39}$ | 0.08 |
| C17_117 | 5 | 2 | **37** | **37.00** $_{\pm 0}$ | 0.06 | **37** | **37.00** $_{\pm 0}$ | 0.06 | **37** | **37.00** $_{\pm 0}$ | 0.06 |
| C7552_119 | 5 | 16 | **202** | **202.00** $_{\pm 0}$ | 0.06 | **202** | **202.00** $_{\pm 0}$ | 0.05 | **202** | **202.00** $_{\pm 0}$ | 0.06 |
| clip_124 | 9 | 5 | 515 | 521.40 $_{\pm 4.59}$ | 0.06 | **512** | **517.80** $_{\pm 6.27}$ | 0.07 | 519 | 522.20 $_{\pm 3.66}$ | 0.09 |
| cm150a_128 | 21 | 1 | **186** | **186.00** $_{\pm 0}$ | 0.52 | **186** | **186.00** $_{\pm 0}$ | 0.33 | **186** | **186.00** $_{\pm 0}$ | 0.30 |
| cm151a_129 | 19 | 9 | **298** | **298.00** $_{\pm 0}$ | 0.10 | **298** | **298.00** $_{\pm 0}$ | 0.12 | **298** | **298.00** $_{\pm 0}$ | 0.09 |
| cm152a_130 | 11 | 1 | **62** | **62.00** $_{\pm 0}$ | 0.07 | **62** | **62.00** $_{\pm 0}$ | 0.07 | **62** | **62.00** $_{\pm 0}$ | 0.07 |
| cm163a_133 | 16 | 13 | **244** | 254.40 $_{\pm 10.52}$ | 0.08 | **244** | **253.20** $_{\pm 11.27}$ | 0.09 | **244** | 261.20 $_{\pm 14.05}$ | 0.07 |
| cm42a_125 | 4 | 10 | **117** | **117.00** $_{\pm 0}$ | 0.06 | **117** | **117.00** $_{\pm 0}$ | 0.05 | **117** | **117.00** $_{\pm 0}$ | 0.06 |
| cm82a_126 | 5 | 3 | **44** | **44.00** $_{\pm 0}$ | 0.06 | **44** | **44.00** $_{\pm 0}$ | 0.06 | **44** | **44.00** $_{\pm 0}$ | 0.08 |
| cm85a_127 | 11 | 3 | **203** | **205.40** $_{\pm 1.96}$ | 0.06 | 207 | 210.20 $_{\pm 1.60}$ | 0.07 | **203** | 208.20 $_{\pm 3.49}$ | 0.08 |
| cmb_134 | 16 | 4 | **153** | **153.00** $_{\pm 0}$ | 0.08 | **153** | **153.00** $_{\pm 0}$ | 0.11 | **153** | **153.00** $_{\pm 0}$ | 0.09 |
| co14_135 | 14 | 1 | **159** | **159.00** $_{\pm 0}$ | 0.08 | **159** | **159.00** $_{\pm 0}$ | 0.10 | **159** | **159.00** $_{\pm 0}$ | 0.08 |
| con1_136 | 7 | 2 | **89** | **90.20** $_{\pm 2.40}$ | 0.06 | **89** | **90.20** $_{\pm 2.40}$ | 0.06 | **89** | 93.80 $_{\pm 2.40}$ | 0.06 |
| cordic_138 | 23 | 2 | **304** | **308.00** $_{\pm 3.74}$ | 0.16 | 306 | 312.00 $_{\pm 3.74}$ | 0.23 | 316 | 323.20 $_{\pm 3.60}$ | 0.17 |
| cu_141 | 14 | 11 | **219** | **219.80** $_{\pm 0.40}$ | 0.08 | 224 | 224.00 $_{\pm 0}$ | 0.09 | 220 | 220.00 $_{\pm 0}$ | 0.09 |
| dc1_142 | 4 | 7 | **186** | **186.00** $_{\pm 0}$ | 0.06 | **186** | **186.00** $_{\pm 0}$ | 0.06 | **186** | **186.00** $_{\pm 0}$ | 0.06 |
| dc2_143 | 8 | 7 | **431** | **431.00** $_{\pm 0}$ | 0.06 | **431** | **431.00** $_{\pm 0}$ | 0.06 | **431** | **431.00** $_{\pm 0}$ | 0.07 |
| decod24-enable_32 | 3 | 4 | **38** | **38.00** $_{\pm 0}$ | 0.06 | **38** | **38.00** $_{\pm 0}$ | 0.05 | **38** | **38.00** $_{\pm 0}$ | 0.06 |
| decod_137 | 5 | 16 | **202** | **202.00** $_{\pm 0}$ | 0.05 | **202** | **202.00** $_{\pm 0}$ | 0.05 | **202** | **202.00** $_{\pm 0}$ | 0.06 |
| dist_144 | 8 | 5 | **975** | 975.80 $_{\pm 1.60}$ | 0.06 | 979 | 979.00 $_{\pm 0}$ | 0.07 | **975** | **975.00** $_{\pm 0}$ | 0.09 |
| dk17_145 | 10 | 11 | **426** | **426.00** $_{\pm 0}$ | 0.06 | **426** | **426.00** $_{\pm 0}$ | 0.07 | **426** | **426.00** $_{\pm 0}$ | 0.09 |
| dk27_146 | 9 | 9 | **140** | 140.80 $_{\pm 1.60}$ | 0.06 | 144 | 144.00 $_{\pm 0}$ | 0.06 | **140** | **140.00** $_{\pm 0}$ | 0.06 |
| e64_149 | 65 | 65 | **886** | **886.00** $_{\pm 0}$ | 0.39 | **886** | **886.00** $_{\pm 0}$ | 1.14 | **886** | 890.20 $_{\pm 8.40}$ | 1.07 |
| ex1010_155 | 10 | 10 | 9681 | 9724.60 $_{\pm 33.76}$ | 0.23 | **9643** | **9703.20** $_{\pm 54.16}$ | 0.22 | 9717 | 9803.60 $_{\pm 50.53}$ | 0.36 |
| ex1_150 | 5 | 1 | **8** | **8.00** $_{\pm 0}$ | 0.06 | **8** | **8.00** $_{\pm 0}$ | 0.05 | **8** | **8.00** $_{\pm 0}$ | 0.06 |
| ex2_151 | 5 | 1 | **60** | **60.00** $_{\pm 0}$ | 0.06 | **60** | **60.00** $_{\pm 0}$ | 0.06 | **60** | 64.00 $_{\pm 4.90}$ | 0.06 |

Table 17: Full benchmark results, RevLib functions, stochastic methods, part 2 of 2.

| Function | I | O | QuantumEvo | | | GA | | | SA | | |
|---|---|---|---|---|---|---|---|---|---|---|---|
| | | | Best | Avg ±Std | Time | Best | Avg ±Std | Time | Best | Avg ±Std | Time |
| ex3_152 | 5 | 1 | **38** | **40.00** $_{\pm 4}$ | 0.06 | **38** | 44.00 $_{\pm 4.90}$ | 0.06 | **38** | 45.00 $_{\pm 4}$ | 0.06 |
| ex5p_154 | 8 | 63 | **1843** | **1843.00** $_{\pm 0}$ | 0.11 | **1843** | **1843.00** $_{\pm 0}$ | 0.11 | **1843** | **1843.00** $_{\pm 0}$ | 0.14 |
| example2_156 | 10 | 6 | **1266** | **1276.80** $_{\pm 8.82}$ | 0.08 | 1269 | 1287.00 $_{\pm 11.22}$ | 0.09 | **1266** | 1278.80 $_{\pm 9.62}$ | 0.11 |
| f2_158 | 4 | 4 | **108** | 111.00 $_{\pm 2.45}$ | 0.05 | **108** | **108.00** $_{\pm 0}$ | 0.05 | 113 | 113.00 $_{\pm 0}$ | 0.05 |
| f51m_159 | 14 | 8 | **2422** | **2429.20** $_{\pm 4.21}$ | 0.14 | 2427 | 2429.40 $_{\pm 2.94}$ | 0.20 | **2422** | 2430.60 $_{\pm 4.32}$ | 0.25 |
| frg1_160 | 28 | 3 | **559** | **597.40** $_{\pm 38.66}$ | 0.17 | 560 | 610.20 $_{\pm 41.06}$ | 0.24 | 567 | 637.40 $_{\pm 67.22}$ | 0.17 |
| frg2_161 | 143 | 139 | 7017 | **7601.60** $_{\pm 506.67}$ | 1.47 | 9429 | 9429.00 $_{\pm 0}$ | 1.63 | **6972** | 7689.80 $_{\pm 593.33}$ | 3.22 |
| in0_162 | 15 | 11 | 2299 | 2313.40 $_{\pm 14.44}$ | 0.11 | 2299 | 2317.40 $_{\pm 13.53}$ | 0.16 | **2271** | **2295.80** $_{\pm 12.50}$ | 0.19 |
| inc_170 | 7 | 9 | **592** | **592.00** $_{\pm 0}$ | 0.06 | **592** | **592.00** $_{\pm 0}$ | 0.06 | **592** | **592.00** $_{\pm 0}$ | 0.08 |
| life_175 | 9 | 1 | **204** | **204.00** $_{\pm 0}$ | 0.06 | 210 | 210.00 $_{\pm 0}$ | 0.07 | **204** | **204.00** $_{\pm 0}$ | 0.07 |
| majority_176 | 5 | 1 | **41** | **41.00** $_{\pm 0}$ | 0.06 | **41** | **41.00** $_{\pm 0}$ | 0.05 | **41** | **41.00** $_{\pm 0}$ | 0.06 |
| max46_177 | 9 | 1 | 556 | 562.80 $_{\pm 8.89}$ | 0.07 | 546 | 558.00 $_{\pm 6.78}$ | 0.07 | **545** | **552.60** $_{\pm 4.22}$ | 0.09 |
| misex1_178 | 8 | 7 | **279** | 290.40 $_{\pm 9.31}$ | 0.06 | 287 | **287.00** $_{\pm 0}$ | 0.06 | 288 | 288.00 $_{\pm 0}$ | 0.07 |
| misex3_180 | 14 | 14 | **3789** | **3791.40** $_{\pm 2.94}$ | 0.16 | **3789** | 3792.60 $_{\pm 2.94}$ | 0.22 | **3789** | 3833.40 $_{\pm 52.29}$ | 0.35 |
| misex3c_181 | 14 | 14 | **3936** | **3950.40** $_{\pm 10.80}$ | 0.16 | **3936** | **3950.40** $_{\pm 10.80}$ | 0.24 | **3936** | 3979.60 $_{\pm 73.78}$ | 0.32 |
| mlp4_184 | 8 | 8 | **1158** | 1158.40 $_{\pm 0.49}$ | 0.07 | 1159 | 1159.00 $_{\pm 0}$ | 0.07 | **1158** | **1158.00** $_{\pm 0}$ | 0.10 |
| mux_185 | 21 | 1 | **170** | **170.00** $_{\pm 0}$ | 0.55 | **170** | **170.00** $_{\pm 0}$ | 0.41 | **170** | **170.00** $_{\pm 0}$ | 0.41 |
| one-two-three_27 | 3 | 3 | **44** | **44.00** $_{\pm 0}$ | 0.06 | **44** | **44.00** $_{\pm 0}$ | 0.06 | **44** | **44.00** $_{\pm 0}$ | 0.05 |
| parity_188 | 16 | 1 | **31** | **31.00** $_{\pm 0}$ | 0.50 | **31** | **31.00** $_{\pm 0}$ | 0.49 | **31** | **31.00** $_{\pm 0}$ | 0.49 |
| pcler8_190 | 16 | 5 | **124** | **124.00** $_{\pm 0}$ | 0.08 | **124** | **124.00** $_{\pm 0}$ | 0.08 | **124** | **124.00** $_{\pm 0}$ | 0.08 |
| pdc_191 | 16 | 40 | **6599** | **6599.00** $_{\pm 0}$ | 0.41 | **6599** | **6599.00** $_{\pm 0}$ | 0.60 | **6599** | **6599.00** $_{\pm 0}$ | 0.84 |
| pm1_192 | 4 | 10 | **117** | **117.00** $_{\pm 0}$ | 0.06 | **117** | **117.00** $_{\pm 0}$ | 0.06 | **117** | **117.00** $_{\pm 0}$ | 0.06 |
| radd_193 | 8 | 5 | **74** | **74.00** $_{\pm 0}$ | 0.06 | **74** | **74.00** $_{\pm 0}$ | 0.06 | **74** | **74.00** $_{\pm 0}$ | 0.06 |
| rd53_68 | 5 | 3 | **98** | **98.00** $_{\pm 0}$ | 0.05 | **98** | **98.00** $_{\pm 0}$ | 0.06 | **98** | **98.00** $_{\pm 0}$ | 0.05 |
| rd73_69 | 7 | 3 | **217** | **217.00** $_{\pm 0}$ | 0.06 | **217** | **217.00** $_{\pm 0}$ | 0.06 | **217** | **217.00** $_{\pm 0}$ | 0.06 |
| rd84_70 | 8 | 4 | **304** | **304.00** $_{\pm 0}$ | 0.06 | **304** | **304.00** $_{\pm 0}$ | 0.06 | **304** | **304.00** $_{\pm 0}$ | 0.07 |
| root_197 | 8 | 5 | **444** | **444.00** $_{\pm 0}$ | 0.06 | 446 | 446.00 $_{\pm 0}$ | 0.06 | **444** | **444.00** $_{\pm 0}$ | 0.08 |
| ryy6_198 | 16 | 1 | **107** | **109.40** $_{\pm 4.80}$ | 0.09 | **107** | 112.60 $_{\pm 4.80}$ | 0.11 | **107** | 117.20 $_{\pm 5.23}$ | 0.09 |
| sao2_199 | 10 | 4 | **653** | **653.00** $_{\pm 0}$ | 0.07 | **653** | 654.80 $_{\pm 2.23}$ | 0.07 | **653** | 655.80 $_{\pm 2.32}$ | 0.11 |
| seq_201 | 41 | 35 | **9080** | 9151.00 $_{\pm 44.62}$ | 1.17 | 9113 | **9142.00** $_{\pm 26.48}$ | 3.03 | 9165 | 9237.40 $_{\pm 96.16}$ | 1.99 |
| sf_232 | 4 | 1 | **21** | **26.80** $_{\pm 3.25}$ | 0.06 | 43 | 43.00 $_{\pm 0}$ | 0.06 | 36 | 36.00 $_{\pm 0}$ | 0.06 |
| spla_202 | 16 | 46 | **5858** | **5860.00** $_{\pm 2.45}$ | 0.33 | **5858** | **5860.00** $_{\pm 2.45}$ | 0.35 | **5858** | 5861.00 $_{\pm 2.45}$ | 0.56 |
| sqn_203 | 7 | 3 | **356** | 358.00 $_{\pm 2.45}$ | 0.06 | **356** | 358.00 $_{\pm 2.45}$ | 0.06 | **356** | **357.00** $_{\pm 2}$ | 0.07 |
| sqr6_204 | 6 | 12 | 486 | 486.00 $_{\pm 0}$ | 0.06 | **482** | **482.00** $_{\pm 0}$ | 0.06 | 486 | 486.00 $_{\pm 0}$ | 0.07 |
| sqrt8_205 | 8 | 4 | **221** | 233.80 $_{\pm 6.40}$ | 0.06 | **221** | 230.60 $_{\pm 7.84}$ | 0.06 | **221** | **230.60** $_{\pm 7.84}$ | 0.07 |
| squar5_206 | 5 | 8 | **232** | **232.00** $_{\pm 0}$ | 0.06 | **232** | **232.00** $_{\pm 0}$ | 0.05 | **232** | **232.00** $_{\pm 0}$ | 0.06 |
| sym10_207 | 10 | 1 | **253** | **253.00** $_{\pm 0}$ | 0.07 | **253** | **253.00** $_{\pm 0}$ | 0.08 | **253** | **253.00** $_{\pm 0}$ | 0.07 |
| sym6_63 | 6 | 1 | **93** | **93.00** $_{\pm 0}$ | 0.06 | **93** | **93.00** $_{\pm 0}$ | 0.06 | **93** | **93.00** $_{\pm 0}$ | 0.06 |
| sym9_71 | 9 | 1 | **206** | **206.00** $_{\pm 0}$ | 0.06 | **206** | **206.00** $_{\pm 0}$ | 0.07 | **206** | **206.00** $_{\pm 0}$ | 0.06 |
| t481_208 | 16 | 1 | **147** | **147.00** $_{\pm 0}$ | 0.10 | **147** | **147.00** $_{\pm 0}$ | 0.12 | **147** | **147.00** $_{\pm 0}$ | 0.09 |
| table3_209 | 14 | 14 | **5898** | **5899.40** $_{\pm 2.80}$ | 0.18 | **5898** | 5902.20 $_{\pm 3.43}$ | 0.22 | **5898** | 5908.80 $_{\pm 21.60}$ | 0.43 |
| tial_214 | 14 | 8 | 4460 | 4487.60 $_{\pm 20.76}$ | 0.13 | 4354 | 4401.00 $_{\pm 30.13}$ | 0.21 | **4231** | **4299.20** $_{\pm 47.13}$ | 0.38 |
| urf4_89 | 11 | 11 | **28406** | **28537.40** $_{\pm 101.35}$ | 0.41 | 28538 | 28591.80 $_{\pm 36.85}$ | 0.54 | 28489 | 28590.80 $_{\pm 80.86}$ | 0.97 |
| wim_220 | 4 | 7 | **107** | **107.00** $_{\pm 0}$ | 0.06 | **107** | **107.00** $_{\pm 0}$ | 0.06 | **107** | **107.00** $_{\pm 0}$ | 0.05 |
| x2_223 | 10 | 7 | **191** | **191.00** $_{\pm 0}$ | 0.06 | **191** | **191.00** $_{\pm 0}$ | 0.07 | **191** | **191.00** $_{\pm 0}$ | 0.08 |
| xor5_195 | 5 | 1 | **8** | **8.00** $_{\pm 0}$ | 0.05 | **8** | **8.00** $_{\pm 0}$ | 0.06 | **8** | **8.00** $_{\pm 0}$ | 0.06 |
| z4_224 | 7 | 4 | **66** | **66.00** $_{\pm 0}$ | 0.06 | **66** | **66.00** $_{\pm 0}$ | 0.05 | **66** | **66.00** $_{\pm 0}$ | 0.06 |
| z4ml_225 | 7 | 4 | **66** | **66.00** $_{\pm 0}$ | 0.05 | **66** | **66.00** $_{\pm 0}$ | 0.06 | **66** | **66.00** $_{\pm 0}$ | 0.06 |

Table 18: Full benchmark results, ISCAS85/89 and LGSynth91 functions, stochastic methods.

| Function | I | O | QuantumEvo | | | GA | | | SA | | |
|---|---|---|---|---|---|---|---|---|---|---|---|
| | | | Best | Avg ±Std | Time | Best | Avg ±Std | Time | Best | Avg ±Std | Time |
| apex6_orig[L] | 135 | 99 | **3736** | **3787.20** ±36.31 | 0.65 | 4159 | 4368.40 ±169.77 | 0.64 | 4392 | 4551.40 ±148.45 | 1.08 |
| apex7_orig[L] | 49 | 37 | **1431** | **1440.60** ±7.45 | 0.28 | 1441 | 1443.40 ±2.94 | 0.58 | 1703 | 1759.20 ±46.63 | 0.44 |
| b1_orig[L] | 3 | 4 | **22** | 23.00 ±2 | 0.06 | **22** | **22.00** ±0 | 0.05 | **22** | **22.00** ±0 | 0.06 |
| b9_orig[L] | 41 | 21 | **713** | 719.40 ±6.59 | 0.17 | 715 | **719.00** ±3.58 | 0.27 | 804 | 804.00 ±0 | 0.19 |
| c17[I] | 5 | 2 | 37 | 37.00 ±0 | 0.06 | 37 | 37.00 ±0 | 0.06 | 37 | 37.00 ±0 | 0.06 |
| c432[I] | 36 | 7 | **10494** | **11053.20** ±305.36 | 245.19 | 11141 | 11170.60 ±40.35 | 6.20 | 11101 | 11148.60 ±95.20 | 1.46 |
| c880[I] | 60 | 26 | **34241** | 34761.40 ±367.34 | 329.32 | **34241** | 35095.20 ±528.82 | 44.13 | **34241** | **34610.60** ±321.65 | 12.39 |
| c8_orig[L] | 28 | 18 | **436** | **436.00** ±0 | 0.14 | **436** | 446.40 ±6.97 | 0.18 | **436** | 450.80 ±11.65 | 0.15 |
| cc_orig[L] | 21 | 20 | 283 | 289.00 ±5.83 | 0.08 | **267** | **280.00** ±8.12 | 0.11 | 278 | 287.20 ±5.15 | 0.08 |
| cht_orig[L] | 47 | 36 | **714** | **714.00** ±0 | 0.12 | **714** | **714.00** ±0 | 0.21 | **714** | **714.00** ±0 | 0.14 |
| cm138a_orig[L] | 6 | 8 | **104** | **104.00** ±0 | 0.06 | **104** | **104.00** ±0 | 0.06 | **104** | **104.00** ±0 | 0.06 |
| cm150a_orig[L] | 21 | 1 | **186** | **186.00** ±0 | 0.52 | **186** | **186.00** ±0 | 0.45 | **186** | **186.00** ±0 | 0.36 |
| cm151a_orig[L] | 12 | 2 | **92** | **92.00** ±0 | 0.07 | **92** | **92.00** ±0 | 0.08 | **92** | **92.00** ±0 | 0.07 |
| cm152a_orig[L] | 11 | 1 | **62** | **62.00** ±0 | 0.07 | **62** | **62.00** ±0 | 0.07 | **62** | **62.00** ±0 | 0.07 |
| cm162a_orig[L] | 14 | 5 | 222 | 222.00 ±0 | 0.08 | **219** | **220.20** ±1.47 | 0.09 | **219** | 220.80 ±1.47 | 0.09 |
| cm163a_orig[L] | 16 | 5 | **124** | **124.00** ±0 | 0.08 | **124** | **124.00** ±0 | 0.08 | **124** | **124.00** ±0 | 0.07 |
| cm42a_orig[L] | 4 | 10 | **117** | **117.00** ±0 | 0.05 | **117** | **117.00** ±0 | 0.05 | **117** | **117.00** ±0 | 0.06 |
| cm82a_orig[L] | 5 | 3 | **44** | **44.00** ±0 | 0.06 | **44** | **44.00** ±0 | 0.06 | **44** | **44.00** ±0 | 0.06 |
| cm85a_orig[L] | 11 | 3 | **203** | **205.40** ±1.96 | 0.07 | 207 | 210.20 ±1.60 | 0.07 | **203** | 208.20 ±3.49 | 0.08 |
| cmb_orig[L] | 16 | 4 | **153** | **153.00** ±0 | 0.09 | **153** | **153.00** ±0 | 0.11 | **153** | **153.00** ±0 | 0.09 |
| comp_orig[L] | 32 | 3 | **796** | **796.00** ±0 | 2.41 | 824 | 824.00 ±0 | 2.28 | 897 | 936.00 ±22.08 | 2.01 |
| cordic_orig[L] | 23 | 2 | **304** | **308.00** ±3.74 | 0.12 | 306 | 312.00 ±3.74 | 0.20 | 316 | 323.20 ±3.60 | 0.14 |
| count_orig[L] | 35 | 16 | **441** | **441.00** ±0 | 0.17 | **441** | **441.00** ±0 | 0.29 | **441** | **441.00** ±0 | 0.21 |
| cu_orig[L] | 14 | 11 | **219** | 219.80 ±0.40 | 0.08 | 224 | 224.00 ±0 | 0.09 | 220 | **220.00** ±0 | 0.09 |
| dalu_orig[L] | 75 | 16 | **5230** | 5301.60 ±66.64 | 71.73 | 5240 | **5291.20** ±36.74 | 71.72 | 5316 | 5663.20 ±571.35 | 69.72 |
| decod_orig[L] | 5 | 16 | **202** | **202.00** ±0 | 0.06 | **202** | **202.00** ±0 | 0.06 | **202** | **202.00** ±0 | 0.06 |
| example2_orig[L] | 85 | 66 | 1790 | 1844.40 ±48.38 | 0.28 | 1784 | **1835.80** ±63.21 | 0.54 | 1894 | 1954.00 ±59.14 | 0.45 |
| frg1_orig[L] | 28 | 3 | **559** | **597.40** ±38.66 | 0.15 | 560 | 610.20 ±41.06 | 0.27 | 567 | 637.40 ±67.22 | 0.17 |
| frg2_orig[L] | 143 | 139 | 7017 | **7601.60** ±506.67 | 1.44 | 9429 | 9429.00 ±0 | 1.69 | **6972** | 7689.80 ±593.33 | 3.08 |
| k2_orig[L] | 45 | 45 | **10156** | **10212.60** ±50.19 | 0.97 | 10200 | 10265.40 ±50.28 | 1.78 | 10276 | 10410.00 ±93.77 | 1.49 |
| lal_orig[L] | 26 | 19 | 496 | 496.00 ±0 | 0.10 | **478** | **490.80** ±11.07 | 0.14 | 484 | 507.80 ±16.90 | 0.12 |
| majority_orig[L] | 5 | 1 | **41** | **41.00** ±0 | 0.06 | **41** | **41.00** ±0 | 0.06 | **41** | **41.00** ±0 | 0.05 |
| mux_orig[L] | 21 | 1 | **170** | **170.00** ±0 | 0.52 | **170** | **170.00** ±0 | 0.39 | **170** | **170.00** ±0 | 0.40 |
| my_adder_orig[L] | 33 | 17 | **352** | **352.00** ±0 | 1.24 | **352** | **352.00** ±0 | 0.66 | **352** | **352.00** ±0 | 0.37 |
| pair_orig[L] | 173 | 137 | **20236** | **22344.20** ±1371.13 | 6.24 | 47110 | 602976.40 ±454972.53 | 8.28 | 20729 | 22438.00 ±1928.73 | 20.16 |
| parity_orig[L] | 16 | 1 | **31** | **31.00** ±0 | 0.09 | **31** | **31.00** ±0 | 0.11 | **31** | **31.00** ±0 | 0.06 |
| pcle_orig[L] | 19 | 9 | **298** | **298.00** ±0 | 0.09 | **298** | **298.00** ±0 | 0.11 | **298** | **298.00** ±0 | 0.10 |
| pcler8_orig[L] | 27 | 17 | 639 | 647.00 ±4 | 0.15 | 674 | 674.00 ±0 | 0.23 | **638** | **638.00** ±0 | 0.16 |
| pm1_orig[L] | 16 | 13 | **244** | 254.40 ±11.27 | 0.08 | **244** | **253.20** ±11.27 | 0.08 | **244** | 261.20 ±14.05 | 0.08 |
| rot_orig[L] | 135 | 107 | **23926** | 26838.00 ±2004.99 | 845.19 | 5928978 | 9069721.00 ±2622975.41 | 62.87 | 24124 | **25464.60** ±1500 | 25.99 |
| s1196_orig[I] | 14 | 14 | **3765** | **3769.40** ±5.39 | 0.14 | **3765** | 3771.60 ±5.39 | 0.18 | **3765** | 3816.80 ±59.21 | 0.29 |
| sct_orig[L] | 19 | 15 | 375 | 386.80 ±12.62 | 0.10 | 373 | **375.80** ±2.79 | 0.12 | 369 | 382.40 ±14.07 | 0.11 |
| tcon_orig[L] | 17 | 16 | **88** | **88.00** ±0 | 0.06 | **88** | **88.00** ±0 | 0.07 | **88** | **88.00** ±0 | 0.06 |
| term1_orig[L] | 34 | 10 | 497 | 500.80 ±3.12 | 0.18 | 473 | **489.60** ±9.33 | 0.33 | 563 | 639.80 ±114.23 | 0.25 |
| too_large_orig[L] | 38 | 3 | 2676 | **2736.40** ±41.96 | 0.71 | **2675** | 2830.80 ±122.28 | 1.83 | 2694 | 2816.60 ±154.97 | 0.90 |
| ttt2_orig[L] | 24 | 21 | 736 | 738.80 ±3.43 | 0.13 | **711** | **727.20** ±9.70 | 0.16 | 727 | 730.20 ±6.40 | 0.15 |
| unreg_orig[L] | 36 | 16 | **494** | 494.40 ±0.49 | 0.12 | 495 | 495.00 ±0 | 0.19 | **494** | **494.00** ±0 | 0.11 |
| vda_orig[L] | 17 | 39 | 4169 | 4181.00 ±9.80 | 0.18 | 4169 | 4177.00 ±9.80 | 0.26 | **4107** | **4176.20** ±50.98 | 0.28 |
| x1_orig[L] | 51 | 35 | 3062 | 3133.60 ±60.90 | 0.50 | 3030 | **3089.80** ±60.19 | 1.06 | 3144 | 3308.40 ±192.40 | 0.76 |
| x2_orig[L] | 10 | 7 | **191** | **191.00** ±0 | 0.06 | **191** | **191.00** ±0 | 0.07 | **191** | **191.00** ±0 | 0.08 |
| x3_orig[L] | 135 | 99 | **3736** | **3787.20** ±36.31 | 0.62 | 4159 | 4368.40 ±169.77 | 0.66 | 4392 | 4551.40 ±148.45 | 1.15 |
| x4_orig[L] | 94 | 71 | **2437** | **2513.60** ±87.43 | 0.37 | 2505 | 2809.40 ±231.03 | 0.65 | 2738 | 2863.20 ±109.57 | 0.69 |

Superscripts denote source dataset: [I] ISCAS85/89; [L] LGSynth91.

Table 19: Full benchmark results, RevLib functions, prior baselines, part 1 of 2. Best QCC compared among QuantumEvo, BDD2Seq, and deterministic CUDD basselines; lowest values bold. BDD2Seq(B*) Time is hardware-adjusted to our platform; BDD2Seq(E*) Time is as originally reported.

| Function | QuantumEvo Best | Time | BDD2Seq(B*) Best | Time | BDD2Seq(E*) Best | Time | SIFT Best | Time | SYMM_SIFT Best | Time | GROUP_SIFT Best | Time |
|---|---|---|---|---|---|---|---|---|---|---|---|---|
| 4gt10_22 | **17** | 0.04 | **17** | 1.49 | **17** | 2.13 | **17** | 0.02 | 23 | 0.01 | 23 | 0.02 |
| 4gt11_23 | **5** | 0.06 | **5** | 0.05 | **5** | 0.01 | **5** | 0.01 | **5** | 0.01 | **5** | 0.03 |
| 4gt12_24 | 17 | 0.06 | 17 | 0.04 | 17 | 0.01 | 17 | 0.01 | **16** | 0.01 | **16** | 0.03 |
| 4gt13_25 | **10** | 0.06 | **10** | 0.05 | **10** | 0.01 | **10** | 0.01 | **10** | 0.01 | **10** | 0.03 |
| 4gt4_20 | **19** | 0.06 | **19** | 0.04 | 25 | 0.01 | **19** | 0.03 | 23 | 0.01 | 23 | 0.03 |
| 4gt5_21 | **12** | 0.05 | **12** | 0.05 | **12** | 0.01 | **12** | 0.03 | 18 | 0.01 | 18 | 0.03 |
| 4mod5_8 | **24** | 0.06 | **24** | 0.05 | **24** | 0.02 | **24** | 0.03 | **24** | 0.01 | **24** | 0.02 |
| 4mod7_26 | 86 | 0.06 | 86 | 0.04 | 86 | 0.01 | 86 | 0.03 | 86 | 0.01 | **80** | 0.03 |
| 5xp1_90 | **254** | 0.06 | **254** | 0.09 | 280 | 0.01 | **254** | 0.03 | **254** | 0.01 | **254** | 0.03 |
| 9symml_91 | **206** | 0.06 | **206** | 0.15 | **206** | 0.03 | **206** | 0.03 | **206** | 0.01 | **206** | 0.03 |
| add6_92 | **118** | 0.07 | **118** | 0.21 | 566 | 0.01 | 499 | 0.03 | 499 | 0.02 | 474 | 0.03 |
| adr4_93 | **74** | 0.05 | **74** | 0.12 | **74** | 0.02 | **74** | 0.03 | **74** | 0.01 | **74** | 0.03 |
| alu1_94 | **139** | 0.06 | **139** | 0.21 | **139** | 0.02 | **139** | 0.03 | **139** | 0.01 | **139** | 0.03 |
| alu2_96 | 1266 | 0.08 | **1218** | 0.17 | 1415 | 0.03 | 1436 | 0.03 | 1436 | 0.01 | 1366 | 0.03 |
| alu3_97 | **430** | 0.06 | **430** | 0.16 | 464 | 0.01 | 644 | 0.03 | 644 | 0.01 | 497 | 0.03 |
| alu4_98 | **4385** | 0.19 | 4455 | 0.33 | 4934 | 0.09 | 7222 | 0.04 | 7222 | 0.02 | 7623 | 0.04 |
| alu_9 | **29** | 0.05 | **29** | 0.06 | **29** | 0.02 | **29** | 0.03 | **29** | 0.01 | 35 | 0.03 |
| apex2_101 | **2663** | 0.62 | 3103 | 1.70 | 3178 | 0.16 | 5922 | 0.11 | 5393 | 0.08 | 6439 | 0.16 |
| apex4_103 | **8236** | 0.13 | 8409 | 0.35 | 8813 | 0.19 | 8343 | 0.02 | 8343 | 0.02 | 8381 | 0.04 |
| apex5_104 | **9184** | 0.97 | 9852 | 12.83 | 10697 | 0.20 | 10358 | 0.06 | 10228 | 0.03 | 10227 | 0.06 |
| apla_107 | **731** | 0.07 | 732 | 0.16 | 955 | 0.03 | 1002 | 0.03 | 1002 | 0.03 | 979 | 0.03 |
| bw_116 | 926 | 0.06 | **924** | 0.06 | 937 | 0.02 | 943 | 0.03 | 943 | 0.03 | 943 | 0.03 |
| C17_117 | **37** | 0.06 | **37** | 0.06 | **37** | 0.01 | 49 | 0.03 | 49 | 0.01 | **37** | 0.03 |
| C7552_119 | **202** | 0.06 | **202** | 0.06 | **202** | 0.01 | **202** | 0.03 | **202** | 0.01 | **202** | 0.03 |
| clip_124 | **515** | 0.06 | 695 | 0.13 | 698 | 0.02 | 704 | 0.03 | 704 | 0.02 | 520 | 0.03 |
| cm150a_128 | 186 | 0.52 | **136** | 0.66 | **136** | 0.24 | 186 | 0.21 | 186 | 0.16 | 186 | 0.20 |
| cm151a_129 | **298** | 0.10 | **298** | 0.43 | **298** | 0.02 | **298** | 0.03 | 310 | 0.02 | **298** | 0.03 |
| cm152a_130 | **62** | 0.07 | **62** | 0.18 | **62** | 0.02 | **62** | 0.03 | **62** | 0.02 | **62** | 0.03 |
| cm163a_133 | **244** | 0.08 | **244** | 0.32 | 267 | 0.02 | 273 | 0.03 | **244** | 0.02 | 267 | 0.03 |
| cm42a_125 | **117** | 0.06 | **117** | 0.05 | **117** | 0.01 | **117** | 0.03 | **117** | 0.02 | **117** | 0.03 |
| cm82a_126 | **44** | 0.05 | **44** | 0.05 | **44** | 0.01 | 82 | 0.03 | **44** | 0.01 | **44** | 0.03 |
| cm85a_127 | **203** | 0.06 | 227 | 0.18 | 283 | 0.03 | 275 | 0.03 | 221 | 0.01 | 221 | 0.03 |
| cmb_134 | **153** | 0.08 | **153** | 0.32 | **153** | 0.02 | 158 | 0.03 | **153** | 0.01 | **153** | 0.03 |
| co14_135 | **159** | 0.08 | **159** | 0.26 | **159** | 0.02 | **159** | 0.02 | **159** | 0.01 | **159** | 0.03 |
| con1_136 | 89 | 0.06 | **88** | 0.10 | 103 | 0.03 | 96 | 0.02 | 96 | 0.02 | 96 | 0.01 |
| cordic_138 | **304** | 0.16 | 344 | 0.62 | 446 | 0.05 | 325 | 0.02 | 323 | 0.02 | 318 | 0.04 |
| cu_141 | **219** | 0.08 | **219** | 0.26 | 230 | 0.02 | 220 | 0.01 | 220 | 0.01 | 224 | 0.02 |
| dc1_142 | 186 | 0.06 | **160** | 0.05 | 168 | 0.01 | **160** | 0.03 | **160** | 0.03 | 186 | 0.03 |
| dc2_143 | **431** | 0.06 | **431** | 0.11 | **431** | 0.03 | **431** | 0.03 | **431** | 0.02 | **431** | 0.03 |
| decod24-enable_32 | **38** | 0.06 | **38** | 0.04 | **38** | 0.01 | **38** | 0.03 | **38** | 0.03 | **38** | 0.03 |
| decod_137 | **202** | 0.05 | **202** | 0.06 | **202** | 0.01 | **202** | 0.03 | **202** | 0.03 | **202** | 0.03 |
| dist_144 | **975** | 0.06 | **975** | 0.13 | 979 | 0.04 | **975** | 0.03 | **975** | 0.03 | 979 | 0.03 |
| dk17_145 | **426** | 0.06 | 429 | 0.16 | 429 | 0.03 | **426** | 0.03 | **426** | 0.03 | 574 | 0.03 |
| dk27_146 | 140 | 0.06 | **131** | 0.13 | **131** | 0.02 | 140 | 0.03 | 140 | 0.03 | 141 | 0.03 |
| e64_149 | **886** | 0.39 | 1019 | 4.17 | 1208 | 0.17 | 907 | 0.05 | 907 | 0.05 | 1124 | 0.04 |
| ex1010_155 | 9681 | 0.23 | **9633** | 0.33 | 9810 | 0.14 | 9766 | 0.07 | 9766 | 0.04 | 9696 | 0.05 |
| ex1_150 | **8** | 0.06 | **8** | 0.07 | **8** | 0.01 | **8** | 0.03 | **8** | 0.01 | **8** | 0.03 |
| ex2_151 | **60** | 0.06 | **60** | 0.07 | 68 | 0.01 | 73 | 0.03 | 70 | 0.03 | **60** | 0.03 |

Table 20: Full benchmark results, RevLib functions, prior baselines, part 2 of 2.

| Function | QuantumEvo | | BDD2Seq(B*) | | BDD2Seq(E*) | | SIFT | | SYMM_SIFT | | GROUP_SIFT | |
|---|---|---|---|---|---|---|---|---|---|---|---|---|
| | Best | Time | Best | Time | Best | Time | Best | Time | Best | Time | Best | Time |
| ex3_152 | **38** | 0.06 | **38** | 0.08 | **38** | 0.02 | 61 | 0.03 | 61 | 0.03 | 43 | 0.03 |
| ex5p_154 | 1843 | 0.11 | **1837** | 0.17 | 1871 | 0.06 | 1843 | 0.04 | 1843 | 0.05 | 1843 | 0.05 |
| example2_156 | 1266 | 0.08 | **1218** | 0.18 | 1415 | 0.03 | 1436 | 0.03 | 1436 | 0.03 | 1366 | 0.02 |
| f2_158 | **108** | 0.05 | **108** | 0.05 | 113 | 0.01 | 113 | 0.03 | 113 | 0.03 | 113 | 0.02 |
| f51m_159 | **2422** | 0.14 | 2427 | 0.46 | 2517 | 0.14 | 5392 | 0.05 | 5392 | 0.05 | 4603 | 0.05 |
| frg1_160 | **559** | 0.17 | 598 | 0.95 | 629 | 0.03 | 747 | 0.03 | 747 | 0.03 | 827 | 0.03 |
| frg2_161 | **7017** | 1.47 | 8584 | 18.95 | 10768 | 0.37 | 12468 | 0.07 | 12361 | 0.08 | 12224 | 0.09 |
| in0_162 | 2299 | 0.11 | 2296 | 0.31 | 2417 | 0.04 | **2283** | 0.01 | **2283** | 0.03 | 2328 | 0.04 |
| inc_170 | 592 | 0.06 | **579** | 0.09 | **579** | 0.01 | **579** | 0.01 | **579** | 0.03 | 592 | 0.03 |
| life_175 | **204** | 0.06 | **204** | 0.14 | 210 | 0.01 | **204** | 0.01 | 210 | 0.01 | **204** | 0.03 |
| majority_176 | **41** | 0.06 | **41** | 0.06 | **41** | 0.01 | **41** | 0.01 | **41** | 0.02 | **41** | 0.03 |
| max46_177 | **556** | 0.07 | 562 | 0.14 | 596 | 0.02 | 598 | 0.01 | 598 | 0.03 | 598 | 0.03 |
| misex1_178 | 279 | 0.06 | 281 | 0.12 | 289 | 0.02 | 288 | 0.01 | 288 | 0.03 | 289 | 0.03 |
| misex3_180 | **3789** | 0.16 | 3830 | 0.33 | 4054 | 0.09 | 4661 | 0.03 | 4661 | 0.05 | 4619 | 0.05 |
| misex3c_181 | **3936** | 0.16 | 3969 | 0.29 | 4457 | 0.05 | 4769 | 0.02 | 4769 | 0.04 | 4823 | 0.04 |
| mlp4_184 | **1158** | 0.07 | 1191 | 0.13 | 1228 | 0.03 | **1158** | 0.02 | **1158** | 0.03 | **1158** | 0.03 |
| mux_185 | 170 | 0.55 | **135** | 0.90 | 224 | 0.26 | 170 | 0.23 | 170 | 0.25 | 170 | 0.35 |
| one-two-three_27 | 44 | 0.06 | 44 | 0.04 | 44 | 0.01 | 44 | 0.01 | 44 | 0.03 | 44 | 0.01 |
| parity_188 | 31 | 0.50 | 31 | 0.82 | 31 | 0.44 | 31 | 0.18 | 31 | 0.21 | 31 | 0.17 |
| pcler8_190 | **124** | 0.08 | **124** | 0.32 | 137 | 0.01 | **124** | 0.01 | **124** | 0.03 | 137 | 0.01 |
| pdc_191 | 6599 | 0.41 | 6742 | 0.57 | 6742 | 0.23 | **6500** | 0.13 | **6500** | 0.19 | **6500** | 0.13 |
| pm1_192 | **117** | 0.06 | **117** | 0.05 | **117** | 0.02 | **117** | 0.01 | **117** | 0.01 | **117** | 0.01 |
| radd_193 | **74** | 0.06 | **74** | 0.11 | 140 | 0.01 | 217 | 0.01 | 192 | 0.03 | 212 | 0.01 |
| rd53_68 | **98** | 0.05 | **98** | 0.05 | **98** | 0.01 | **98** | 0.01 | **98** | 0.03 | **98** | 0.01 |
| rd73_69 | **217** | 0.06 | **217** | 0.09 | **217** | 0.02 | **217** | 0.01 | **217** | 0.03 | **217** | 0.01 |
| rd84_70 | **304** | 0.06 | **304** | 0.12 | **304** | 0.02 | **304** | 0.01 | **304** | 0.03 | **304** | 0.01 |
| root_197 | **444** | 0.06 | **444** | 0.11 | 446 | 0.01 | **444** | 0.01 | **444** | 0.03 | **444** | 0.01 |
| ryy6_198 | 107 | 0.09 | **103** | 0.33 | 107 | 0.02 | 133 | 0.02 | 132 | 0.03 | 119 | 0.03 |
| sao2_199 | **653** | 0.07 | 657 | 0.16 | 698 | 0.01 | 667 | 0.01 | 667 | 0.03 | 684 | 0.03 |
| seq_201 | **9080** | 1.17 | 9349 | 2.12 | 9908 | 0.33 | 19362 | 0.24 | 19350 | 0.28 | 15309 | 0.27 |
| sf_232 | **21** | 0.06 | 31 | 0.05 | 36 | 0.02 | 36 | 0.01 | 36 | 0.03 | **21** | 0.03 |
| spla_202 | **5858** | 0.33 | 5947 | 0.52 | 6092 | 0.19 | 5925 | 0.08 | 5925 | 0.11 | 5925 | 0.10 |
| sqn_203 | 356 | 0.06 | 374 | 0.10 | 426 | 0.02 | 426 | 0.01 | 426 | 0.03 | 426 | 0.03 |
| sqr6_204 | 486 | 0.06 | 470 | 0.08 | 524 | 0.02 | 486 | 0.01 | 486 | 0.03 | 486 | 0.03 |
| sqrt8_205 | **221** | 0.06 | 237 | 0.11 | 283 | 0.01 | 240 | 0.01 | 240 | 0.03 | 240 | 0.03 |
| squar5_206 | **232** | 0.06 | **232** | 0.07 | 264 | 0.02 | 253 | 0.01 | 253 | 0.03 | **232** | 0.03 |
| sym10_207 | **253** | 0.07 | **253** | 0.18 | **253** | 0.03 | **253** | 0.02 | **253** | 0.03 | **253** | 0.03 |
| sym6_63 | **93** | 0.06 | **93** | 0.08 | **93** | 0.01 | **93** | 0.01 | **93** | 0.03 | **93** | 0.03 |
| sym9_71 | **206** | 0.06 | **206** | 0.15 | **206** | 0.03 | **206** | 0.01 | **206** | 0.03 | **206** | 0.03 |
| t481_208 | 147 | 0.10 | **139** | 0.34 | 140 | 0.03 | 147 | 0.02 | 152 | 0.03 | 152 | 0.03 |
| table3_209 | **5898** | 0.18 | 5954 | 0.36 | 7005 | 0.08 | 6276 | 0.02 | 6276 | 0.04 | 5927 | 0.03 |
| tial_214 | **4460** | 0.13 | 4475 | 0.33 | 5157 | 0.09 | 7609 | 0.02 | 7609 | 0.04 | 4852 | 0.04 |
| urf4_89 | **28406** | 0.41 | 28488 | 0.68 | 28833 | 0.47 | 28523 | 0.06 | 28523 | 0.10 | 28523 | 0.11 |
| wim_220 | 107 | 0.06 | **103** | 0.05 | 108 | 0.01 | 107 | 0.01 | 107 | 0.03 | 107 | 0.03 |
| x2_223 | **191** | 0.06 | **191** | 0.17 | **191** | 0.03 | 273 | 0.01 | 273 | 0.03 | 283 | 0.03 |
| xor5_195 | **8** | 0.05 | **8** | 0.05 | **8** | 0.01 | **8** | 0.01 | **8** | 0.03 | **8** | 0.03 |
| z4_224 | **66** | 0.06 | **66** | 0.09 | **66** | 0.01 | **66** | 0.01 | **66** | 0.03 | **66** | 0.03 |
| z4ml_225 | **66** | 0.05 | **66** | 0.10 | **66** | 0.02 | **66** | 0.01 | **66** | 0.03 | **66** | 0.03 |

Table 21: Full benchmark results, ISCAS85/89 and LGSynth91 functions, prior baselines.

| Function | QuantumEvo | | BDD2Seq(B*) | | BDD2Seq(E*) | | SIFT | | SYMM_SIFT | | GROUP_SIFT | |
|---|---|---|---|---|---|---|---|---|---|---|---|---|
| | Best | Time | Best | Time | Best | Time | Best | Time | Best | Time | Best | Time |
| apex6_orig[L] | **3736** | 0.65 | 3895 | 16.72 | 4021 | 0.16 | 4822 | 0.05 | 4810 | 0.03 | 4269 | 0.03 |
| apex7_orig[L] | **1431** | 0.28 | 1455 | 2.40 | 1456 | 0.07 | 2653 | 0.04 | 2653 | 0.04 | 2377 | 0.04 |
| b1_orig[L] | **22** | 0.06 | **22** | 0.04 | **22** | 0.01 | **22** | 0.03 | **22** | 0.03 | **22** | 0.03 |
| b9_orig[L] | **713** | 0.17 | 737 | 1.71 | 737 | 0.05 | 804 | 0.03 | 810 | 0.03 | 836 | 0.03 |
| c17[I] | **37** | 0.06 | **37** | 0.06 | 49 | 0.01 | 49 | 0.03 | 49 | 0.03 | **37** | 0.03 |
| c432[I] | **10494** | 245.19 | 12255 | 10.01 | 20391 | 6.79 | 11101 | 0.04 | 11101 | 0.02 | 11141 | 0.04 |
| c880[I] | **34241** | 329.32 | 37802 | 23.00 | 39992 | 15.52 | 61079 | 0.53 | 61079 | 0.65 | 78113 | 0.59 |
| c8_orig[L] | 436 | 0.14 | **432** | 0.86 | 436 | 0.04 | 445 | 0.03 | 445 | 0.02 | 486 | 0.03 |
| cc_orig[L] | 283 | 0.08 | **272** | 0.51 | 278 | 0.03 | 385 | 0.03 | 385 | 0.01 | 379 | 0.03 |
| cht_orig[L] | **714** | 0.12 | **714** | 2.22 | **714** | 0.06 | **714** | 0.03 | **714** | 0.02 | **714** | 0.03 |
| cm138a_orig[L] | **104** | 0.06 | **104** | 0.08 | **104** | 0.01 | **104** | 0.02 | **104** | 0.02 | **104** | 0.03 |
| cm150a_orig[L] | 186 | 0.52 | **136** | 0.75 | **136** | 0.34 | 186 | 0.38 | 186 | 0.25 | 186 | 0.32 |
| cm151a_orig[L] | 92 | 0.07 | **70** | 0.21 | **70** | 0.02 | 92 | 0.03 | 92 | 0.02 | 92 | 0.03 |
| cm152a_orig[L] | **62** | 0.07 | **62** | 0.18 | 81 | 0.02 | **62** | 0.03 | **62** | 0.02 | **62** | 0.03 |
| cm162a_orig[L] | 222 | 0.08 | 222 | 0.26 | 222 | 0.03 | **192** | 0.03 | 258 | 0.02 | 244 | 0.03 |
| cm163a_orig[L] | **124** | 0.08 | 136 | 0.32 | 136 | 0.03 | **124** | 0.03 | **124** | 0.02 | 137 | 0.03 |
| cm42a_orig[L] | **117** | 0.05 | **117** | 0.04 | **117** | 0.02 | **117** | 0.03 | **117** | 0.01 | **117** | 0.03 |
| cm82a_orig[L] | **44** | 0.06 | **44** | 0.05 | **44** | 0.01 | 82 | 0.03 | **44** | 0.01 | **44** | 0.03 |
| cm85a_orig[L] | **203** | 0.07 | 225 | 0.18 | 225 | 0.02 | 275 | 0.03 | 221 | 0.01 | 221 | 0.03 |
| cmb_orig[L] | **153** | 0.09 | **153** | 0.32 | **153** | 0.03 | 158 | 0.03 | **153** | 0.01 | **153** | 0.03 |
| comp_orig[L] | **796** | 2.41 | 824 | 40.21 | 824 | 28.75 | 961 | 1.77 | 1071 | 1.96 | 884 | 1.79 |
| cordic_orig[L] | **304** | 0.12 | 320 | 0.62 | 320 | 0.06 | 325 | 0.01 | 323 | 0.01 | 318 | 0.03 |
| count_orig[L] | **441** | 0.17 | **441** | 1.27 | **441** | 0.04 | **441** | 0.01 | **441** | 0.01 | **441** | 0.02 |
| cu_orig[L] | **219** | 0.08 | 224 | 0.26 | 231 | 0.02 | 220 | 0.01 | 220 | 0.01 | 224 | 0.03 |
| dalu_orig[L] | **5230** | 71.73 | 5297 | 55.53 | 5652 | 28.17 | 31145 | 66.13 | 31145 | 71.95 | 47170 | 75.07 |
| decod_orig[L] | **202** | 0.06 | **202** | 0.05 | **202** | 0.01 | **202** | 0.03 | **202** | 0.03 | **202** | 0.03 |
| example2_orig[L] | **1790** | 0.28 | 1800 | 6.83 | 1806 | 0.08 | 2066 | 0.03 | 2066 | 0.02 | 2041 | 0.03 |
| frg1_orig[L] | **559** | 0.15 | 757 | 0.95 | 762 | 0.03 | 747 | 0.03 | 747 | 0.02 | 827 | 0.02 |
| frg2_orig[L] | **7017** | 1.44 | 7189 | 19.35 | 7189 | 0.35 | 12468 | 0.03 | 12361 | 0.06 | 12224 | 0.07 |
| k2_orig[L] | **10156** | 0.97 | 10432 | 2.38 | 10686 | 0.19 | 11058 | 0.07 | 11058 | 0.13 | 11275 | 0.09 |
| lal_orig[L] | 496 | 0.10 | **478** | 0.76 | 496 | 0.04 | 527 | 0.01 | 768 | 0.02 | 724 | 0.03 |
| majority_orig[L] | **41** | 0.06 | **41** | 0.05 | **41** | 0.01 | **41** | 0.01 | **41** | 0.03 | **41** | 0.03 |
| mux_orig[L] | 170 | 0.52 | **135** | 0.84 | **135** | 0.40 | 170 | 0.24 | 170 | 0.35 | 170 | 0.26 |
| my_adder_orig[L] | **352** | 1.24 | **352** | 59.50 | **352** | 48.54 | **352** | 0.15 | **352** | 0.35 | **352** | 0.16 |
| pair_orig[L] | **20236** | 6.24 | 20754 | 28.19 | 20799 | 1.06 | 46491 | 0.11 | 46391 | 0.17 | 49917 | 0.15 |
| parity_orig[L] | **31** | 0.09 | **31** | 0.83 | **31** | 0.61 | **31** | 0.01 | **31** | 0.03 | **31** | 0.01 |
| pcle_orig[L] | **298** | 0.09 | **298** | 0.43 | **298** | 0.02 | **298** | 0.01 | 310 | 0.03 | **298** | 0.01 |
| pcler8_orig[L] | 639 | 0.15 | 765 | 0.80 | 765 | 0.02 | **638** | 0.01 | **638** | 0.04 | 639 | 0.01 |
| pm1_orig[L] | 244 | 0.08 | **234** | 0.33 | 273 | 0.03 | 273 | 0.01 | 244 | 0.01 | 267 | 0.01 |
| rot_orig[L] | **23926** | 845.19 | 38453 | 37.98 | 48344 | 15.69 | 78639 | 0.56 | 78374 | 0.62 | 110936 | 0.74 |
| s1196_orig[I] | **3765** | 0.14 | 4104 | 0.30 | 4110 | 0.05 | 4691 | 0.02 | 4691 | 0.04 | 4653 | 0.04 |
| sct_orig[L] | **375** | 0.10 | 405 | 0.43 | 513 | 0.02 | 545 | 0.01 | 545 | 0.03 | 548 | 0.03 |
| tcon_orig[L] | **88** | 0.06 | **88** | 0.36 | **88** | 0.02 | **88** | 0.01 | **88** | 0.03 | **88** | 0.02 |
| term1_orig[L] | 497 | 0.18 | **480** | 1.25 | 483 | 0.06 | 1193 | 0.02 | 1151 | 0.04 | 1070 | 0.04 |
| too_large_orig[L] | **2676** | 0.71 | 3744 | 2.19 | 5379 | 0.46 | 5922 | 0.07 | 5393 | 0.12 | 6334 | 0.10 |
| ttt2_orig[L] | 736 | 0.13 | 749 | 0.66 | 1006 | 0.04 | **727** | 0.02 | **727** | 0.04 | 1139 | 0.03 |
| unreg_orig[L] | **494** | 0.12 | **494** | 1.36 | **494** | 0.04 | **494** | 0.01 | **494** | 0.03 | **494** | 0.03 |
| vda_orig[L] | **4169** | 0.16 | 4224 | 0.42 | 4224 | 0.08 | 4477 | 0.03 | 4477 | 0.05 | 4460 | 0.05 |
| x1_orig[L] | **3062** | 0.50 | 3567 | 2.68 | 3703 | 0.16 | 3678 | 0.02 | 4189 | 0.04 | 4172 | 0.05 |
| x2_orig[L] | **191** | 0.06 | **191** | 0.15 | **191** | 0.02 | 273 | 0.01 | 273 | 0.03 | 283 | 0.03 |
| x3_orig[L] | **3736** | 0.62 | 3941 | 16.78 | 4645 | 0.19 | 4822 | 0.03 | 4810 | 0.05 | 4269 | 0.03 |
| x4_orig[L] | **2437** | 0.37 | 2662 | 8.31 | 2782 | 0.11 | 4470 | 0.02 | 4470 | 0.04 | 4334 | 0.04 |

Superscripts denote source dataset: [I] ISCAS85/89; [L] LGSynth91.

