# OpenReview forum: "LLM-Driven Algorithm Design for Quantum Circuit Synthesis based on Binary Decision Diagrams"
_TMLR — Under review for TMLR_

### Review · Reviewer_trH9 · 2026-07-05

**Summary Of Contributions:**

This paper studies BDD variable ordering for reversible circuit synthesis. The authors argue that the commonly used BDD size objective is only an indirect proxy for the quantum cost of the synthesized circuit (QCC). They propose QuantumEvo, an LLM-driven evolutionary framework that generates executable CUDD-based ordering heuristics and evaluates them by downstream QCC after reversible circuit synthesis. The best discovered heuristic, HGA-QE, modifies the sifting component inside a genetic algorithm through PartialSift/MiniSift. Experiments on RevLib, LGSynth91, and ISCAS85/89 benchmarks show that HGA-QE achieves the best tie-or-win rate and competitive mean QCC compared with classical CUDD ordering methods, GA/SA, and BDD2Seq.

**Audience:**

Yes

**Audience Explanation:**

Some TMLR readers interested in LLM-based algorithm design, automated heuristic discovery, and optimization inside specialized scientific or engineering pipelines may find the paper relevant. The paper is also potentially interesting for readers working on quantum compilation or reversible circuit synthesis, because it highlights a concrete mismatch between BDD size and downstream QCC.

However, the audience may be relatively narrow. The current manuscript does not sufficiently explain why QCC-aware BDD variable ordering is a central or high-impact bottleneck in practical quantum computing workflows. The paper would be more compelling if it connected the benchmark improvements to downstream use cases such as oracle construction, arithmetic subroutines, or larger quantum compilation pipelines.

**Broader Impact Concerns:**

I do not see major ethical concerns requiring a separate broader impact statement.

**Claims And Evidence:**

No

**Claims Explanation:**

First, the improvement over strong baselines is moderate in mean QCC, and the strict-best rate is relatively limited. For example, on RevLib, BDD2Seq has more strict wins than HGA-QE. Second, the computational cost is not analyzed sufficiently. QCC-aware evaluation requires full synthesis, and some reported runtimes are large. A method that improves QCC moderately but incurs much higher evaluation or search cost needs a clearer cost-quality tradeoff analysis. Third, the reliability of the discovered heuristic is not fully established. The paper reports multiple runs for some settings, but it would benefit from confidence intervals, statistical tests on final benchmark performance, failure-case analysis, and equal-budget comparisons. Finally, because the framework resembles existing LLM-based program evolution methods, the paper should more clearly separate what is inherited from prior systems and what is new in this work.

**Requested Changes:**

Critical changes:

1. Clarify the novelty relative to existing LLM-based algorithm discovery systems. The paper should explicitly compare QuantumEvo with AlphaEvolve/FunSearch/ReEvo-style frameworks and state which components are inherited and which are new. At present, the framework contribution appears mostly to be domain adaptation.

2. Provide a more complete cost-quality analysis. Since QCC-based evaluation requires full circuit synthesis, the paper should report total search cost, number of synthesis calls, LLM cost, runtime distributions, timeout behavior, and anytime performance. Equal wall-clock or equal evaluation-budget comparisons against GA, SA, and other baselines would make the empirical conclusions more convincing.

3. Improve reliability analysis. The final benchmark results should include variance or confidence intervals across seeds where applicable, statistical tests for the main comparisons, and a clearer discussion of cases where HGA-QE does not improve over baselines. The paper should also analyze sensitivity to the search set, validation set, timeout, and LLM choice.

4. Add stronger ablations for the algorithmic contribution. The paper should isolate the effects of PartialSift, MiniSift, local refinement, mixed initialization, and QCC-based fitness under comparable budgets. A non-LLM evolutionary or random-mutation baseline using the same candidate representation would help determine whether the LLM is essential.


Changes that would strengthen the work:
1. Report quality-runtime Pareto curves rather than only final QCC values.

2. Provide more interpretable examples showing how MiniSift changes BDD structure in cases where BDD size is unchanged but QCC improves.

3. Discuss whether HGA-QE can be used as a standalone CUDD reorderer and whether the discovered heuristic transfers to other BDD-based synthesis or logic optimization tasks.

---

> ### Author Response · Authors · 2026-07-20
> **Response to Reviewer 1 (Required Changes #1, #2)**
>
> We thank the reviewer for the thorough and constructive feedback. We addressed every point with new analyses and revised text, summarized below with the key numbers. Full details are in the referenced sections/appendices of the revised manuscript.
>
> ## Required Changes
>
> ### 1. Clarified the novelty of QuantumEvo relative to existing LLM-based algorithm discovery frameworks (Addressed in: Introduction, Section 3.2)
> > Clarify the novelty relative to existing LLM-based algorithm discovery systems. ... At present, the framework contribution appears mostly to be domain adaptation.
>
> We understood this comment as asking us to state explicitly what QuantumEvo inherits from prior LLM-based algorithm-design frameworks and what is specific to our setting, rather than leaving the relationship implicit. We revised Section 3.2 to state that QuantumEvo's search procedure builds primarily on ReEvo's reflective evolve-and-refine loop. Accordingly, we do not position the framework itself as an entirely new search paradigm. Instead, our contribution lies in adapting this framework to BDD variable ordering through the choice of representation, multi-family initialization, and QCC-based evaluation, and in the resulting discovery of HGA-QE, a new hybrid GA heuristic for this domain. We also moderated the corresponding contribution claim in the Introduction to avoid overstating the framework-level novelty.
>
>
> ### 2. Added a cost–quality analysis (Adressed in: Appendix E, Section 4.2.2)
> > Provide a more complete cost-quality analysis. ... Equal wall-clock or equal evaluation-budget comparisons ... would make the empirical conclusions more convincing.
>
> We addressed each sub-point:
> - We added Appendix E, "Offline and Online Cost," which separates the one-time cost of discovering HGA-QE from the recurring cost of applying it. Averaged over the 10 independent production searches with gpt-oss-120b, one search used 291.6 LLM calls, 3.11M input tokens, and 1.04M output tokens (about 4.15M tokens total), over 108.5 minutes of wall-clock time on a single self-hosted NVIDIA RTX PRO 6000. Each candidate reaching evaluation is compiled and run on the 100 function search set with the synthesis pipeline invoked twice per function, so a full run (up to 200 candidate evaluations) corresponds to up to ~40,000 synthesis calls. For the runtime distributions of generated candidates, 8.3% of candidate evaluations hit the 40 s per-instance timeout, and completed evaluations took a mean of 9.3 s, median 7.0 s, and 95th-percentile 26.0 s.
> - For an equal-budget, anytime-style comparison, we added a quality--runtime Pareto analysis in Section 4.2.2 (Figure 5). For GA, SA, and HGA-QE, we plot the best-so-far relative QCC gap against cumulative wall-clock runtime as the run budget increases from one to five runs. Both axes are normalized per circuit and aggregated across all 148 benchmark circuits using the geometric mean, which is appropriate for combining normalized ratios across heterogeneous instances and avoids dependence on the choice of normalization baseline [1]. Deterministic baselines and BDD2Seq are shown as single points. HGA-QE forms the higher-runtime, lower-gap segment of the Pareto frontier.
>
> [1] Fleming, P. J., & Wallace, J. J. (1986). How not to lie with statistics: the correct way to summarize benchmark results. Communications of the ACM, 29(3), 218-221.

---

> ### Author Response · Authors · 2026-07-20
> **Response to Reviewer 1 (Required Changes #3)**
>
> ### 3. Expanded reliability and sensitivity analysis (Addressed in: Appendix G, Appendix D.3, Appendix C.4, Appendix C.1)
>
> > Improve reliability analysis. ... variance or confidence intervals ..., statistical tests ..., a clearer discussion of cases where HGA-QE does not improve ..., analyze sensitivity to the search set, validation set, timeout, and LLM choice.
>
> We addressed each sub-point:
> - Variance across seeds: Appendix G now reports, for every stochastic method on all 148 benchmark functions, average QCC with standard deviation.
> - Statistical tests on the main comparison: Appendix D.3.2 reports paired Wilcoxon signed-rank tests (Holm–Bonferroni corrected across the 6 baselines) between HGA-QE and each baseline, per suite and overall. After correction, across all 148 functions HGA-QE is significantly better than every baseline. The suite-level picture is less uniform, which we now state directly. On RevLib, HGA-QE is significantly better than the three sifting variants and BDD2Seq, but *not* significantly different from GA or SA. On LGSynth91, it is significant against every baseline except GA. On ISCAS85/89, no comparison reaches significance, which we attribute to insufficient statistical power rather than equal performance.
>
>     | Suite | $n$ | Mean $\Delta$QCC vs. SA | $p$ (Holm) | Mean $\Delta$QCC vs. GA | $p$ (Holm) | Mean $\Delta$QCC vs. BDD2Seq | $p$ (Holm) |
>     |---|---:|---:|---:|---:|---:|---:|---:|
>     | RevLib | 96 | $-0.21$ | 0.571 | $-29.24$ | 0.139 | $-39.99$ | **0.025** |
>     | LGSynth91 | 48 | $-65.04$ | **0.006** | $-123651.40$ | 0.055 | $-377.46$ | **0.002** |
>     | ISCAS85/89 | 4 | $-151.75$ | 0.653 | $-161.75$ | 0.653 | $-1415.25$ | 0.653 |
>     | All | 148 | $-25.33$ | **0.008** | $-40126.49$ | **0.008** | $-186.61$ | **$1.2{\times}10^{-5}$** |
>
>
> - Failure-case discussion: In Appendix D.3.2, Table 13 reports that HGA-QE is not (tied-)best on 43/148 circuits (29.1%), which is the *lowest* loss rate of any method on every suite (SA: 55/148 (37.2%), BDD2Seq: 60/148 (40.5%), GA: 64/148 (43.2%)). As shown in the table below, among HGA-QE's 43 losses, 21 are attributed to BDD2Seq and 11 to SA.
>     | Suite | HGA-QE losses | BDD2Seq | SA | GA | Others |
>     |---|---:|---:|---:|---:|---:|
>     | RevLib | 26/96 | 15 | 7 | 1 | 3 |
>     | LGSynth91 | 17/48 | 6 | 4 | 6 | 1 |
>     | ISCAS85/89 | 0/4 | 0 | 0 | 0 | 0 |
>     | Overall | 43/148 | 21 | 11 | 7 | 4 |
>
> - Sensitivity to search set, validation set, timeout: Appendix C.4 examines the robustness of the discovery process to three experimental choices. First, we reevaluate the 30 leading candidates from the run that produced HGA-QE on the original and nine alternative 100 function search sets. HGA-QE ranks first on 5 of the 10 sets, including the original. Second, we reselect the finalist using the original and nine alternative 50 function validation sets. The full QuantumEvo setting selects the same finalist in 9 of the 10 cases. Finally, replaying the recorded candidate-evaluation times under shorter time limits shows a gradual increase in the failure rate, from 8.3% at the original 40 s timeout to 17.7% at 20 s.
> - Sensitivity to LLM choice: We had already compared gpt-oss-120b and gpt-5.4-mini in Appendix C.1. The runs using gpt-oss-120b achieve stronger validation performance, with a higher win-or-tie rate and lower mean QCC. The two models also tend to produce different heuristic families. gpt-oss-120b selects SA-family individuals in 9 of 10 runs, whereas gpt-5.4-mini more often converges to sifting-family individuals.

---

> ### Author Response · Authors · 2026-07-20
> **Response to Reviewer 1 (Required Changes #4)**
>
> ### 4. Added stronger ablations on the algorithmic components (Addressed in: Section 3.3 - Table 1, Section 4.2.3, Appendix C.1, Appendix C.3)
>
> > Add stronger ablations for the algorithmic contribution. ... A non-LLM evolutionary or random-mutation baseline using the same candidate representation would help determine whether the LLM is essential.
>
> We separate the request into two levels: PartialSift/MiniSift/local refinement are components of the discovered heuristic HGA-QE, while QCC-based fitness, mixed initialization, and the LLM itself are design choices of the QuantumEvo framework.
>
> - PartialSift/MiniSift/local refinement: In section 3.3, Table 1 already shows that relative to the original GA with full sifting, replacing sifting with MiniSift reduces the mean QCC gap from 2.096% to 0.532% while also slightly reducing BDD size. The additional local-refinement step gives only a marginal further change from 0.532% to 0.527%. PartialSift is embedded inside MiniSift's targeted local-sifting mechanism and is not evaluated as a fully independent module.
>     | Method | QCC Mean Gap ($\downarrow$) | W/T/L | BDD Mean Gap ($\downarrow$) | Total Time (s) |
>     |---|---:|---:|---:|---:|
>     | GA + sifting | 2.096% | — | $-0.211\%$ | 23.24 |
>     | GA + MiniSift | 0.532% | 34/53/13 | $-0.350\%$ | 21.95 |
>     | GA + MiniSift + local refinement | 0.527% | 34/53/13 | $-0.354\%$ | 22.01 |
>
> - QCC-based vs. BDD-size fitness, and multi-family vs. single-family initialization: Section 4.2.3 compares these settings using the validation performance of the finalists from ten independent runs and the subsequent full-benchmark evaluation of the best-validation finalist selected for each setting. The results support the specific design choices used in QuantumEvo: the multi-family, QCC-based setting yields the strongest full-benchmark performance among the evaluated configurations, although the validation results do not uniformly favor every component. In particular, GA-only initialization achieves marginally better validation fitness in some comparisons, showing that the benefit of multi-family initialization is more apparent in benchmark generalization than in validation performance alone. However, the validation results do not uniformly favor this setting. In particular, GA-only initialization achieves marginally better validation fitness in some comparisons.
>
> - Non-LLM evolutionary search: We construct a non-LLM evolutionary search that retains QuantumEvo's evolutionary loop, fitness function, initialization scheme, and search/validation split, but replaces LLM-based code generation with search over a predefined library of sifting, GA, and SA components, as described in Appendix C.1. Under the matched search budget, the non-LLM control achieves stronger validation performance. On the full 148-function benchmark, however, the HGA-QE finalist achieves higher Best and Strictly Best rates and lower mean-QCC statistics.
>
>     | Guidance | Best/Total (%) | Strictly Best/Total (%) | Mean Best QCC | Mean Avg QCC | Mean Worst QCC |
>     |---|---:|---:|---:|---:|---:|
>     | **LLM** | **70.9** | **13.5** | **1923.83** | **1979.76** | **2033.46** |
>     | Non-LLM | 66.9 | 5.4 | 2075.15 | 2220.27 | 2432.07 |
>
>     As stated in the manuscript, this result does not establish that LLM guidance is uniformly superior. Rather, they show that the LLM-guided search can discover algorithmic variants beyond the predefined component library and, in this study, produced an HGA-QE finalist that transferred more effectively to the full benchmark.
>
> We recalibrated the interpretation of the experimental results throughout the manuscript to more accurately reflect the empirical evidence. We also added a complementary analysis of the top 5 and top 10 elite candidate pools for each setting in Appendix C.3.

---

> ### Author Response · Authors · 2026-07-20
> **Response to Reviewer 1 (Recommended Changes #1, #2, #3)**
>
> ## Recommended Changes
>
> ### 1. Quality–runtime Pareto curves (Addressed in: Section 4.2.2.)
> > Report quality-runtime Pareto curves rather than only final QCC values.
>
> This curve has been added as Figure 5 in Section 4.2.2 and is described above under Required Change 2.
>
> ### 2. Interpretable example of MiniSift changing BDD structure at equal BDD size (Addressed in: Appendix D.2)
> > Provide more interpretable examples showing how MiniSift changes BDD structure in cases where BDD size is unchanged but QCC improves.
>
> Appendix D.2 adds a worked example on a four-input, single-output augmented RevLib function. Standard sifting orders variables as $(x_0,x_3,x_1,x_2)$ and MiniSift as $(x_3,x_0,x_1,x_2)$. both give BDD size 7, but QCC drops from 40 to 30. Swapping only the top two variables changes which node a shared, twice-referenced child is attached to, eliminating one "child-node line preservation" case in the synthesis trace and removing 2 Toffoli gates and 5 controls. Across all 31 same-size instances, the same trend is observed, with statistically significant reductions in Toffoli gates, total controls, and QCC (Wilcoxon $p<10^{-4}$).
>
>
> ### 3. Standalone use of HGA-QE and transfer to other tasks (Addressed in: Appendix D.4, Conclusion.)
> > Discuss whether HGA-QE can be used as a standalone CUDD reorderer and whether the discovered heuristic transfers to other BDD-based synthesis or logic optimization tasks.
>
> HGA-QE requires no LLM calls or QCC evaluation at deployment time. It is a fixed compiled `C` function that operates on a CUDD manager using only BDD-level API calls, so it can be used as a standalone CUDD reorderer in the same way as sifting/GA/SA. To test transfer to a different downstream task, we evaluated HGA-QE in an FPGA logic-synthesis flow and reported the results in Appendix D.4. Each ordered BDD is exported as a multiplexer network and mapped to 4-input LUTs with Yosys, on 52 functions from LGSynth91/ISCAS85/89. The results are as shown in the table below. Although HGA-QE remains competitive for network size, its advantage does not consistently transfer to the further downstream LUT-count objective.
>
> | Method | Stage | Success | Best | Strict | Mean Best | Mean Avg | Mean Worst | Geo-mean Runtime (s) |
> |---|---|---:|---:|---:|---:|---:|---:|---:|
> | HGA-QE | Net. Size | 49/52 | **47/49** | 5/49 | 217.39 | 222.12 | **227.86** | 0.066 |
> | HGA-QE | # LUT | 49/52 | 30/48 | 3/48 | 83.46 | 87.45 | 91.73 | **3.036** |
> | SA | Net. Size | **52/52** | 36/49 | 1/49 | **215.47** | **222.06** | 235.41 | 0.092 |
> | SA | # LUT | **52/52** | **35/48** | 6/48 | 85.04 | 90.52 | 98.50 | 3.041 |
> | GA | Net. Size | **52/52** | 42/49 | 1/49 | 274.24 | 1575.83 | 2564.90 | 0.080 |
> | GA | # LUT | 50/52 | 30/48 | 4/48 | **83.38** | **87.24** | **90.35** | 3.060 |
> | SIFT | Net. Size | **52/52** | 18/49 | 0/49 | 369.10 | 369.10 | 369.10 | **0.023** |
> | SIFT | # LUT | **52/52** | 30/48 | 4/48 | 134.52 | 134.52 | 134.52 | 3.229 |

---

### Review · Reviewer_gKa1 · 2026-07-05

**Summary Of Contributions:**

This paper studies BDD variable ordering for BDD-based reversible circuit synthesis, with the goal of reducing the quantum cost of the synthesized circuit rather than optimizing BDD size as a proxy. The authors formulate QCC-aware variable ordering as a heuristic design problem and propose QuantumEvo, an LLM-driven evolutionary framework that generates executable C/CUDD heuristic programs, evaluates them through downstream reversible circuit synthesis, and iteratively improves them using fitness feedback. The best discovered heuristic, HGA-QE, is a hybrid genetic algorithm that replaces standard sifting in the GA pipeline with a more targeted PartialSift/MiniSift procedure intended to preserve more ordering diversity while still keeping BDD size competitive.

Key strengths are the practically relevant objective, the integration of LLM-based heuristic search with a real synthesis pipeline, and the fact that the generated heuristic is interpretable enough to analyze at the algorithmic level rather than being only a black-box predictor. The paper also usefully diagnoses why MiniSift may help, including diversity and same-BDD-size/QCC comparisons.

The main weaknesses are in the interpretation and accounting of the empirical evidence. The reported advantage relies heavily on tie-or-win results rather than strict wins; it is not clear whether the benchmark gains mainly come from instances structurally similar to the search set used during heuristic discovery or also hold on structurally different instances; and the paper does not yet provide a clear cost-benefit account of runtime, offline LLM discovery overhead, and token consumption.

**Audience:**

Yes

**Audience Explanation:**

I expect the paper to be of interest to at least several parts of the TMLR audience. First, it contributes to the growing area of LLM-driven heuristic and algorithm design, where the goal is not simply to use an LLM as a predictor but to generate executable heuristic programs that can be evaluated inside a domain-specific optimization pipeline. The paper provides a concrete case study showing how this paradigm can be applied to a specialized synthesis problem with a downstream objective.

Second, the paper is relevant to readers interested in quantum compilation and reversible logic synthesis. While the method is specialized, reducing quantum cost for BDD-based synthesis is a meaningful objective, and the empirical results suggest that LLM-driven heuristic search can produce competitive ordering heuristics in this setting.

The audience may be somewhat specialized because the application domain requires familiarity with BDDs, reversible synthesis, and quantum-cost metrics. However, the broader methodological framing should make the work accessible and useful to a wider TMLR readership interested in learned or LLM-assisted algorithm design.

**Broader Impact Concerns:**

I do not see major ethical or broader-impact concerns that would require a dedicated Broader Impact Statement. The work is primarily a technical contribution to heuristic design for BDD-based reversible/quantum circuit synthesis. Its likely impacts are methodological and computational rather than directly societal.

**Claims And Evidence:**

Yes

**Claims Explanation:**

Overall, the main technical claims are supported by reasonably accurate and relevant evidence. The paper evaluates the discovered heuristic on held-out benchmark functions from RevLib, LGSynth91, and ISCAS85/89, compares against several standard CUDD variable-ordering methods and BDD2Seq, and reports both aggregate and per-function results. The ablation studies are also useful: they support the claims that direct QCC-based fitness is preferable to BDD-size-based fitness, and that mixed initialization from multiple heuristic families improves transfer to structurally different benchmarks. The diagnostic comparison between standard sifting and MiniSift further gives a plausible explanation for why the discovered method can reduce QCC even when BDD size is similar.

However, some claims would be more convincing with clearer experimental reporting and interpretation:

1. The runtime evidence is not fully specified. The paper defines a synthesis time budget and reports Avg/Total Time, but it is not sufficiently clear whether these timings include BDD construction, reordering, circuit synthesis, QCC evaluation, parsing/I/O, heuristic compilation, or BDD2Seq inference/training costs. This matters because the paper compares stochastic methods, deterministic CUDD baselines, and a learning-based baseline whose reported numbers may not have exactly the same timing protocol.

2. The paper does report both "Best" counts including ties and "Strict" wins, and these numbers are important for interpreting the strength of the empirical advantage. The overall tie-or-win rate of 70.9% is encouraging, but the strictly-best rate is only 13.5%, indicating that a substantial part of the apparent advantage comes from matching the best baseline rather than clearly outperforming it. This distinction is also visible in the per-suite results: on RevLib, for example, HGA-QE has a high Best count, but its strict wins are fewer than BDD2Seq's, and its mean best QCC is very close to SA's. The advantage appears clearer on LGSynth91 and ISCAS85/89, but overall the evidence supports a competitive and often robust method more strongly than it supports broad dominance over prior methods.

3. The evidence for transfer beyond the search distribution would be clearer with a more explicit analysis of how similarity between the search set and benchmark set affects performance. Appendix Figure 9 compares the two sets using structural properties such as input count, output count, and gate count, and it suggests that some benchmark instances are well covered by the search set while others are more structurally distinct. Because a large part of the reported advantage comes from ties rather than strict wins, it is hard to tell from the aggregate results whether the strong benchmark performance is mainly obtained on instances that resemble the search set used during heuristic discovery. Conversely, it remains unclear whether HGA-QE still provides an advantage when the benchmark instances are structurally different from the discovery data.

4. Because QuantumEvo uses downstream QCC as the fitness signal during heuristic discovery and the final evaluation is also based on QCC, the paper should better isolate what is gained from LLM-driven heuristic design beyond simply having access to the evaluation objective during search. Optimizing and evaluating the same metric is reasonable, but the current evidence would be more convincing if it included a stronger QCC-aware control, such as a non-LLM evolutionary/random search over comparable heuristic variants or a QCC-aware version of a classical ordering search. This would help distinguish the benefit of objective alignment from the benefit of the LLM-generated heuristic program.

5. The paper does not sufficiently discuss the overhead of using an LLM-driven design process. The reported benchmark runtimes appear to measure the cost of applying the final heuristic, but QuantumEvo also incurs offline cost from repeated LLM calls, token consumption, candidate program generation/compilation, and QCC-based evaluation. Token consumption is not merely an implementation detail for an LLM-based method, since LLM usage is often priced and constrained by input/output tokens. The paper should quantify both time and token overhead, separate offline discovery cost from online application cost, and compare these costs with the baselines where applicable. Given the 13.5% strictly-best rate and the large fraction of ties, the paper should directly discuss whether the LLM token and time cost is justified by the measured QCC improvements.

In summary, I find the main empirical claims mostly supported, but the clarity and persuasiveness of the evidence would improve with a more precise timing protocol, a more explicit interpretation of what the gap between tie-or-win counts, strict wins, and average-cost improvements implies, a clearer breakdown of performance across search-like versus structurally distinct benchmark instances, a stronger control for the effect of using QCC as the discovery fitness, and a clearer accounting of offline versus online overhead.

**Requested Changes:**

Critical changes:

1. Clarify the runtime measurement protocol. The paper should explicitly state what is included in the reported Avg Time and Total Time: BDD construction, variable reordering, reversible circuit synthesis, QCC evaluation, benchmark parsing/I/O, and/or heuristic compilation. It should also clarify how BDD2Seq timing numbers are obtained and whether they are measured under the same protocol and hardware. Without this information, the runtime comparisons are difficult to interpret fairly.

2. Calibrate the empirical claims to the distinction between tie-or-win and strict improvement. The paper reports both Best counts including ties and Strict wins, which is good, but the interpretation should more directly reflect the gap between these metrics. The overall tie-or-win rate of 70.9% is encouraging, while the strict-best rate of 13.5% indicates that much of the advantage comes from matching the best baseline rather than clearly outperforming it. The authors should frame the results as showing competitive and robust QCC performance, with clearer gains on LGSynth91 and ISCAS85/89, rather than implying broad dominance across baselines and suites.

3. Add a performance breakdown by similarity between the search set and benchmark set. Appendix Figure 9 compares the search and benchmark sets by structural descriptors, but the benchmark results are mostly reported in aggregate or by suite. To support the claim that the discovered heuristic transfers beyond the search distribution, the authors should separate benchmark instances that are structurally similar to the search set used during heuristic discovery from those that are structurally different, and report tie-or-win, strict-win, and mean-QCC behavior for each group. This would clarify whether the good benchmark results mainly come from cases where the benchmark resembles the discovery data, or whether the method remains advantageous when the search and benchmark distributions do not match.

Changes that would strengthen the work:

1. Add a stronger QCC-aware control to isolate the contribution of LLM-driven heuristic design. QuantumEvo uses downstream QCC as the fitness signal during heuristic discovery, and the final evaluation is also based on QCC. This is a reasonable objective choice, but it makes it important to distinguish the benefit of using the target metric during search from the benefit of the LLM-generated heuristic program itself. A non-LLM QCC-aware search/control, such as a classical evolutionary or random search over comparable heuristic variants or a QCC-aware adaptation of an existing ordering search, would make the claim that LLM-driven heuristic design adds value more convincing.

2. Quantify and discuss the offline and online overhead of QuantumEvo, including token cost, and compare it with the baselines where applicable. The paper should separate the one-time offline cost of discovering HGA-QE from the online cost of applying the discovered heuristic to a new Boolean function. For the offline phase, the authors should report the number of LLM calls, approximate input/output token counts or token budget, total wall-clock time, and the compute used for generating, compiling, and evaluating candidate heuristics. The token overhead should not be treated as a minor implementation detail: for an LLM-based method, token usage is a primary cost driver and may be expensive relative to the achieved performance gain. The paper should also state whether the cost of using a large pretrained LLM is treated as external or amortized, and discuss this assumption explicitly. For the online phase, the authors should state whether applying HGA-QE requires any LLM calls, or whether it is simply a fixed C/CUDD heuristic whose cost is reflected in the benchmark runtime tables. These costs should be compared with the corresponding costs of the classical CUDD baselines and BDD2Seq to the extent possible, distinguishing one-time training/discovery/tuning cost from per-instance inference or synthesis cost. Finally, the authors should provide a cost-benefit discussion: given the 13.5% strictly-best rate and the large fraction of ties, is the LLM token and time overhead justified by the achieved QCC improvements?

---

> ### Author Response · Authors · 2026-07-20
> **Response to Reviewer 2 (Required Changes #1, #2, #3)**
>
> We thank the reviewer for the careful and detailed reading, and for identifying the points where our reporting needed to be more precise. We address each below. Full details are in the revised sections/appendices.
>
> ## Required Changes
>
> ### 1. Clarified runtime measurement protocol and hardware differences (Addressed in: Section 4.2.2)
> > Clarify the runtime measurement protocol. ... It should also clarify how BDD2Seq timing numbers are obtained and whether they are measured under the same protocol and hardware.
>
> We understood this as two separate questions: what our own reported times include, and whether BDD2Seq's times are comparable to them. On the first, we clarified that Avg Time denotes the mean runtime per seed, and Total Time denotes the cumulative runtime across all seeds. Both measurements include BDD construction, BDD variable ordering, and reversible-circuit synthesis.
>
> On the second question, the original BDD2Seq checkpoint is not publicly available. We before use the QCC and runtime values reported in the original paper rather than rerunning BDD2Seq on our hardware. Because the original experiments used dual Intel Xeon Platinum 8375C processors, whereas ours use a single-thread AMD Ryzen 9 5900X, we apply a hardware normalization to the reported runtime of BDD2Seq(B*). Specifically, we leave the BDD2Seq(E*) runtime unchanged because its single synthesis call represents only a small fraction of the total runtime, and rescale only the estimated cost of the 19 additional CPU-based synthesis calls performed by BDD2Seq(B*). The normalization uses the PassMark single-thread ratio $\gamma=2474/3465\approx0.714$ between the two platforms. The complete assumptions and derivation are provided in Appendix G. In particular, the calculation assumes that the reported runtime difference between BDD2Seq(B*) and BDD2Seq(E*) is attributable to the 19 additional synthesis calls.
>
>
>
> ### 2. Recalibrated empirical claims based on tie-or-win vs. strict improvement (Addressed in: Abstract, Introduction, Section 4.2.2)
> > Calibrate the empirical claims to the distinction between tie-or-win and strict improvement. ... The authors should frame the results as showing competitive and robust QCC performance, with clearer gains on LGSynth91 and ISCAS85/89, rather than implying broad dominance.
>
> We agree with this framing and adopted it. We revised the abstract, the introduction, and Section  4.2.2. The revised contribution statement now characterizes HGA-QE as broadly competitive with both classical and learning-based baselines, while noting that its relative advantage in strict wins is clearer on the two benchmark suites drawn from sources different from those used for heuristic discovery. We also made these suite-level differences explicit in Section 4.2.2.
>
>
> ### 3. Added performance analysis by search–benchmark structural similarity (Addressed in: Appendix B.3, Section 4.2.2)
> > Add a performance breakdown by similarity between the search set and benchmark set. ... separate benchmark instances that are structurally similar to the search set from those that are structurally different.
>
> We agree that the original manuscript presented the structural comparison in Appendix B.3 and the suite-level results separately, without explicitly connecting them. We revised Appendix B.3 to make this connection clear. Figure 9 in Appendix B.3 compares the search and benchmark sets in terms of input, output, and gate counts. In these projections, the benchmark points that overlap with the search-set distribution are the RevLib functions. Although the search set, augmented-RevLib validation set, and RevLib benchmark set contain no shared functions, they are drawn from the same source family. By contrast, the benchmark points outside the search-set distribution correspond to LGSynth91 and ISCAS85/89.
> The suite-level results in Table 2 show that HGA-QE is broadly comparable to the strongest baselines on RevLib, while its relative advantage in strict wins is clearer on LGSynth91 and ISCAS85/89. We therefore revised Section 4.2.2 to state more precisely that HGA-QE transfers to benchmark functions not used during discovery, with clearer gains on the suites drawn from structurally distinct source families, rather than claiming uniform superiority across all suites.

---

> ### Author Response · Authors · 2026-07-20
> **Response to Reviewer 2 (Recommended Changes #1, #2)**
>
> ## Recommended Changes
>
> ### 1. Added a QCC-aware non-LLM control to isolate the contribution of LLM-driven design (Addressed in: Section 4.2.3, Appendix C.1, Appendix C.3)
> > Add a stronger QCC-aware control to isolate the contribution of LLM-driven heuristic design. ... A non-LLM evolutionary/random search over comparable heuristic variants ... would make the claim that LLM-driven heuristic design adds value more convincing.
>
> We implemented this control in Section 4.2.3. The non-LLM search retains QuantumEvo's outer evolutionary loop, QCC-based fitness, search/validation split, and search budget, but replaces LLM-based heuristic generation with a search over a predefined library of sifting, GA, and SA components, as detailed in Appendix C.1.
>
> On the validation set, the non-LLM control seems to perform stronger than the LLM-driven search, and all ten runs converge to GA-family individuals. However, the difference becomes clearer in transfer to the full set of 148 benchmark functions. The non-LLM control is strictly best on 5.4% of the functions, compared with 13.5% for HGA-QE, and obtains a higher Mean Avg QCC. This comparison is reported in the Non-LLM row of Table 4 in Section 4.2.3.
>
> We do not interpret these results as showing that LLM guidance is uniformly superior. Rather, they show that the LLM-driven search can discover algorithmic variants beyond the predefined component library and, in this study, produced an HGA-QE finalist that transferred more effectively to the full benchmark.
>
> | Guidance | Best/Total (%) | Strictly Best/Total (%) | Mean Best QCC | Mean Avg QCC | Mean Worst QCC |
> |---|---:|---:|---:|---:|---:|
> | **LLM** | **70.9** | **13.5** | **1923.83** | **1979.76** | **2033.46** |
> | Non-LLM control | 66.9 | 5.4 | 2075.15 | 2220.27 | 2432.07 |
>
>
> ### 2. Added offline and online cost analysis of QuantumEvo. (Reflected in: Appendix E**)
> > Quantify and discuss the offline and online overhead of QuantumEvo, including token cost, and compare it with the baselines where applicable.
>
> We agree the one-time offline cost should be reported separately from the recurring online cost, and added this analysis in Appendix E.
> - Offline cost: Across the ten full QuantumEvo searches using gpt-oss-120b, each run required, on average, 291.6 LLM calls, 3.11 million input tokens, and 1.04 million output tokens, for approximately 4.15 million tokens in total. A run took an average of 108.5 minutes of wall-clock time using one self-hosted NVIDIA RTX PRO 6000 GPU. We treat the pretraining cost of the underlying LLM as external and report only the inference resources consumed during heuristic discovery. Each candidate is compiled once before evaluation. Evaluating approximately 200 candidates on 100 search functions requires about 40,000 synthesis calls per run, because each candidate--function evaluation performs two synthesis calls.
>     | Quantity | Average per production run |
>     |---|---:|
>     | LLM calls | 291.6 |
>     | Input tokens | 3.11M |
>     | Output tokens | 1.04M |
>     | Wall-clock time | 108.5 min |
>     | Hardware | 1× NVIDIA RTX PRO 6000 |
> - Online cost: After discovery, HGA-QE is deployed as a fixed compiled C heuristic and requires no further LLM calls or GPU inference. Its per-instance online cost consists of BDD construction and variable ordering, reversible-circuit synthesis, and QCC evaluation, all of which are included in the benchmark runtimes. By contrast, BDD2Seq requires neural inference on a GPU for each new instance. Appendix E further emphasizes that HGA-QE runs entirely on the CPU after discovery, whereas BDD2Seq incurs per-instance GPU inference.
>
> The large number of ties should be interpreted in the context of this problem, where QCC is discrete and different variable orderings can produce circuits with the same cost. The 13.5% strict-best rate therefore captures unique improvements, whereas the broader win-or-tie rate reflects overall competitiveness. Moreover, the statistical tests in Appendix D.3.1, conducted over the full benchmark set, show that HGA-QE achieves statistically significant QCC improvements over several baselines. These results demonstrate meaningful performance gains, but do not imply that the LLM overhead is justified in every setting. The trade-off is more favorable when QCC reductions are particularly valuable and the discovered heuristic can be reused across many synthesis tasks.

---

### Review · Reviewer_UDrx · 2026-07-07

**Summary Of Contributions:**

This paper studies BDD variable ordering for reversible/quantum circuit synthesis. Existing BDD ordering methods often optimize BDD size, since BDD size is closely related to the size of the synthesized circuit. The authors argue that BDD size is only an indirect proxy for the final quantum cost of the synthesized circuit (QCC), and that different variable orderings may lead to similar BDD sizes but different QCC values.
To address this, the paper proposes QuantumEvo, an LLM-driven evolutionary framework for designing BDD variable ordering heuristics. Instead of using the LLM to directly predict an ordering for each input function, QuantumEvo uses the LLM to generate executable heuristic programs. These heuristics are implemented as C functions using standard CUDD operations, initialized from existing heuristic families such as sifting, genetic algorithms, and simulated annealing, and evaluated using downstream QCC.
The best discovered heuristic, HGA-QE, is a hybrid GA/sifting method. Its main modification is to replace standard sifting inside the GA with a more local targeted sifting procedure, MiniSift/PartialSift. The experiments evaluate HGA-QE on RevLib, LGSynth91, and ISCAS85/89 benchmarks against several classical BDD ordering heuristics and BDD2Seq. The results show that HGA-QE is competitive and achieves lower QCC on a subset of benchmark functions.

**Audience:**

Yes

**Audience Explanation:**

This paper would be of interest to part of the TMLR audience. Although reversible/quantum circuit synthesis is a specialized application area, the work is also relevant to LLM-driven algorithm design, automatic heuristic discovery, combinatorial optimization, ML for EDA, and quantum compilation.
The paper does not use the LLM merely to output a direct solution. Instead, it uses the LLM to generate reusable heuristic programs, which are then evaluated by a domain-specific downstream objective. This makes the work relevant to broader questions about how LLMs can assist algorithm design beyond natural language tasks. Therefore, even if the audience is specialized, the paper should still be of interest to some TMLR readers.

**Broader Impact Concerns:**

I do not see major broader impact concerns. The paper studies an algorithmic optimization problem in reversible/quantum circuit synthesis and does not involve human subjects, personal data, privacy-sensitive information, or socially sensitive decision-making.
A short broader impact statement would be sufficient. The authors may mention that the work could contribute to automated quantum compilation and automatic heuristic design, while also noting that LLM-driven search can involve nontrivial computational cost.

**Claims And Evidence:**

No

**Claims Explanation:**

The paper provides reasonable evidence for the main motivation: BDD size is not a perfect proxy for downstream QCC, and directly considering QCC is meaningful for BDD-based reversible circuit synthesis. The experiments also show that the discovered HGA-QE heuristic is competitive with existing BDD variable ordering methods and improves QCC on some instances.
However, the current evidence does not fully support the stronger parts of the paper’s claim. The main issue is that the roles of different components are not clearly separated. The paper combines QCC-based evaluation, LLM-driven heuristic generation, multiple seed heuristic families, and the final MiniSift/PartialSift modification. As a result, it is not fully clear whether the observed gains mainly come from the QCC-aware objective, from the LLM-driven search process, or from a relatively simple modification to the GA/sifting procedure.
The method’s novelty also needs to be stated more carefully. QuantumEvo applies an LLM-based algorithm design framework to a meaningful domain problem, which is valuable. However, the final heuristic HGA-QE is still structurally close to existing GA/sifting-based BDD reordering methods. The contribution is better described as discovering an effective hybrid GA/sifting heuristic for QCC-aware BDD ordering, rather than introducing a fundamentally new variable ordering paradigm.
Finally, some of the empirical improvements appear modest against the strongest baselines. This does not invalidate the contribution, but the paper should be more precise in its conclusions. The results support that HGA-QE is competitive and sometimes better, but not that it clearly dominates existing methods across all settings.

**Requested Changes:**

1. Clarify the main claim and the logic of the paper.
The paper should more clearly connect its motivation, method, and evidence. The motivation is that BDD size is not always aligned with QCC, so a QCC-aware ordering method is needed. The method is an LLM-driven heuristic search framework, while the final output is a GA/sifting variant. These are related but distinct points.
The authors should clearly state what the paper is primarily claiming: that QCC-aware evaluation is useful, that LLM-driven heuristic search is useful, or that the discovered MiniSift/PartialSift modification is useful. The current version mixes these claims, making it harder to assess what has been demonstrated.
2. Calibrate the novelty claim.
HGA-QE is mainly a hybrid GA/sifting heuristic. Its key change is to replace standard sifting with a more local targeted sifting procedure. This is a useful modification, but it should not be framed as a fundamentally new BDD variable ordering paradigm.
The authors should describe the contribution more precisely: QuantumEvo discovers an effective hybrid GA/sifting heuristic for QCC-aware BDD ordering. This would make the novelty claim more accurate and better aligned with the actual algorithm.
3. Add focused ablation studies.
The paper should add a small number of targeted ablations to explain where the improvement comes from. The most important comparisons are:
- QCC-based fitness versus BDD-size-based fitness;
- the full QuantumEvo framework versus the final HGA-QE/MiniSift variant alone;
- LLM-generated modifications versus simple non-LLM modifications or random/evolutionary modifications;
- multiple seed heuristic families versus a single seed family.
These ablations are important because the current results do not fully isolate the contribution of LLM-driven search from the contribution of the final local sifting modification.
4. Provide clearer result analysis.
The aggregate results show that HGA-QE is competitive, but they do not fully explain when and why it works. The paper should include a more concise per-instance or grouped analysis comparing HGA-QE with the strongest baselines, especially SA and BDD2Seq.
The authors should also discuss cases where the improvement is small or where HGA-QE loses. This would make the conclusions more balanced and help readers understand the scope of the method’s advantage.

---

> ### Author Response · Authors · 2026-07-20
> **Response to Reviewer 3 (Required Changes #1, #2, #3)**
>
> We thank the reviewer for pushing us to state precisely what the paper claims.  We address each requested change below.
>
> ## Required Changes
>
> ### 1. Clarified the main claim and the logic of the paper (Addressed in: Introduction, Section 4.2.1, Section 4.2.2, Section 4.2.3)
>
> > Clarify the main claim and the logic of the paper. ... The authors should clearly state what the paper is primarily claiming.
>
> We agree that the original manuscript did not clearly separate its related claims. We therefore revised the manuscript to distinguish the primary framework contribution from the contribution of the discovered heuristic and to connect each claim to the evidence that supports it.
>
> First, the paper introduces a QCC-aware framework for BDD variable-ordering heuristic discovery. Finding 1 motivates direct QCC-based evaluation by showing that BDD size is an imperfect proxy for downstream QCC. Second, the search produces HGA-QE, a practically effective hybrid GA whose performance is evaluated against classical and learning-based baselines in Section 4.2.2.
> Accordingly, the revised Introduction organizes the contributions into two parts: the QCC-aware discovery framework and the discovered HGA-QE heuristic with its empirical benchmark evaluation.
>
> ### 2. Recalibrated the novelty claim of HGA-QE (Addressed in: Introduction, Section 3.3, Discussion.)
> > Calibrate the novelty claim. ... The authors should describe the contribution more precisely: QuantumEvo discovers an effective hybrid GA/sifting heuristic for QCC-aware BDD ordering.
>
> We adopted this framing directly. We removed "novel" from Contribution (2) in the introduction and revised it to state that QuantumEvo discovers an *effective* hybrid genetic algorithm that replaces standard sifting with a targeted sifting procedure better aligned with QCC, rather than describing HGA-QE as a fundamentally new variable-ordering paradigm.
>
>
> ### 3. Added focused ablations on the sources of performance improvement (Addressed in: Section 3.3, Section 4.2.3, Appendix C.1, Appendix C.3)
> > Add focused ablation studies: QCC-based fitness vs. BDD-size-based fitness; the full QuantumEvo framework vs. the final HGA-QE/MiniSift variant alone; LLM-generated modifications vs. non-LLM/random modifications; multiple seed families vs. a single seed family.
>
> - Two of the four analyses were already included in Sections 4.2.2 and 4.2.3 of the original manuscript. In the revision, we relabeled them in Section 4.2.3 and made their connection to the corresponding claims explicit. Additionally, we report an elite-candidate view (top-5/top-10 candidates by search-set rank, pooled per setting) in Appendix C.3 rather than only each run's single finalist.
> -  Full QuantumEvo vs. HGA-QE/MiniSift alone. This comparison presents a conceptual difficulty because the defining structure of HGA-QE, particularly the MiniSift and PartialSift subroutines, does not exist before the QuantumEvo search. Evaluating “HGA-QE without QuantumEvo” would therefore require assuming prior knowledge of the algorithmic structure that the framework is intended to discover.
> -  We consequently addressed this request together with the fourth item, which asks for an LLM-versus-non-LLM comparison. This yields a runnable control without assuming the answer in advance. Specifically, we constructed a non-LLM search in Appendix C.1 that retains QuantumEvo's evolutionary loop, fitness function, search/validation split, and search budget, while replacing only the LLM proposal step with search over a predefined library of sifting, GA, and SA components. Under this comparison, the non-LLM control is stronger under some validation summaries, but it transfers less effectively to the full benchmark. On the 148-function benchmark, it is strictly best on 5.4% of circuits, compared with 13.5% for HGA-QE, and obtains a higher Mean Avg QCC. These results are reported in the Non-LLM row of Table 4 in Section 4.2.3. All ten non-LLM runs also converge to the GA family. We do not interpret this result as showing that LLM guidance is uniformly superior. Rather, it shows that LLM-based generation can explore algorithmic variants beyond the predefined component library and, in this study, produced an HGA-QE finalist that transferred more effectively to the full benchmark.
>
>     | Method | Best/Total (%) | Strictly Best/Total (%) | Mean Best QCC | Mean Avg QCC | Mean Worst QCC |
>     |---|---:|---:|---:|---:|---:|
>     | **LLM** | **70.9** | **13.5** | **1923.83** | **1979.76** | **2033.46** |
>     | Non-LLM control | 66.9 | 5.4 | 2075.15 | 2220.27 | 2432.07 |

---

> ### Author Response · Authors · 2026-07-20
> **Response to Reviewer 3 (Required Changes #4)**
>
> ### 4. Provided a more detailed result analysis, including loss cases (Addressed in: Section 4.2.2, Appendix D.3.1, Appendix D.3.2)
> > Provide clearer result analysis. ... a more concise per-instance or grouped analysis comparing HGA-QE with the strongest baselines, especially SA and BDD2Seq. ... discuss cases where the improvement is small or where HGA-QE loses.
>
> - We revised Section 4.2.2 to present the grouped, suite-level comparison directly. On RevLib, HGA-QE achieves the highest Best count (70/96), but its mean QCC is comparable to that of SA, while BDD2Seq achieves the most strict wins. Consistent with this observation, a paired Wilcoxon test with Holm correction finds no significant difference between HGA-QE and SA on RevLib ($p=0.571$). The strongest statistical gains appear on LGSynth91, where HGA-QE significantly outperforms every baseline except GA. On ISCAS85/89, HGA-QE shows a large descriptive improvement, but the suite contains only four circuits and therefore does not support a statistically significant conclusion. But we can say that across all 148 functions, HGA-QE is significantly better than every baseline.
>
>     | Suite | $n$ | Mean $\Delta$QCC vs. SA | $p$ (Holm) | Mean $\Delta$QCC vs. GA | $p$ (Holm) | Mean $\Delta$QCC vs. BDD2Seq | $p$ (Holm) |
>     |---|---:|---:|---:|---:|---:|---:|---:|
>     | RevLib | 96 | $-0.21$ | 0.571 | $-29.24$ | 0.139 | $-39.99$ | **0.025** |
>     | LGSynth91 | 48 | $-65.04$ | **0.006** | $-123651.40$ | 0.055 | $-377.46$ | **0.002** |
>     | ISCAS85/89 | 4 | $-151.75$ | 0.653 | $-161.75$ | 0.653 | $-1415.25$ | 0.653 |
>     | All | 148 | $-25.33$ | **0.008** | $-40126.49$ | **0.008** | $-186.61$ | **$1.2{\times}10^{-5}$** |
>
>
> - For loss cases, in Appendix D.3.2 Table 13 reports that HGA-QE is not (tied-)best on 43/148 circuits (29.1%), which is the *lowest* loss rate of any method on every suite (SA: 55/148 (37.2%), BDD2Seq: 60/148 (40.5%), GA: 64/148 (43.2%)). As shown in the table below, among HGA-QE's 43 losses, 21 are attributed to BDD2Seq and 11 to SA.
>
>     | Suite | HGA-QE losses | BDD2Seq | SA | GA | Others |
>     |---|---:|---:|---:|---:|---:|
>     | RevLib | 26/96 | 15 | 7 | 1 | 3 |
>     | LGSynth91 | 17/48 | 6 | 4 | 6 | 1 |
>     | ISCAS85/89 | 0/4 | 0 | 0 | 0 | 0 |
>     | Overall | 43/148 | 21 | 11 | 7 | 4 |